# Linear Partial Gromov-Wasserstein Embedding

Yikun Bai[1], Abihith Kothapalli[*1], Hengrong Du[*2],
Rocio Diaz Martin[*3], and Soheil Kolouri[1]

[1]Department of Computer Science, Vanderbilt University
[2]Department of Mathematics, University of California, Irvine
[3]Department of Mathematics, Tufts University
[1]yikun.bai, abi.kothapalli, soheil.kolouri@vanderbilt.edu
[2]hengrond@uci.edu
[3]rocio.diaz_martin@tufts.edu

## Abstract

The Gromov–Wasserstein (GW) problem, a variant of the classical optimal transport (OT) problem, has attracted growing interest in the machine learning and data science communities due to its ability to quantify similarity between measures in different metric spaces. However, like the classical OT problem, GW imposes an equal mass constraint between measures, which restricts its application in many machine learning tasks. To address this limitation, the partial Gromov-Wasserstein (PGW) problem has been introduced. It relaxes the equal mass constraint, allowing the comparison of general positive Radon measures. Despite this, both GW and PGW face significant computational challenges due to their non-convex nature. To overcome these challenges, we propose the linear partial Gromov-Wasserstein (LPGW) embedding, a linearized embedding technique for the PGW problem. For $K$ different metric measure spaces, the pairwise computation of the PGW distance requires solving the PGW problem $\mathcal{O}(K^2)$ times. In contrast, the proposed linearization technique reduces this to $\mathcal{O}(K)$ times. Similar to the linearization technique for the classical OT problem, we prove that LPGW defines a valid metric for metric measure spaces. Finally, we demonstrate the effectiveness of LPGW in practical applications such as shape retrieval and learning with transport-based embeddings, showing that LPGW preserves the advantages of PGW in partial matching while significantly enhancing computational efficiency. The code is available at https://github.com/mint-vu/Linearized_Partial_Gromov_Wasserstein.

## 1 Introduction

**Optimal transport (OT), unbalanced and partial OT, and linearized OT.** The core objective of this work, and a general challenge in machine learning, is to match objects and devise practical, computationally efficient notions of similarity between shapes. One of the most well-known techniques for this task arises from optimal transport (OT) theory, which has found extensive applications in generative modeling (Arjovsky et al., 2017; Genevay et al., 2016), domain adaptation (Courty et al., 2016; Balaji et al., 2020), representation learning (Wu et al., 2023), and many other domains. The classical OT problem, as detailed by (Villani, 2009), involves matching two probability distributions to minimize the expected transportation cost, giving rise to the so-called Wasserstein distances. Although the classical OT problem imposes a mass conservation requirement, recent advances have extended the utility of OT beyond probability measures, allowing for comparisons of non-negative measures with different total masses and partial matching of sets through the unbalanced and partial OT frameworks (Chizat et al., 2018a;b; Fatras et al., 2021; Séjourné et al., 2021). Additionally, in large-scale machine learning applications, since the OT approach is computationally expensive, it has motivated numerous approximations that lead to significant speedups, such as those in (Cuturi, 2013; Chizat et al., 2020; Scetbon & Cuturi,

---

*These authors contributed equally to this work.

2022), and the linearization technique, termed linear optimal transportation (LOT), proposed by (Wang et al., 2013). There have also been advances in reducing the computational complexity of comparing unbalanced measures (Cai et al., 2022; Bai et al., 2023).

**The Gromov-Wasserstein (GW) distance, unbalanced and partial GW, and linearized GW.** Another line of research has focused on the comparison of probability measures across *different* metric spaces using Gromov-Wasserstein (GW) distances (Mémoli, 2011). These innovations have applications in areas ranging from quantum chemistry (Peyré et al., 2016) to natural language processing (Alvarez-Melis & Jaakkola, 2018), enhancing fundamental tasks such as clustering (Chowdhury & Needham, 2021), dimensionality reduction (Van Assel et al., 2024), shape correspondence (Kong et al., 2024), shape analysis (Mémoli & Needham, 2022), and inference-based simulation (Hur et al., 2024). The GW distance is a valuable tool across a variety of domains, as it allows for comparisons between distributions while being independent of the ambient spaces. To extend GW to unbalanced settings, GW-like comparisons have been developed for metric measure spaces with differing total masses (Liu et al., 2020; Chapel et al., 2020; Séjourné et al., 2021; Bai et al., 2024; Zhang et al., 2022). Furthermore, to address the computational expense of pairwise GW distance calculations in large-scale applications, a linearized version of GW has been proposed in (Beier et al., 2022). Similar to the LOT framework (Wang et al., 2013), this approach is based on barycentric projection maps of transport plans.

**Contributions.**

- We propose the linear Partial Gromov-Wasserstein embedding (LPGW) technique, which extends the classical LOT embedding technique to the partial GW setting.

- Based on the embedding, we propose the so-called LPGW distance, which we prove to be a metric under certain assumptions. Given $K$ different metric measure spaces, computing their pairwise PGW distances requires solving the PGW problem $\mathcal{O}(K^2)$ times. However, computing the LPGW distance pairwise requires only $\mathcal{O}(K)$ distance computations.

- Numerically, we test our proposed LPGW embedding and LPGW distance in two experiments: shape retrieval and learning with transform-based embeddings. In both experiments, we observe that the LPGW-based approach can preserve the partial matching property of PGW while significantly improving computational efficiency.

## 2 BACKGROUND

### 2.1 THE WASSERSTEIN DISTANCE

Let $\mathcal{P}(\mathbb{R}^d)$ be the space of probability measures on the Borel $\sigma$-algebra on $\mathbb{R}^d$, and let $\mathcal{P}_2(\mathbb{R}^d) := \{\mu \in \mathcal{P}(\mathbb{R}^d) : \int_{\mathbb{R}^d} |x|^2 d\mu(x) < \infty\}$. Given a (measurable) mapping $T : \mathbb{R}^d \to \mathbb{R}^d$, the pushforward measure $T_\#\mu$ is defined as $T_\#\mu(B) := \mu(T^{-1}(B))$ for all Borel sets $B \subseteq \mathbb{R}^d$, where $T^{-1}(B) := \{x : T(x) \in B\}$ is the preimage of $B$ under $T$.

Given $\mu, \nu \in \mathcal{P}_2(\mathbb{R}^d)$, the **2-Wasserstein distance** is defined as

$$W_2^2(\mu, \nu) := \min_{\gamma \in \Gamma(\mu, \nu)} \int_{\mathbb{R}^d \times \mathbb{R}^d} \|x - y\|^2 d\gamma(x, y) \tag{1}$$

where $\Gamma(\mu, \nu)$ is set of joint probability measures on $\mathbb{R}^d \times \mathbb{R}^d$ whose first and second marginals are $\mu$ and $\nu$, respectively. The classical OT theory (Villani, 2021; 2009) guarantees that a minimizer to (1) always exists, so such a formulation is well-defined. In fact, there might exist multiple minimizers, and we denote by $\Gamma^*(\mu, \nu)$ the set of all optimal transportation plans for problem (1).

When an optimal transportation plan $\gamma$ is induced by a mapping $T : \mathbb{R}^d \to \mathbb{R}^d$, that is,

$$\gamma = (\mathrm{id} \times T)_\#\mu \in \Gamma^*(\mu, \nu), \qquad \text{with} \quad T_\#\mu = \nu, \tag{2}$$

we say that the **Monge mapping assumption** holds, and the function $T$ is called a Monge mapping. In particular, by Brenier's theorem, when $\mu$ is a continuous measure (i.e.,

absolutely continuous with respect to the Lebesgue measure on $\mathbb{R}^d$), a Monge mapping $T$ always exists and it is unique.

We refer to Appendix A for a detailed introduction to the LOT formulation.

## 2.2 THE GROMOV-WASSERSTEIN DISTANCE

Consider two compact gauged measure spaces (gm-spaces) $\mathbb{X} = (X, g_X, \mu)$ and $\mathbb{Y} = (Y, g_Y, \nu)$, where $X \subset \mathbb{R}^{d_x}, Y \subset \mathbb{R}^{d_y}$ are non-empty (compact) convex sets (with $d_x, d_y \in \mathbb{N}$); $g_X : X^{\otimes 2} \to \mathbb{R}$ is a gauge function in $L^2_{sys}(X^{\otimes 2}, \mu^{\otimes 2})$, i.e., the space of symmetric, real-valued functions on $X^{\otimes 2} := X \times X$ that are square-integrable with respect to $\mu^{\otimes 2} := \mu \otimes \mu$ (similarly for $g_Y \in L^2_{sys}(Y^{\otimes 2}, \nu^{\otimes 2})$); and $\mu, \nu$ are probability measures defined on $X, Y$ respectively. In addition, assume that $\mathbb{X}$ and $\mathbb{Y}$ are metric measure spaces (mm-spaces); that is, there exist distances $d_X$ and $d_Y$ on $X$ and $Y$, respectively. In general, we will assume that $g_X$ is a continuous function on $X^{\otimes 2}$ with respect to the metric $d_{X^{\otimes 2}}((x_1, x_1'), (x_2, x_2')) := d_X(x_1, x_2) + d_X(x_1', x_2')$ on $X^{\otimes 2}$ (respectively, for $g_Y$). Thus, since $X$ is compact, $g_X$ (respectively, $g_Y$) is a Lipschitz continuous function, i.e., there exists $K \in \mathbb{R}_{>0}$ such that

$$|g_X(x_1, x_1') - g_X(x_2, x_2')| \leq K(d_X(x_1, x_2) + d_X(x_1', x_2')), \quad \forall x_1, x_1', x_2, x_2' \in X. \tag{3}$$

For the reader's convenience, unless stated otherwise, we default to $g_X(x_1, x_2) = d_X^2(x_1, x_2)$ (which is the classical setting for the Gromov-Wasserstein problem (Alvarez-Melis & Jaakkola, 2018; Vayer, 2020)), or to the inner product $g_X(x_1, x_2) = x_1^\top x_2$ (which has been studied in (Vayer, 2020; Sturm, 2012)). The same applies to $g_Y$.

The **Gromov-Wasserstein (GW) problem** between $\mathbb{X}$ and $\mathbb{Y}$ is defined as

$$GW^2(\mathbb{X}, \mathbb{Y}) := \min_{\gamma \in \Gamma(\mu, \nu)} \langle |g_X - g_Y|^2, \gamma^{\otimes 2} \rangle \tag{4}$$

where

$$\langle |g_X - g_Y|^2, \gamma^{\otimes 2} \rangle := \int_{(X \times Y)^{\otimes 2}} |g_X(x, x') - g_Y(y, y')|^2 \, d\gamma(x, y) d\gamma(x', y').$$

A minimizer for (4) always exists (Mémoli, 2011), and we use $\Gamma^*(\mathbb{X}, \mathbb{Y})$ to denote the set of all optimal transportation plans $\gamma \in \Gamma(\mu, \nu)$ for the GW problem defined in (4). In addition, when $g_X = d_X^s$ and $g_Y = d_Y^s$ for $s \geq 1$, $GW(\cdot, \cdot)$ defines a metric on the space of mm-spaces up to isomorphism[1].

Similar to classical OT theory, we say that the **GW-Monge mapping assumption** holds if there exists a minimizer $\gamma$ for (4) of the form $\gamma = (\text{id} \times T)_{\#}\mu$, where $\text{id} : X \to X$ is the identity map on $X$, and $T : X \to Y$ is a measurable function that pushes forward $\mu$ to $\nu$. In recent years, the existence of Monge mappings for the GW problem has been studied in works such as (Vayer, 2020; Maron & Lipman, 2018; Dumont et al., 2024).

Note that the GW minimization problem defined in (4) is quadratic but non-convex. The computational cost of $GW(\cdot, \cdot)$ for discrete measures $\mu, \nu$ of size $n$ is $\mathcal{O}(n^3 \cdot L)$ to obtain a solution that is a local minimum, where $n^3$ is the computational cost of one iteration and $L$ is the required number of iterations[2]. Thus, when faced with the task of computing the pairwise GW distances between discrete gm-spaces $\mathbb{Y}^1, \ldots, \mathbb{Y}^K$ with $n$ points each, the resulting complexity is $\mathcal{O}(K^2 \cdot n^3 \cdot L)$. To reduce the complexity of computing the pairwise distances in this setting, (Beier et al., 2022) introduces a linear embedding technique called **linear Gromov-Wasserstein (LGW)**, which we discuss in the following section.

## 2.3 THE LINEAR GROMOV-WASSERSTEIN EMBEDDING AND DISTANCE

Consider a fixed gm-space $\mathbb{X} = (X, g_X, \mu)$, which serves as a **reference** space. Given a **target** gm-space $\mathbb{Y} = (Y, g_Y, \nu)$, suppose that the GW-Monge mapping assumption holds, i.e., there exists an optimal transportation plan $\gamma = (\text{id} \times T)_{\#}\mu$ for the GW problem

---

[1]Two mm-spaces $\mathbb{X}, \mathbb{Y}$ are isomorphic is there exists an isometry $\psi : X \to Y$, i.e., $d_X(x, x') = d_Y(\psi(x), \psi(x'))$, which is surjective and such that $\nu = \psi_{\#}\mu$.

[2]We refer to Appendix D for details.

(4) between $\mathbb{X}$ and $\mathbb{Y}$. Then, the **linear Gromov-Wasserstein (LGW) embedding** is defined as the following mapping

$$\mathbb{Y} \mapsto k_\gamma := g_X(\cdot_1, \cdot_2) - g_Y(T(\cdot_1), T(\cdot_2)), \tag{5}$$

which assigns each gm-space $\mathbb{Y}$ to a gauge function $k_\gamma : X \times X \to \mathbb{R}$ defined by (5). This embedding can recover the GW distance $GW(\mathbb{X}, \mathbb{Y})$ by the following:

$$GW(\mathbb{X}, \mathbb{Y}) = \int_{X \times X} |k_\gamma(x, x')|^2 d\mu(x) d\mu(x').$$

Given two gm-spaces $\mathbb{Y}^1 = \{Y^1, g_{Y^1}, \nu^1\}, \mathbb{Y}^2 = \{Y^2, g_{Y^2}, \nu^2\}$, assume that there exist optimal transportation plans $\gamma^1 = (\mathrm{id} \times T^1)_\# \mu \in \Gamma^*(\mathbb{X}, \mathbb{Y}^1), \gamma^2 = (\mathrm{id} \times T^2)_\# \mu \in \Gamma^*(\mathbb{X}, \mathbb{Y}^2)$. Let $k_{\gamma^1}, k_{\gamma^2}$ denote the corresponding linear GW embeddings as defined by (5). Then, the **LGW distance conditioned on** $\gamma^1, \gamma^2$, is defined as

$$LGW(\mathbb{Y}^1, \mathbb{Y}^2; \mathbb{X}, \gamma^1, \gamma^2) := \|k_{\gamma^1} - k_{\gamma^2}\|_{L^2(\mu^{\otimes 2})}. \tag{6}$$

When the GW-Monge mapping assumption does not hold, (Beier et al., 2022) extend the above distance conditioned on a pair of arbitrary plans $\gamma^1 \in \Gamma^*(\mathbb{X}, \mathbb{Y}^1), \gamma^2 \in \Gamma^*(\mathbb{X}, \mathbb{Y}^2)$:

$$LGW(\mathbb{Y}^1, \mathbb{Y}^2; \mathbb{X}, \gamma^1, \gamma^2) = \inf_{\gamma \in \Gamma(\gamma^1, \gamma^2; \mu)} \langle |g_{Y^1} - g_{Y^2}|^2, \gamma^{\otimes 2} \rangle \tag{7}$$

$$= \inf_{\gamma \in \Gamma(\gamma^1, \gamma^2; \mu)} \int_{(X \times Y^1 \times Y^2)^{\otimes 2}} |g_{Y^1}(y^1, y^{1\prime}) - g_{Y^2}(y^2, y^{2\prime})|^2 d\gamma(x, y^1, y^2)\gamma(x', y^{1\prime}, y^{2\prime}),$$

where

$$\Gamma(\gamma^1, \gamma^2; \mu) := \{\gamma \in \mathcal{M}_+(X \times Y^1 \times Y^2) : (\pi_{X,Y^1})_\# \gamma = \gamma^1, (\pi_{X,Y^2})_\# \gamma = \gamma^2, (\pi_X)_\# \gamma = \mu\},$$

for $\pi_{X,Y^i}(x, y^1, y^2) := (x, y^i)$ for $i = 1, 2$, and $\pi_X(x, y^1, y^2) := x$. Finally, the **LGW distance** between $\mathbb{Y}^1$ and $\mathbb{Y}^2$ is defined as

$$LGW(\mathbb{Y}^1, \mathbb{Y}^2; \mathbb{X}) = \inf_{\substack{\gamma^1 \in \Gamma^*(\mu, \nu^1) \\ \gamma^2 \in \Gamma^*(\mu, \nu^2)}} LGW(\mathbb{Y}^1, \mathbb{Y}^2; \mathbb{X}, \gamma^1, \gamma^2). \tag{8}$$

We refer to Appendix B for further details.

To simplify the computation of the above formulation, (Beier et al., 2022) propose the **barycentric projection** method, which we will briefly review here.

Let $\gamma \in \Gamma(\mu, \nu)$ be a transportation plan between $\mu \in \mathcal{P}(X)$ and $\nu \in \mathcal{P}(Y)$. By using [Definition 5.4.2, Ambrosio et al. (2005)], the **barycentric projection map** $\mathcal{T}_\gamma : X \to Y$ is defined as

$$\mathcal{T}_\gamma(x) := \int_Y y \, d\gamma_{Y|X}(y|x). \tag{9}$$

where $\{\gamma_{Y|X}(\cdot|x)\}_{x \in X}$ is the corresponding **disintegration** of $\gamma$ with respect to its first marginal $\mu$. That is, $\{\gamma_{Y|X}(\cdot|x)\}_{x \in X}$ is a family of measures satisfying

$$\int_{X \times Y} \phi(x, y) \, d\gamma(x, y) = \int_X \int_Y \phi(x, y) \, d\gamma_{Y|X}(y|x) d\mu(x), \quad \forall \phi \in C(X \times Y).$$

If $\gamma^* \in \Gamma^*(\mathbb{X}, \mathbb{Y})$, we define the measure $\widetilde{\nu}_{\gamma^*} := (\mathcal{T}_{\gamma^*})_\# \mu \in \mathcal{P}(Y)$ and the gm-space $\widetilde{\mathbb{Y}}_{\gamma^*} := (Y, g_Y, \nu_{\gamma^*})$. We provide the following results as a complement to the work in (Beier et al., 2022).

**Proposition 2.1.** *Let $\gamma^* \in \Gamma^*(\mathbb{X}, \mathbb{Y})$ and $\gamma^0 \in \Gamma^*(\mathbb{X}, \mathbb{X})$. Then, we have the following:*

*(1) The LGW embedding of $\mathbb{Y}$ can recover the GW distance, i.e.,*

$$LGW(\mathbb{X}, \mathbb{Y}; \mathbb{X}, \gamma^0, \gamma^*) = GW(\mathbb{X}, \mathbb{Y}).$$

(2) If $T$ is a Monge mapping for $GW(\mathbb{X}, \mathbb{Y})$ such that $\gamma^* = (\mathrm{id} \times T)_{\#}\mu$, then $\mathcal{T}_{\gamma^*} = T$, $\mu$-a.e. As a consequence, $\widetilde{\nu}_{\gamma^*} = \nu$, and thus $\widetilde{\mathbb{Y}}_{\gamma^*} = \mathbb{Y}$.

(3) If $g_Y(\cdot_1, \cdot_2) = \alpha\langle\cdot_1, \cdot_2\rangle_Y$, where $\langle\cdot_1, \cdot_2\rangle_Y$ is an inner product restricted to $Y$ and $\alpha \in \mathbb{R}$, then the barycentric projection map $\mathcal{T}_{\gamma^*}$ defined by (9) is a Monge mapping in the sense that $\widetilde{\gamma} = (\mathrm{id} \times \mathcal{T}_{\gamma^*})_{\#}\mu$ is optimal for the GW problem $GW(\mathbb{X}, \widetilde{\mathbb{Y}}_{\gamma^*})$.

We refer the reader to Appendix E for the proof.

Based on the above proposition, given $\mathbb{Y}^1 = (Y^1, g_{Y^1}, \nu^1)$ and $\mathbb{Y}^2 = (Y^2, g_{Y^2}, \nu^2)$, let $\gamma^1 \in \Gamma^*(\mathbb{X}, \mathbb{Y}^1)$ and $\gamma^2 \in \Gamma^*(\mathbb{X}, \mathbb{Y}^2)$. The following **approximated LGW (aLGW) distance** can be used as a proxy for the LGW and GW distances. Furthermore, in practice, we can fix a single pair $(\gamma^1, \gamma^2)$ to compute the distance rather than compute the infimum:

$$aLGW(\mathbb{Y}^1, \mathbb{Y}^2; \mathbb{X}) := \inf_{\substack{\gamma^1 \in \Gamma^*(\mathbb{X}, \mathbb{Y}^1) \\ \gamma^2 \in \Gamma^*(\mathbb{X}, \mathbb{Y}^2)}} LGW(\mathbb{Y}^1_{\gamma^1}, \mathbb{Y}^2_{\gamma^2}; \mathbb{X}, \gamma^1, \gamma^2)$$

$$= \inf_{\substack{\gamma^1 \in \Gamma^*(\mathbb{X}, \mathbb{Y}^1) \\ \gamma^2 \in \Gamma^*(\mathbb{X}, \mathbb{Y}^2)}} \|g_{Y^1}(\mathcal{T}_{\gamma^1}(\cdot_1), \mathcal{T}_{\gamma^1}(\cdot_2)) - g_{Y^2}(\mathcal{T}_{\gamma^2}(\cdot_1), \mathcal{T}_{\gamma^2}(\cdot_2))\|^2_{L^2(\mu^{\otimes 2})} \quad (10)$$

$$\approx \|g_{Y^1}(\mathcal{T}_{\gamma^1}(\cdot_1), \mathcal{T}_{\gamma^1}(\cdot_2)) - g_{Y^2}(\mathcal{T}_{\gamma^2}(\cdot_1), \mathcal{T}_{\gamma^2}(\cdot_2))\|^2_{L^2(\mu^{\otimes 2})}. \quad (11)$$

## 2.4 The Partial Gromow-Wasserstein Distance

**Partial Gromov-Wasserstein (PGW)**, or more generally, unbalanced GW, relaxes the assumption that $\mu, \nu$ are normalized probability measures (Chapel et al., 2020; Séjourné et al., 2021; Bai et al., 2024). In particular, the PGW distance is defined as

$$PGW^2_\lambda(\mathbb{X}, \mathbb{Y}) := \inf_{\gamma \in \Gamma_{\leq}(\mu, \nu)} \langle|g_X - g_Y|^2, \gamma^{\otimes 2}\rangle + \lambda(|\mu^{\otimes 2} - \gamma_X^{\otimes 2}| + |\nu^{\otimes 2} - \gamma_Y^{\otimes 2}|), \quad (12)$$

where $\lambda > 0$ is a parameter; $\gamma_X := (\pi_X)_{\#}\gamma, \gamma_Y := (\pi_Y)_{\#}\gamma$ are the corresponding marginals of $\gamma$ on $X$ and $Y$, respectively; and $\Gamma_{\leq}(\mu, \nu) := \{\gamma \in \mathcal{M}_+(X \times Y) : \gamma_X \leq \mu, \gamma_Y \leq \nu\}$, where $\mathcal{M}_+(X \times Y)$ denotes the set of finite non-negative measures on $X \times Y$, and $\gamma_X \leq \mu$ denotes that for any Borel set $A$, $\gamma_X(A) \leq \mu(A)$ (similarly for $\gamma_Y \leq \nu$).

In the discrete setting, the PGW problem can be solved by variants of the Frank-Wolfe algorithm (see the algorithms in (Bai et al., 2024) for details). From a theoretical perspective, there always exists a minimizer for the optimization problem in (12) (see [Proposition 3.3, (Bai et al., 2024)]). We use the notation $\Gamma^*_{\leq,\lambda}(\mathbb{X}, \mathbb{Y})$ to denote the set of all optimal transportation plans for $PGW_\lambda(\mathbb{X}, \mathbb{Y})$. For convenience, when $\lambda$ is fixed, we may omit the subscript $\lambda$ and write $\Gamma^*_{\leq}(\mathbb{X}, \mathbb{Y})$ instead.

We say that the **PGW-Monge mapping assumption** holds if there exists an optimal transportation plan $\gamma$ for the optimization problem in (12) of the form $\gamma = (\mathrm{id} \times T)_{\#}\gamma_X$ for some $T : X \to Y$.

## 3 Linear Partial Gromov-Wasserstein Embedding and Distance

Inspired by the work in (Beier et al., 2022), we extend the LGW distance to the unbalanced setting. In particular, we present a linearization technique for the PGW problem proposed in (Bai et al., 2024).

Given gm-spaces $\mathbb{X} = (X, g_X, \mu)$ and $\mathbb{Y} = (Y, g_Y, \nu)$, let $\gamma \in \Gamma^*_{\leq,\lambda}(\mathbb{X}, \mathbb{Y})$. We first suppose that the *PGW-Monge mapping assumption* holds; that is, $\gamma = (\mathrm{id} \times T)_{\#}\gamma_X$ for some $T : X \to Y$. We subsequently define the **linear partial Gromov-Wasserstein (LPGW) embedding** as

$$\mathbb{Y} \mapsto (\gamma_X, k_\gamma, \gamma_c), \quad (13)$$

where $\gamma_X := (\pi_X)_{\#}\gamma$, $k_\gamma(\cdot_1, \cdot_2) := g_X(\cdot_1, \cdot_2) - g_Y(T(\cdot_1), T(\cdot_2))$, and $\gamma_c := \nu^{\otimes 2} - (\gamma_Y)^{\otimes 2}$ for $\gamma_Y = (\pi_Y)_{\#}\gamma$. Here, the first component of the embedding, $\gamma_X$, encodes the mass transportation from the source domain $X$. The second component, $k_\gamma$, describes the transportation cost function. Finally, the third component, $\gamma_c$, encodes the mass creation in the target domain $Y$.

Similarly to the LGW embedding given by (5), we show in Proposition 3.1 that the above embedding can recover $PGW_\lambda(\mathbb{X}, \mathbb{Y})$ in the sense that

$$\|k_{\gamma^*}\|^2_{L^2((\gamma_X^*)^{\otimes 2})} + \lambda(|\mu|^2 - |\gamma_X^*|^2 + |\gamma_c^*|) = PGW_\lambda(\mathbb{X}, \mathbb{Y}).$$

Given two gm-spaces $\mathbb{Y}^1, \mathbb{Y}^2$, let $\gamma^1 \in \Gamma^*_{\leq, \lambda}(\mathbb{X}, \mathbb{Y}^1)$ and $\gamma^2 \in \Gamma^*_{\leq, \lambda}(\mathbb{X}, \mathbb{Y}^2)$. Suppose $\gamma^1, \gamma^2$ are induced by Monge mappings $T^1, T^2$ and compute the corresponding LPGW embeddings via (13). We define the **LPGW discrepancy conditioned on** $\gamma^1, \gamma^2$ by

$$LPGW_\lambda(\mathbb{Y}^1, \mathbb{Y}^2; \mathbb{X}, \gamma^1, \gamma^2) := \inf_{\mu' \leq \gamma_X^1 \wedge \gamma_X^2} \|k_{\gamma^1} - k_{\gamma^2}\|^2_{L^2(\mu'^{\otimes 2})} + \lambda(|\nu^1|^2 + |\nu^2|^2 - 2|\mu'|^2), \quad (14)$$

where $(\gamma_X^1 \wedge \gamma_X^2)(E) = \min_{F \subset E}\{\gamma_X^1(E) + \gamma_X^2(E \setminus F)\}$. Note that the above formulation can be written entirely in terms of the LPGW embeddings since $|\nu^1|^2 = |\gamma_c^1| + |\gamma_X^1|^2$, and similarly $|\nu^2|^2 = |\gamma_c^2| + |\gamma_X^2|^2$.

When the PGW-Monge mapping assumption does not hold, we extend the above LPGW discrepancy conditioned on arbitrary transportation plans $\gamma^1 \in \Gamma^*_{\leq, \lambda}(\mathbb{X}, \mathbb{Y}^1), \gamma^2 \in \Gamma^*_{\leq, \lambda}(\mathbb{X}, \mathbb{Y}^2)$ as follows:

$$LPGW_\lambda(\mathbb{Y}^1, \mathbb{Y}^2; \mathbb{X}, \gamma^1, \gamma^2) := \inf_{\gamma \in \Gamma_{\leq}(\gamma^1, \gamma^2; \mu)} \|g_{Y^1} - g_{Y^2}\|^2_{L^2(\gamma^{\otimes 2})} + \lambda(|\nu^1|^2 + |\nu^2|^2 - 2|\gamma|^2), \quad (15)$$

where

$$\Gamma_{\leq}(\gamma^1, \gamma^2; \mu) := \{\gamma \in \mathcal{M}_+(X \times Y^1 \times Y^2) : (\pi_{X,Y^1})_\# \gamma \leq \gamma^1, (\pi_{X,Y^2})_\# \gamma \leq \gamma^2\}.$$

Finally, to gain independence from the transportation plans $\gamma^1, \gamma^2$, we formally define the **LPGW discrepancy** as

$$LPGW_\lambda(\mathbb{Y}^1, \mathbb{Y}^2; \mathbb{X}) := \inf_{\substack{\gamma^1 \in \Gamma^*_{\leq, \lambda}(\mathbb{X}, \mathbb{Y}^1) \\ \gamma^2 \in \Gamma^*_{\leq, \lambda}(\mathbb{X}, \mathbb{Y}^2)}} LPGW_\lambda(\mathbb{Y}^1, \mathbb{Y}^2; \mathbb{X}, \gamma^1, \gamma^2). \quad (16)$$

**Proposition 3.1.** *Given compact gm-spaces* $\mathbb{X} = (X, g_X, \mu)$ *and* $\mathbb{Y} = (Y, g_Y, \nu)$, *let* $\gamma^0 \in \Gamma^*_{\leq, \lambda}(\mathbb{X}, \mathbb{X})$ *and* $\gamma^* \in \Gamma^*_{\leq, \lambda}(\mathbb{X}, \mathbb{Y})$. *We have the following:*

*(1) Problem (15) admits a minimizer.*

*(2) Under the PGW-Monge mapping assumption, the problems (14) and (15) coincide.*

*(3) Under the PGW-Monge mapping assumption, when*

$$2\lambda \geq \max_{\substack{y^1, y^{1\prime} \in Y^1, \\ y^2, y^{2\prime} \in Y^2}} |g_Y(y^1, y^{1\prime}) - g_Y(y^2, y^{2\prime})|^2 \quad (17)$$

*problem (14) achieves the value*

$$LPGW_\lambda(\mathbb{Y}^1, \mathbb{Y}^2; \mathbb{X}, \gamma^1, \gamma^2) := \|k_{\gamma^1} - k_{\gamma^2}\|^2_{L((\gamma_X^1 \wedge \gamma_X^2)^{\otimes 2})} + \lambda(|\nu^1|^2 + |\nu^2|^2 - 2|\gamma_X^1 \wedge \gamma_X^2|^2). \quad (18)$$

*(4) The LPGW embedding of* $\mathbb{Y}$ *can recover* $PGW_\lambda(\mathbb{X}, \mathbb{Y})$ *in the sense that, under the PGW-Monge mapping assumption, we have*

$$\|k_{\gamma^*}\|^2_{L^2((\gamma_X^*)^{\otimes 2})} + \lambda(|\mu|^2 - |\gamma_X^*|^2 + |\gamma_c^*|) = PGW_\lambda(\mathbb{X}, \mathbb{Y}),$$

*and in general, we have*

$$LPGW_\lambda(\mathbb{X}, \mathbb{Y}; \gamma^0, \gamma^*) = PGW_\lambda(\mathbb{X}, \mathbb{Y}).$$

*(5) The LPGW discrepancies defined in (15) and (16) are pseudo-metrics. Indeed, under certain conditions, (16) is a rigorous metric. We refer to Appendix F for detailed discussion.*

The proof of the above proposition is included in Appendix F.

Similar to LGW, to accelerate the computation of the above formulation, we propose the following **barycentric projection** method in the LPGW setting.

First, similar to Proposition 2.1, we present the following theorem, for which we provide the proof in Appendix H.

**Theorem 3.2.** *Given gm-spaces* $\mathbb{X} = (X, g_X, \mu)$ *and* $\mathbb{Y} = (Y, g_Y, \nu)$, *let* $\gamma^0 \in \Gamma^*_{\leq, \lambda}(\mathbb{X}, \mathbb{X})$ *and* $\gamma^* \in \mathcal{M}_+(X \times Y)$. *Further, let* $\widetilde{\nu} := \widetilde{\nu}_{\gamma^*} := (\mathcal{T}_{\gamma^*})_\# \gamma$ *and* $\widehat{\mathbb{Y}}_{\gamma^*} = (Y, d_Y, \widetilde{\nu})$. *Then,*

*(1) If* $\gamma^*$ *is induced by a mapping* $T$, *then* $\mathcal{T}_{\gamma^*} = T, \gamma^*_X - a.s.$

*(2) If* $\gamma^* \in \Gamma^*_\leq(\mathbb{X}, \mathbb{Y})$, *and the conditions for* $g_Y(\cdot_1, \cdot_2)$ *in statement (3) of Proposition 2.1 hold, then* $\widetilde{\gamma} := (\mathrm{id} \times \mathcal{T}_{\gamma^*})_\# \gamma^*_X$ *is optimal for* $PGW_\lambda(\mathbb{X}, \widetilde{\mathbb{Y}}_{\gamma^*})$.

*(3) If* $\gamma^1 \in \Gamma^*_\leq(\mathbb{X}, \mathbb{Y}^1)$ *and* $\gamma^2 \in \Gamma^*_\leq(\mathbb{X}, \mathbb{Y}^2)$, *and the PGW-Monge mapping assumption holds, then*

$$LPGW_\lambda(\mathbb{Y}^1, \mathbb{Y}^2; \mathbb{X}, \gamma^1, \gamma^2) = LPGW_\lambda(\widetilde{\mathbb{Y}}_{\gamma^1}, \widetilde{\mathbb{Y}}_{\gamma^2}; \mathbb{X}, \widetilde{\gamma}^1, \widetilde{\gamma}^2) + \lambda(|\gamma^1_c| + |\gamma^2_c|). \quad (19)$$

Based on part (3) in the above theorem, we define the **approximated LPGW (aLPGW) discrepancy** by

$$aLPGW_\lambda(\mathbb{Y}^1, \mathbb{Y}^2; \mathbb{X}) := \inf_{\substack{\gamma^1 \in \Gamma^*_{\leq, \lambda}(\mathbb{X}, \mathbb{Y}^1), \\ \gamma^2 \in \Gamma^*_{\leq, \lambda}(\mathbb{X}, \mathbb{Y}^2)}} LPGW_\lambda(\widetilde{\mathbb{Y}}_{\gamma^1}, \widetilde{\mathbb{Y}}_{\gamma^2}; \mathbb{X}, \gamma^1, \gamma^2) + \lambda(|\gamma^1_c| + |\gamma^2_c|). \quad (20)$$

**Proposition 3.3.** *If* $X, Y$ *are compact sets, when* $\lambda$ *is sufficiently large, in particular, (17) is satisfied, the above formulation becomes:*

$$aLPGW_\lambda(\mathbb{Y}^1, \mathbb{Y}^2; \mathbb{X}) = \inf_{\substack{\gamma^1 \in \Gamma^*_{\leq, \lambda}(\mathbb{X}, \mathbb{Y}^1), \\ \gamma^2 \in \Gamma^*_{\leq, \lambda}(\mathbb{X}, \mathbb{Y}^2)}} \|k_{\gamma^1} - k_{\gamma^2}\|^2_{(\gamma^1_X \wedge \gamma^2_X)^{\otimes 2}} + \lambda(|\nu^1|^2 + |\nu^2|^2 - 2|\gamma^1_X \wedge \gamma^2_X|^2). \quad (21)$$

In practice, similar to LGW, we only select one pair of optimal transportation plans $\gamma^1, \gamma^2$ to compute the embedding and the above *aLPGW* discrepancy. In addition, we apply (21) to approximate the original formula (20). That is, in our experiments, we directly compute

$$\|k_{\gamma^1} - k_{\gamma^2}\|^2_{(\gamma^1_X \wedge \gamma^2_X)^{\otimes 2}} + \lambda(|\nu^1|^2 + |\nu^2|^2 - 2|\gamma^1_X \wedge \gamma^2_X|^2). \quad (22)$$

### 3.1 NUMERICAL IMPLEMENTATION OF LPGW IN DISCRETE SETTING

In practice, we consider discrete distributions (or, more generally, discrete Radon measures). Suppose $\mathbb{X} = \{X, \|\cdot\|_{d_0}, \mu = \sum_{i=1}^n p_i \delta_{x_i}\}$, where $X \subset \mathbb{R}^{d_0}$ is a convex compact set containing $\{0_{d_0}, x_1, \ldots x_n\}$. We similarly let $\mathbb{Y}^1 = (Y^1, \|\cdot\|_{d_1}, \nu = \sum_{j=1}^{m_1} q^1_j \delta_{y^1_j})$ and $\mathbb{Y}^2 = (Y^2, \|\cdot\|_{d^2}, \nu^2 = \sum_{j=1}^{m_2} q^2_j \delta_{y^2_j})$.

Suppose $\gamma^1 \in \mathbb{R}^{n \times m_1}_+$ is an optimal transportation plan for $PGW_\lambda(\mathbb{X}, \mathbb{Y}^1)$. By convexity of $Y$, the barycentric projection given by (9) becomes

$$\mathcal{T}_{\gamma^1}(x_i) = \begin{cases} \widetilde{y}_i = \frac{1}{\widetilde{q}_i}[\gamma^1 y^1]_i & \text{if } \widetilde{q}_i > 0 \\ 0_{d_1} & \text{if } \widetilde{q}_i = 0 \end{cases} \quad \text{where } \widetilde{q} = \gamma^1_X = \gamma^1 1_{m_1}. \quad (23)$$

Thus, the corresponding projected measure becomes $\widetilde{\nu}_{\gamma^1} = \sum_i \widetilde{q}_i \delta_{\widetilde{y}_i}$. In addition, we have $|\gamma_c| = (\sum_{i=1}^n p_i)^2 - (\sum_{i \in 1}^n \widetilde{q}^1_i)^2$. Let

$$\widetilde{K}^1 := k_{\widetilde{\gamma}^*} = [\|\widetilde{y}_i - \widetilde{y}'_i\|^2 - \|x_i - x'_i\|^2]_{i, i' \in [1:n]} \in \mathbb{R}^{n \times n}. \quad (24)$$

Suppose $\gamma^2 \in \mathbb{R}^{n \times m_2}_+$ is an optimal transportation plan for $PGW_\lambda(\mathbb{X}, \mathbb{Y}^2)$. We define all corresponding terms analogously.

Let $\widetilde{q}^{1,2} = \widetilde{q}^1 \wedge \widetilde{q}^2, |\widetilde{q}^1 - \widetilde{q}^2|_{TV} = \sum_{i=1}^{n} |\widetilde{q}_i^1 - \widetilde{q}_i^2|, |q^1| = \sum_{i=1}^{m_1} q_i^1$. The aLPGW distance given by (20) and (21) becomes

$$aLPGW(\mathbb{Y}^1, \mathbb{Y}^2; \mathbb{X})$$

$$\approx \sum_{i=1}^{n} \left( \|\widetilde{y}_i^1 - \widetilde{y}_{i'}^1\|^2 - \|\widetilde{Y}_i^2 - \widetilde{Y}_{i'}^2\|^2 \right)^2 \widetilde{q}_i^{12} \widetilde{q}_{i'}^{12} + \lambda(|q^1|^2 + |q^2|^2 - 2|\widetilde{q}^{12}|^2)$$

$$= (\widetilde{q}^{12})^\top \left[ (\widetilde{K}_1 - \widetilde{K}_2)^2 \right] (\widetilde{q}^{12}) + \lambda(|\gamma_c^1| + |\gamma_c^2| + |\widetilde{q}_1^{\otimes 2} - \widetilde{q}_2^{\otimes 2}|_{TV}), \qquad (25)$$

where $(\widetilde{K}_1 - \widetilde{K}_2)^2$ denotes the element-wise squared matrix and $\widetilde{q}_1^{\otimes 2} = \widetilde{q}_1 \widetilde{q}_1^\top$ (similarly for $\widetilde{q}_2^{\otimes 2}$). The above quantity (25) can be used to approximate the original PGW distance between $\mathbb{Y}^1$ and $\mathbb{Y}^2$.

In addition, the original LPGW embedding (13) of $\mathbb{Y}^1$ (in fact, of $\widetilde{\mathbb{Y}}^1$) is $(\widetilde{K}^1, \widetilde{\nu}_{\gamma^1}, \gamma_c^1)$. We reduce this embedding to $(\widetilde{K}^1, \widetilde{q}^1, |\gamma_c^1|)$ which is sufficient to compute the above aLPGW discrepancy. Thus, $(\widetilde{K}^1, \widetilde{q}^1, |\gamma_c^1|)$ can be regarded as numerical implementation of the LPGW embedding (13).

## 4 EXPERIMENTS

### 4.1 ELLIPTICAL DISKS

In this experiment, we apply the LPGW distance and the PGW distance to the 2D dataset presented in (Beier et al., 2022), consisting of 100 elliptical disks. We first compute the pairwise distances between the samples using the PGW distance and then compare them against the LPGW distance using nine different reference spaces. We present the resulting wall-clock time for each method and evaluate the quality of the approximation of PGW by LPGW using the mean relative error (MRE) and the Pearson correlation coefficient (PCC).

**Experiment setup.** We represent each 2D shape as an mm-space $\mathbb{X}^i = (\mathbb{R}^2, \| \cdot \|_2, \mu^i)$, where $\mu^i = \sum_{i=1}^{n} \frac{1}{n} \delta_{x_i}$. We normalize each shape so that the largest pairwise distance in each mm-space is 1. Based on [Lemma E.2, Bai et al. (2023)], the largest possible choice of $\lambda$ is given by $2\lambda = 1$. We hence test $\lambda \in \{0.05, 0.08, 0.1, 0.3, 0.5\}$, and for each reference space, we compute the pairwise LPGW distances and compute the wall-clock time, MRE, and PCC. We refer to Appendix K for full numerical details, MDS visualizations, and complete results.

**Performance analysis.** We present the results when $\lambda = 0.1$ in Table 1. First, we observe that LPGW is significantly faster than PGW as it only requires $N = 100$ transport plan computations, whereas the PGW methods require $\binom{N}{2}$ transport plan computations. Second, we observe that when the reference space is one of $\mathbb{S}_6, \mathbb{S}_7, \mathbb{S}_8, \mathbb{S}_9$, LPGW admits a relatively smaller MRE and higher PCC, demonstrating the importance of the chosen reference space.

|  | **PGW** | $\mathbb{S}_1$ | $\mathbb{S}_2$ | $\mathbb{S}_3$ | $\mathbb{S}_4$ | $\mathbb{S}_5$ | $\mathbb{S}_6$ | $\mathbb{S}_7$ | $\mathbb{S}_8$ | $\mathbb{S}_9$ |
|---|---|---|---|---|---|---|---|---|---|---|
|  |  |  |  |  |  |  |  |  |  |  |
| **points** |  | 441 | 676 | 625 | 52 | 289 | 545 | 882 | 882 | 317 |
| **time** (min) | 46.97 | 0.76 | 3.78 | 3.13 | 0.08 | 0.62 | 1.56 | 1.91 | 1.98 | 0.71 |
| **MRE** ↓ | — | 0.1941 | 0.1264 | 0.1431 | 0.2542 | 0.0538 | 0.0444 | **0.0205** | **0.0198** | 0.0245 |
| **PCC** ↑ | — | 0.5781 | 0.5738 | 0.5881 | 0.8581 | 0.9871 | 0.9930 | **0.9952** | **0.9954** | 0.9949 |

Table 1: Experimental results when $\lambda = 0.1$. The first row shows the total wall-clock time for PGW and LPGW. The second and third rows display the MRE (mean relative error) and PCC (Pearson correlation coefficient), respectively.

### 4.2 SHAPE RETRIEVAL

**Experiment setup.** We now employ the PGW distance to distinguish between 2D and 3D shapes, similarly to as done in (Beier et al., 2022). We use GW, LGW, and PGW as baselines for comparison against the results of our proposed LPGW. Given a series of shapes, we represent the shapes as mm-spaces $\mathbb{X}^i = (\mathbb{R}^d, \| \cdot \|_2, \mu^i)$, where $\mu^i = \sum_{k=1}^{n^i} \alpha^i \delta_{x_k^i}$.

For the GW and LGW methods, we normalize the mass for the balanced mass constraint setting (i.e., $\alpha^i = \frac{1}{n^i}$), and for the PGW and LPGW methods, we let $\alpha^i = \alpha$ for all the shapes, where $\alpha > 0$ is a fixed constant. In this manner, we compute the pairwise distances between the shapes.

Next, using the approach given by (Beier et al., 2022), we combine each distance matrix with a support vector machine (SVM), applying stratified 10-fold cross-validation. In each iteration of cross-validation, we train an SVM using $\exp(-\sigma D)$ as the kernel, where $D$ is the matrix of pairwise distances (w.r.t. one of the considered distances) restricted to 9 folds, and compute the accuracy of the model on the remaining fold. We report the accuracy averaged over all 10 folds for each model.

**Dataset setup.** We test one 2D shape dataset and one 3D shape dataset. We use the 2D dataset presented in (Bai et al., 2024), consisting of 8 classes, each containing 20 shapes. The 3D dataset is given by (Pan et al., 2021), which provides 100-200 complete shapes in each of 16 different classes, and for each complete shape, provides 26 corresponding "incomplete shapes." We choose four classes of shapes from this dataset, and for each, we sample 15 complete and 45 incomplete shapes, yielding a total of 240 shapes. The datasets are visualized in Figure 4 in the Appendix.

**Performance analysis.** We refer to Appendix L for full numerical details, parameter settings, and the visualization of the resulting confusion matrices. We visualize the resulting pairwise distance matrices in Figure 7. For the SVM experiments, on the 2D dataset, GW achieves the highest accuracy, 98.1%, while the second best method is LPGW, 97.5%. On the 3D dataset, PGW achieves the highest accuracy of 93.8%, and LPGW achieves the next highest accuracy of 93.7%. We refer to Table 2 for details.

|  | GW | PGW | LGW | LPGW (ours) |
|---|---|---|---|---|
| **2D Dataset** | | | | |
| Accuracy ↑ | **98.1%** | 96.2% | 93.7% | **97.5%** |
| Time (s) ↓ | 43s | 39s | **0.4s** | **0.5s** |
| **3D Dataset** | | | | |
| Accuracy ↑ | 92.5% | **93.8%** | 92.5% | **93.7%** |
| Time (m) ↓ | 203.0m | 203.6m | **1.3m** | **1.8m** |

Table 2: Accuracy and wall-clock time comparison in shape retrieval experiment.

In addition, we report the wall-clock time required to compute all pairwise distances for each distance in Table 2. In both datasets, we observe that LGW/LPGW admit similar computational time and are much faster than GW/PGW. This difference is more obvious in the 3D dataset. We note that when $\lambda$ is sufficiently large, the performance between the LPGW method and the LGW method is the same (see Appendix L for details).

### 4.3 LEARNING WITH TRANSFORM-BASED EMBEDDINGS

| Data | Method | LOT | LGW | LPGW (ours) |
|---|---|---|---|---|
| no rotation | Accuracy ↑ | 89.0% | 82.5% | 82.5% |
| | Time ↓ | 183s+16s | 405s+77s | 309s+84s |
| $\eta = 0$ | Accuracy ↑ | 51.2% | 82.5% | 82.5% |
| | Time ↓ | 183s+15s | 405s+77s | 309s+84s |
| $\eta = 0.1$ | Accuracy ↑ | 13.4% | 17.0% | 81.8% |
| | Time ↓ | 183s+17s | 405s+88s | 309s+91s |
| $\eta = 0.3$ | Accuracy ↑ | 12.5% | 13.3% | 75.8% |
| | Time ↓ | 183s+23s | 405s+145s | 309s+123s |
| $\eta = 0.5$ | Accuracy ↑ | 12.5% | 12.5% | 72.9% |
| | Time ↓ | 183s+27s | 405s+248s | 309s+168s |

Table 3: Accuracy and wall-clock time comparison for learning with transform-based embeddings experiments. We report each time as $t_1 + t_2$, where $t_1$ is the time required to compute the training set embeddings and $t_2$ is the time required to compute the testing set embeddings.

In this experiment, we perform machine learning tasks using transform-based embeddings. Specifically, given training and testing datasets, we apply various transform-based embedding methods to both. Additionally, we assume that the testing data is randomly rotated and has been corrupted with random noise. The objective is to evaluate the robustness of

machine learning models trained using the different embedding techniques. The full numerical details for this experiment can be found in Appendix M.

**Baseline methods.** In this experiment, we consider several classical embedding methods, namely the LOT embedding (Wang et al., 2013; Kolouri et al., 2020; Nenna & Pass, 2023), LGW embedding (Beier et al., 2022), and our proposed LPGW embedding.

**Dataset setup.** We adopt the MNIST 2D point cloud dataset for this experiment. Specifically, for each digit, we sample $N_1 = 500$ point clouds per class from the training set and $N_2 = 100$ point clouds per class from the testing set. Each point cloud is represented as a gm-space in $\mathbb{R}^2$ with the measure $\mu = \sum_{i=1}^n p_i \delta_{x_i}$, where $p_i > 0, \forall i \in \{1, \ldots, n\}$ are the pixel intensities provided by the original dataset. We normalize $p_i$ such that $\sum_{i=1}^n p_i = 1$ for all sampled point clouds.

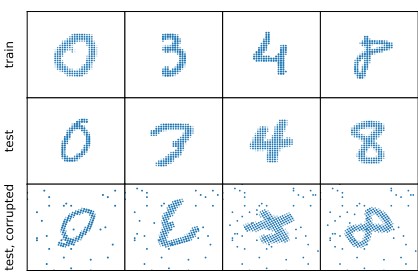

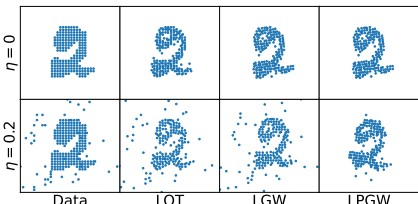

(a) MNIST point cloud dataset.

(b) Reconstructed digits based on the LOT/LGW/LPGW embeddings.

Figure 1: MNIST digits and its reconstruction.

For each shape in the training set, we randomly rotate or flip the shape horizontally and add the transformed shape to the training set. As a result, each class contains $500 \times 2 = 1000$ shapes in the training set. For each test shape, we first randomly rotate or flip the shape, and we then "corrupt" the shape by adding uniformly distributed noise. The mass of each added point is $\frac{1}{n}$, where $n$ is the number of points in the original shape, and the total mass of the added points is $\eta \in \{0, 0.1, 0.3, 0.5\}$.

**Performance analysis.** For each method (LOT, LGW, and LPGW), we first compute the embeddings of the training data and optimize a classification model (logistic regression) using these embeddings. Then, we compute the embeddings for the test dataset and evaluate the accuracy of the model on the test embeddings. The results are presented in Table 3. In this table, we observe that for the original test dataset, LOT achieves the highest accuracy at 89.0%. However, on the corrupted dataset, LOT's performance drops significantly, classifying only 51% correctly when $\eta = 0$ (no noise but with random rotation/flip) and falling dropping to 12.5% (i.e., random guessing) when $\eta > 0.1$. LGW is more robust to the data with random rotations/flips, but its accuracy also reduces to random guessing as $\eta$ is increased. In contrast, LPGW embeddings exhibit substantially greater performance, with its accuracy ranging from 70-85% when $\eta \geq 0.1$.

For an intuitive comparison of the performance of these embeddings, we present the data reconstruction results in Figure 1b and the t-SNE visualization of the embeddings produced by each method in Figure 9. Further details can be found in Appendix N.

## 5 SUMMARY

In this paper, we propose the linear partial Gromov-Wasserstein (LPGW) embedding, a linearization technique for the PGW problem. Theoretically, we prove that LPGW admits a metric with certain assumptions, and numerically, we demonstrate the utility of our proposed LPGW method in shape retrieval and learning with transform-based embedding tasks. We demonstrate that the LPGW-based method can preserve the partial matching property of PGW while significantly improving computational efficiency.

## ACKNOWLEDGMENTS

This research was partially supported by the NSF CAREER Award No. 2339898.

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

# A    BACKGROUND: LINEAR OPTIMAL TRANSPORTATION DISTANCE

The space of measures $\mathcal{P}_2(\mathbb{R}^d)$ can be endowed with the two-Wasserstein metric $W_2$, and the resulting space defines a Riemannian manifold (see, e.g., (Villani, 2021)), which is referred to in the literature as the **Wasserstein space**. Under the Monge mapping assumption, consider $\mu \in \mathcal{P}_2(\mathbb{R}^d)$ and let $\mathrm{Tan}_\mu(\mathcal{P}_2(\mathbb{R}^d))$ denote the corresponding tangent space at $\mu$. Then, each tangent vector $v \in \mathrm{Tan}_\mu(\mathcal{P}_2(\mathbb{R}^d))$ can be regarded as an $L^2(\mu)-$function, that is, $v : \mathbb{R}^d \to \mathbb{R}^d$ such that

$$\|v\|_{L^2(\mu)}^2 = \int_{\mathbb{R}^d} \|v(x)\|^2 \ d\mu(x) < \infty.$$

Given $\nu \in \mathcal{P}_2(\mathbb{R}^d)$, suppose $\gamma = (\mathrm{id} \times T)_{\#}\mu$ is an optimal transportation plan for $W_2(\mu,\nu)$. Then, the **logarithm mapping** is defined by

$$\mathcal{P}_2(\mathbb{R}^d) \ni \nu \mapsto v_\gamma := (T - \mathrm{id}) \in \mathrm{Tan}_\mu(\mathcal{P}_2(\mathbb{R}^d)), \tag{26}$$

and the resulting image $v_\gamma$ is the so-called **Linear Optimal Transportation (LOT) embedding** (Wang et al., 2013).

The LOT embedding $v_\gamma$ is a "representation" of the measure $\nu$ and encodes the optimal displacement from the fixed measure $\mu$ to $\nu$. Indeed,

$$W_2^2(\mu,\nu) = \|v_\gamma - 0\|_{L^2(\mu)}^2.$$

In general, given two probability measures $\nu^1, \nu^2 \in \mathcal{P}_2(\mathbb{R}^d)$, if $\gamma^1, \gamma^2$ are optimal solutions for $W_2(\mu,\nu^1), W_2(\mu,\nu^2)$ of the form (2), one can use the following so-called **LOT distance** to approximate the original OT distance $W_2(\nu^1,\nu^2)$ (Moosmüller & Cloninger, 2023):

$$LOT_2^2(\nu^1, \nu^2; \mu, \gamma^1, \gamma^2) := \|v_{\gamma^1} - v_{\gamma^2}\|_{L^2(\mu)}^2. \tag{27}$$

The above definition relies on the Monge mapping assumption. In (Wang et al., 2013; Moosmüller & Cloninger, 2023), the authors generalize the LOT distance without the Monge mapping assumption; however, in what follows, we will avoid such an assumption.

First, we discuss the notion of a geodesic in the classical OT setting. Suppose $\gamma$ is an optimal transportation plan for the OT problem between $\mu, \nu \in \mathcal{P}_2(\mathbb{R}^d)$. Then, the **geodesic** from $\mu$ to $\nu$ is defined by

$$t \mapsto \gamma_t := ((1-t)\pi_X + t\pi_Y)_{\#}\gamma \qquad (0 \le t \le 1). \tag{28}$$

*Remark* A.1. In the particular case where $\gamma$ is of the form $\gamma = (\mathrm{id} \times T)_{\#}\mu$, the above formula reduces to

$$t \mapsto ((1-t)\mathrm{id} + tT)_{\#}\mu \qquad (0 \le t \le 1).$$

*Remark* A.2. In the discrete case, $\mu = \sum_{i=1}^n p_i \delta_{x_i}$ and $\nu = \sum_{j=1}^m q_j \delta_{y_j}$. By abuse of notation, let $\gamma \in \Gamma^*(\mu,\nu)$ be interpreted as a matrix. Then, the above geodesic becomes

$$t \mapsto \sum_{i=1}^n \sum_{j=1}^m \delta_{(1-t)x_i + ty_j} \gamma_{ij} \qquad (0 \le t \le 1).$$

Given curves $\{\gamma_t^1\}_{t \in [0,1]}, \{\gamma_t^2\}_{t \in [0,1]}$ both originating from $\mu$, we claim they are equivalent if there exists $\epsilon > 0$ such that for all $t \in [0,\epsilon]$, $\gamma_t^1 = \gamma_t^2$. Let $G_\mu$ denote the set of all the equivalence classes. The **generalized tangent space** at $\mu$, denoted by $\mathrm{Tan}_\mu^g(\mathcal{P}_2(\mathbb{R}^d))$, is defined as the closure of $G_\mu$ (see (Beier et al., 2022) for details).

Now, given $\nu^1, \nu^2 \in \mathcal{P}_2(\mathbb{R}^d)$, let $\gamma^1, \gamma^2$ be some optimal transportation plan in $\Gamma^*(\mu,\nu^1)$, $\Gamma^*(\mu,\nu^2)$, respectively. We define the following distance, defined on the space $\mathrm{Tan}_\mu^g \mathcal{P}_2(\mathbb{R}^d)$, as the **LOT distance conditioned on** $\gamma^1, \gamma^2$:

$$LOT_2^2(\nu^1, \nu^2; \mu, \gamma^1, \gamma^2) := \inf_{\gamma \in \Gamma(\gamma^1,\gamma^2;\mu)} \int_{\mathbb{R}^{3d}} \|y^1 - y^2\|^2 d\gamma(x, y^1, y^2). \tag{29}$$

where $\Gamma(\gamma^1, \gamma^2; \mu) := \{\gamma \in \mathcal{P}(\mathbb{R}^{3d}) : (\pi_{01})_{\#}\gamma = \gamma^1, (\pi_{02})_{\#}\gamma = \gamma^2\}$, where $\pi_{01}(x, y^1, y^2) = (x, y^1)$, $\pi_{02}(x, y^1, y^2) = (x, y^2)$.

*Remark* A.3. When $\gamma^1 = (\text{id} \times T^1)_{\#}\mu, \gamma^2 = (\text{id} \times T^2)_{\#}\mu$ for some mappings $T^1, T^2$, there exists only one element in $\Gamma(\gamma^1, \gamma^2; \mu)$, which is $(\text{id} \times T^1 \times T^2)_{\#}\mu$. Thus, the right-hand side of (29) becomes

$$\|T^1(x) - T^2(x)\|_{L^2(\mu)}^2. \tag{30}$$

That is, (29) and (27) coincide.

Finally, to formulate a distance that is independent of the optimal transportation plans $\gamma^1, \gamma^2$, we define the **LOT distance** between $\nu^1, \nu^2$ with respect to $\mu$ by

$$LOT_2^2(\nu^1, \nu^2; \mu) := \inf_{\substack{\gamma^1 \in \Gamma^*(\mu, \nu^1) \\ \gamma^2 \in \Gamma^*(\mu, \nu^2)}} LOT_2^2(\nu^1, \nu^2; \mu, \gamma^1, \gamma^2). \tag{31}$$

**Barycentric projection**. The above LOT distance is complicated to compute. To simplify the computation, (Wang et al., 2013) introduces the "barycentric projection" method that we will recall here.

First, we review some basic concepts in measure theory. Given $\gamma \in \mathcal{P}(\mathbb{R}^{2d})$, with $\pi_{1\#}\gamma = \mu$, where $\pi_1 : \mathbb{R}^{2d} \to \mathbb{R}^d$ is given by $\pi_1(x, y) = x$, by the **disintegration theorem** (see, for e.g., (Ambrosio et al., 2005, Thm 5.3.1) and (Dumont et al., 2024, Theorem 6)), there exists a $\mu-$a.e. uniquely defined family of probability measures $\{\gamma(\cdot|x)\}_{x \in \mathbb{R}^d} \subset \mathcal{P}(\mathbb{R}^d)$, such that

$$\int_{\mathbb{R}^{2d}} \phi(x, y) d\gamma(x, y) = \int_{\mathbb{R}^d} \int_{\mathbb{R}^d} \phi(x, y) d\gamma(y|x) d\mu(x), \qquad \forall \phi \in C_0(\mathbb{R}^{2d}). \tag{32}$$

In this paper, we call $\gamma(\cdot|x)$ the **disintegration of $\gamma$ with respect to its first marginal ($\mu$), given $x$** (for each $x \in \text{supp}(\mu)$).

The **barycentric projection map** $\mathcal{T}_\gamma : \mathbb{R}^d \to \mathbb{R}^d$ of $\gamma$ is defined by

$$\mathcal{T}_\gamma(x) := \arg\min_{y \in Y} \int_Y \|y - y'\|^2 \, d\gamma(y'|x') = \int_Y y' d\gamma(y'|x'). \tag{33}$$

Note that the second equality holds since the distance between $y, y'$ is simply the quadratic Euclidean distance. In the third term, we have a vector integration.

Given an optimal transportation plan $\gamma \in \Gamma^*(\mu, \nu)$, we define $\widetilde{\nu}_\gamma := (\mathcal{T}_\gamma)_{\#}\mu$.

**Proposition A.4** (Prop. II.4 (Beier et al., 2022))**.** *If $\gamma \in \Gamma^*(\mu, \nu)$, then the transportation plan induced by the barycentric projection map $\mathcal{T}_\gamma$, denoted as $\widetilde{\gamma} := (\text{id} \times \mathcal{T}_\gamma)_{\#}\mu$ is optimal for the OT problem $W_2(\mu, \widetilde{\nu}_\gamma)$.*

*In addition, if $\gamma = (\text{id} \times T)_{\#}\mu$ satisfies the Monge mapping assumption, then $\mathcal{T}_{\gamma^*} = T$ $\mu$-a.e. and is a Monge map.*

Thus, if we replace $\nu$ by $\widetilde{\nu}_\gamma$, the OT problem $W_2(\mu, \widetilde{\nu}_\gamma)$ can be solved by the map $\mathcal{T}_\gamma$. Based on this property, given $\nu^1, \nu^2 \in \mathcal{P}(\mathbb{R}^d)$, consider optimal transportation plans $\gamma^1 \in \Gamma^*(\mu, \nu^1), \gamma^2 \in \Gamma(\mu, \nu^2)$, then define the following **approximated LOT distance** (aLOT) to estimate $W_2(\nu^1, \nu^2)$:

$$aLOT_2^2(\nu^1, \nu^2; \mu) := \inf_{\substack{\gamma^1 \in \Gamma^*(\mu, \nu^1) \\ \gamma^2 \in \Gamma^*(\mu, \nu^2)}} LOT_2^2(\widetilde{\nu}_{\gamma^1}, \widetilde{\nu}_{\gamma^2}; \mu, \widetilde{\gamma}^1, \widetilde{\gamma}^2)$$

$$= \inf_{\substack{\gamma^1 \in \Gamma^*(\mu, \nu^1) \\ \gamma^2 \in \Gamma^*(\mu, \nu^2)}} \|\mathcal{T}_{\gamma^1} - \mathcal{T}_{\gamma^2}\|_{L^2(\mu)}^2 \tag{34}$$

$$\approx \|\mathcal{T}_{\gamma^1} - \mathcal{T}_{\gamma^2}\|_{L^2(\mu)}^2 \tag{35}$$

Note, in most of the literature, (35) is referred to as $LOT_2$, and for simplicity, it is approximated by any pair of optimal transportation plans $(\gamma^1, \gamma^2)$.

## B    BACKGROUND: LINEAR GROMOV-WASSERSTEIN DISTANCE

Let $X \subset \mathbb{R}^{d_0}$ be a compact and convex subset, and consider a Borel measure $\mu$ on $X$. A symmetric function $g_X : X^{\otimes 2} \to \mathbb{R}$ with $\int_{X^{\otimes 2}} g_X(x, x') d\mu^{\otimes 2} < \infty$ is called a gauge function. The space of all gauge functions is denoted as $L^2_{sym}(X^{\otimes 2}, \mu^{\otimes 2})$. *In this paper, without specific description, we default set $g_X$ to be continuous (thus, Lipschitz continuous) on $X$.* For example $g_X(x, x') = \|x - x'\|^2$ or $g_X(x, x') = x^\top x'$. The space $\mathbb{X} = (X, g_X, \mu)$ is called a gauged measure space (gm-space), which can be regarded as a generalized version of an mm-space. The GW problem between two gm-spaces $\mathbb{X}, \mathbb{Y}$ is:

$$GW^2(\mathbb{X}, \mathbb{Y}) := \inf_{\gamma \in \Gamma(\mu, \nu)} \int_{(X \times X)^{\otimes 2}} |g_X(x, x') - g_Y(y, y')|^2 d\gamma^{\otimes 2}.$$

**Lemma B.1.** *There exists a minimizer $\gamma$ of $GW(\mathbb{X}, \mathbb{Y})$ in $\Gamma(\mu, \nu)$.*

The existence of minimizers of $GW(\mathbb{X}, \mathbb{Y})$ could be shown by a standard compact-SNEss argument, see for instance, Lemma 10.3 in (Mémoli, 2011). Suppose $\gamma$ is one optimal solution of the above generalized GW problem. The geodesic between $\mathbb{X}$ and $\mathbb{Y}$ is defined by

$$t \mapsto \gamma_t := (X \times Y, (1 - t)g_X + tg_Y, \gamma), t \in [0, 1]. \tag{36}$$

Note, even if $g_X, g_Y$ are metrics, $\gamma_t$ is a gm-space (rather than mm-space), and this is the reason mm-space is generalized to gm-spaces in (Beier et al., 2022) and in this paper.

Two gm-spaces $\mathbb{X}, \mathbb{Y}$ are "equivalent" if $GW^2(\mathbb{X}, \mathbb{Y}) = 0$. Let $[\mathbb{X}]$ denote the equivalence class of $\mathbb{X}$. The tangent space at $[\mathbb{X}]$ is defined by

$$\text{Tan}_{\mathbb{X}} := \left( \bigcup_{(X', g_{X'}, \mu') \in [\mathbb{X}]} L^2_{sym}(X' \times X', \mu'^{\otimes 2}) \right) / \sim \tag{37}$$

where, if $\mathbb{T}_{k_1} = (X^1, g_{X^1}, \mu^1), \mathbb{T}_{k_2} = (X^2, g_{X^2}, \mu^2) \in [\mathbb{X}]$ are **representatives** of $k_1 \in L^2_{sys}((X^1)^{\otimes 2}, (\mu^1)^{\otimes 2}), k_2 \in L^2_{sys}((X^2)^{\otimes 2}, (\mu^2)^{\otimes 2})$, we say that $k_1 \sim k_2$ if there exists $\gamma \in \Gamma(\mathbb{X}^1, \mathbb{X}^2)$ such that $GW(\mathbb{X}^1, \mathbb{X}^2) = 0$. (We refer to (Beier et al., 2022) for more details.)

Given spaces $\mathbb{X}^1 := (X \times Y^1, d_X, \gamma^1), \mathbb{X}^2 := (X \times Y^2, d_X, \gamma^2)$ where $\gamma^1, \gamma^2$ are optimal transportation plans for $GW(\mathbb{X}, \mathbb{Y}^1), GW(\mathbb{X}, \mathbb{Y}^2)$, respectively, it is straightforward to verify

$$GW(\mathbb{X}, \mathbb{X}^1) = GW(\mathbb{X}, \mathbb{X}^2) = 0,$$

thus $\mathbb{X}^1, \mathbb{X}^2 \in [\mathbb{X}]$.

In addition, choose $k_1 \in L^2_{sys}(X^1)^{\otimes 2}, (\gamma^1)^{\otimes 2}), k_2 \in L^2_{sys}((X^2)^{\otimes 2}, (\gamma^2)^{\otimes 2})$, then we have

$$k_1, k_2 \in \text{Tan}_{\mathbb{X}},$$

where $\Gamma^*(\mathbb{T}_{k_1}, \mathbb{T}_{k_2})$ is same to $\Gamma^*(\gamma^1, \gamma^2)$ in $GW(\mathbb{X}^1, \mathbb{X}^2)$.

Their inner product distance is defined by

$$D^2(k_1, k_2) := \inf_{\gamma \in \Gamma^*(\mathbb{T}_{k_1}, \mathbb{T}_{k_2})} \|k_1 - k_2\|^2_{\gamma^{\otimes 2}}.$$

Now, we set $k_1 = k_{\gamma^1} = g_{Y^1} - g_X, k_2 = k_{\gamma^2} = g_{Y^2} - g_X$. Thus, the linear GW distance between $\mathbb{Y}^1$ and $\mathbb{Y}^2$ given $\gamma^1, \gamma^2$ is defined by the above inner product distance. That is,

$$\begin{aligned} \text{LGW}(\mathbb{Y}^1, \mathbb{Y}^2; \mathbb{X}, \gamma^1, \gamma^2) &:= D(k_1, k_2) \\ &= \inf_{\gamma \in \Gamma^*(\mathbb{T}_{k_1}, \mathbb{T}_{k_2})} \|k_1 - k_2\|^2_{L^2(\gamma^{\otimes 2})} \\ &= \inf_{\gamma \in \Gamma(\gamma^1, \gamma^2; \mu)} \int_{(X \times Y^1 \times Y^2)^{\otimes 2}} \|g_{Y^1}(y^1, y^{1'}) - g_{Y^2}(y^2, y^{2'})\|^2 d\gamma^{\otimes 2} \end{aligned} \tag{38}$$

where $\Gamma(\gamma^1, \gamma^2; \mu) := \{\gamma \in \mathcal{P}(X \times Y^1 \times Y^2) : (\pi_{X,Y^1})_{\#}\gamma = \gamma^1, (\pi_{X,Y^2})_{\#}\gamma = \gamma^2\}$. The third line follows from Proposition III.1 in (Beier et al., 2022) (or equivalently from the identity (49) in the next section). Note, under Monge mapping assumptions, $\gamma^1 = (\mathrm{id} \times T^1)_{\#}\mu, \gamma^2 = (\mathrm{id} \times T^2)_{\#}\mu$, $\Gamma(\gamma^1, \gamma^2; \mu) = \{(\mathrm{id} \times T^1 \times T^2)_{\#}\mu\}$ and the above distance (38) coincides with (7).

The original Linear Gromov-Wasserstein distance between $\mathbb{Y}^1$ and $\mathbb{Y}^2$ with respect to $\mathbb{X}$ is defined as

$$\mathrm{LGW}(\mathbb{Y}^1, \mathbb{Y}^2; \mathbb{X}) := \inf_{\substack{\gamma^1 \in \Gamma(\mathbb{X}, \mathbb{Y}^1) \\ \gamma^2 \in \Gamma(\mathbb{X}, \mathbb{Y}^2)}} LGW(\mathbb{Y}^1, \mathbb{Y}^2; \mathbb{X}, \gamma^1, \gamma^2). \tag{39}$$

# C  LOT/LGW/LPGW embedding without Monge mapping.

In this section, we briefly discuss the linear transportation embedding under linear optimal transport, linear Gromov Wasserstein, and linear partial Gromov Wasserstein setting. In summary, the formulations of these embeddings do not rely on the Monge mapping assumption.

**Linear OT embedding without Monge mapping.** Given probability measures $\nu^1, \nu^2 \in \mathcal{P}(\Omega)$, and reference measure $\mu \in \mathcal{P}(\Omega)$, where $\Omega \subset \mathbb{R}^d$ is non-empty set. Choose optimal transportation plans $\gamma^1 \in \Gamma^*(\mu, \nu^1), \gamma^2 \in \Gamma^*(\mu, \nu^2)$, the linear OT embedding 26 can be generalized to

$$\nu^1 \mapsto \hat{T}^1 := \{x \mapsto \gamma^1(\cdot|x)\}, \tag{40}$$

where the conditional distribution $\gamma^1(\cdot|x)$ can be treated as a "random mapping". When $\gamma^1 = (\mathrm{id} \times T^1)_{\#}\mu$, it is straightforward to verify $T^1 = \hat{T}^1$). That is, the embedding (40),(26) coincide.

We define the "distance" between two such two random mappings $T^1, T^2$ via:

$$\|\hat{T}^1 - \hat{T}^2\|_{L(\mu)}^2 := \int \inf_{\gamma \in \Gamma(\gamma^1(\cdot|x), \gamma^2(\cdot|x))} \|y^1 - y^2\|^2 d\gamma(y^1, y^2) d\mu(x) \tag{41}$$

From (29), we have

$$LOT_2^2(\nu^1, \nu^2; \mu) := \int_{\gamma \in \Gamma(\nu^1, \nu^2; \mu)} \|y^1 - y^2\| \gamma(x, y^1, y^2)$$

$$= \int_{\Omega} \inf_{\gamma \in \Gamma(\gamma^1(\cdot|x), \gamma^2(\cdot|x)} \|y^1 - y^2\|^2 d\mu(x)$$

$$= \|\hat{T}^1 - \hat{T}^2\|_{L(\mu)}^2.$$

**Linear GW embedding without Monge mapping.** Similar to above formulation, given two gm spaces $\mathbb{Y}^1 = (Y^1, g_{Y^1}, \nu^1), \mathbb{Y}^2 = (Y^1, g_{Y^2}, \nu^2)$ and a reference $gm-$space $\mathbb{X} = (X, g_X, \mu)$, where $\mu, \nu^1, \nu^2$ are probability measures. Suppose $\gamma^1, \gamma^2$ are optimal transportation plans for $GW(\mathbb{X}, \mathbb{Y}^1), GW(\mathbb{X}, \mathbb{Y}^2)$ respectively. Then the embedding (5) can be generalized to:

$$\mathbb{Y}^1 \mapsto \hat{k}^1 := \{(x, x') \mapsto (g_{Y^1}(\cdot_1, \cdot_2))_{\#}(\gamma^1(\cdot^1|x)\gamma^1(\cdot^2|x'))\}. \tag{42}$$

Note, similar to LOT, when $\gamma^1 = (\mathrm{id} \times T^1)_{\#}\mu$, we have $\hat{T}^1 = T^1$. That is (5),(42) coincide.

Similarly, we define the "distance" between two embeddings $\hat{k}^1, \hat{k}^2$ via:

$$\|\hat{k}^1 - \hat{k}^2\|_{L(\mu^{\otimes 2})}$$

$$:= \int_{X^2} \inf_{\substack{\gamma \in \Gamma(\gamma^1(\cdot|x), \gamma^2(\cdot|x) \\ \gamma' \in \Gamma(\gamma^1(\cdot|x'), \gamma^2(\cdot|x'))}} \|g_{Y^1}(y^1, y^{1\prime}) - g_{Y^2}(y^2, y^{2\prime})\|^2 d\gamma((y^1, y^2|x) d\gamma'(y^{1\prime}, y^{2\prime}|x') d\mu^{\otimes 2}(x, x').$$

$$\tag{43}$$

From (8), we have

$$LGW(\mathbb{Y}^1, \mathbb{Y}^2; \mathbb{Y}^1, \mathbb{Y}^2, \mathbb{X})$$
$$= \inf_{\gamma \in \Gamma(\gamma^1, \gamma^2; \mu)} |g_{Y^1}(y^1, y^{1\prime}) - g_{Y^2}(y^2, y^{2\prime})|^2 \gamma^{\otimes 2}(x, y^1, y^2)$$
$$= \int_{X^2} \inf_{\substack{\gamma \in \Gamma(\gamma^1(\cdot|x), \gamma^2(\cdot|x)) \\ \gamma' \in \Gamma(\gamma^1(\cdot|x'), \gamma^2(\cdot|x'))}} \|g_{Y^1}(y^1, y^{1\prime}) - g_{Y^2}(y^2, y^{2\prime})\|^2 d\gamma((y^1, y^2|x) d\gamma'(y^{1\prime}, y^{2\prime}|x') d\mu^{\otimes 2}(x, x')$$
$$= \|\hat{k}^1 - \hat{k}^2\|_{L(\mu^{\otimes 2})}.$$

That is, the distance between two embeddings coincides with the LGW distance between two gm spaces.

**LPGW embedding without Monge mapping.** Similar to above setting, given gm-spaces $\mathbb{X}, \mathbb{Y}^1, \mathbb{Y}^2$, and suppose $\gamma^1, \gamma^2$ are optimal transportation plans for $PGW_\lambda(\mathbb{X}, \mathbb{Y}^1), PGW_\lambda(\mathbb{X}, \mathbb{Y}^2)$, the LPGW embedding in general case is defined by

$$\mathbb{Y}^1 \mapsto E^1 := (\hat{k}^1, \gamma_X^1, \gamma_c^1 := (\nu^1)^{\otimes 2} - (\gamma_{Y^1}^1)^{\otimes 2}). \tag{44}$$

Note, when the Monge mapping assumption holds, (13),(44) coincide.

The distance between two embeddings is defined as:

$$D(E^1, E^2) := \inf_{\mu' \leq \gamma_X^1 \wedge \gamma_X^2} \|\hat{k}^1 - \hat{k}^1\|_{(\mu')^{\otimes 2}} + \lambda(|\gamma_X^1|^2 + |\gamma_X^2|^2 - 2|\mu'|^2 + |\gamma_c^1| + |\gamma_c^2|). \tag{45}$$

We have

$$LPGW_\lambda(\mathbb{Y}^1, \mathbb{Y}^2; \mathbb{X})$$
$$= \inf_{\gamma \in \Gamma_{\leq}(\nu^1, \nu^2; \mu)} \int_{(X \times Y^1 \times Y^2)^2} \|g_{Y^1}(y^1, y^{1\prime}) - g_{Y^2}(y^2, y^{2\prime})\|^2 d\gamma(x, y^1, y^2) d\gamma(x', y^{1\prime}, y^{2\prime})$$
$$\quad + \lambda(|\nu^1|^2 + |\nu^2|^2 - |\gamma|^2)$$
$$= \inf_{\mu' \leq \gamma^1 \wedge \gamma^2} \underbrace{\inf_{\substack{\gamma \in \Gamma(\gamma^1(\cdot|x), \gamma^2(\cdot|x)) \\ \gamma' \in \Gamma(\gamma^1(\cdot|x'), \gamma^2(\cdot|x'))}} \int_{X^2} \|g_{Y^1}(y^1, y^{1\prime}) - g_{Y^2}(y^2, y^{2\prime})\| d(\mu')^{\otimes 2}(x, x')}_{\|\hat{k}^1 - \hat{k}^2\|_{L((\mu')^{\otimes 2})}}$$
$$\quad + \lambda(\underbrace{|\nu^1|^2 - |\gamma_X^1|^2}_{|\gamma_c^1|} + \underbrace{|\nu^2|^2 - |\gamma_X^2|^2}_{|\gamma_X^2|^2} + |\gamma_X^1|^2 + |\gamma_X^2|^2 - 2|\mu'|^2)$$
$$= D(E^1, E^2)$$

That is, the distance between two embeddings coincides with the LPGW distance between the two gm-spaces.

## D  COMPUTATIONAL COMPLEXITY OF LPGW

**Computational Complexity of GW and PGW.** The classical solvers for the GW and PGW problems are variants of the Frank-Wolf algorithms (Chapel et al., 2020; Bai et al., 2024).

Consider metric measure spaces $\mathbb{X}$ and $\mathbb{Y}$, with measures given by $\mu = \sum_{i=1}^{n} p_i \delta_{x_i}$ and $\nu = \sum_{j=1}^{m} q_j \delta_{y_j}$, respectively. To achieve an $\epsilon-$accurate solution for the GW problem, the number of required iterations of the FW algorithm is

$$\frac{\max\left\{2L_1, 2 \cdot n^2 m^2 \max(\{2(C^X)^2 + 2(C^Y)^2\})\right\}^2}{\epsilon^2},$$

and for PGW is

$$\frac{\max\left\{2L_1, 2\min(|\mathrm{p}|,|\mathrm{q}|)\cdot n^2m^2\max(\{2(C^X)^2+2(C^Y)^2,2\lambda\})\right\}^2}{\epsilon^2}.$$

Here, $L_1$ is a value determined by the initial guess $\gamma^0$ used in the FW algorithm and $C^X, C^Y$ are the corresponding cost matrices for $\mathbb{X}, \mathbb{Y}$. Note, since the largest value for $\lambda$ is $2\lambda = \max(2(C^X)^2, 2(C^Y)^2)$, the above two quantities coincide.

In each iteration of FW, we are required to solve an OT/POT problem, whose complexity is $\mathcal{O}(nm(n+m))$ (Bonneel et al., 2011). Thus, the theoretical time complexity to solve GW/PGW is $\mathcal{O}(\frac{1}{\epsilon^2}nm(n+m)n^2m^2)$.

Given $K$ mm-spaces, $\mathbb{X}_1,\ldots\mathbb{X}_K$, then, computing their pairwise GW/PGW distances requires $\binom{K}{2}\cdot\mathcal{O}(\frac{1}{\epsilon^2}nm(n+m)n^2m^2)$. Meanwhile, computing the pairwise LPGW distances between them requires only

$$K\mathcal{O}\left(\frac{1}{\epsilon^2}nm(n+m)+n^2m^2\right)+\binom{K}{2}\mathcal{O}(n_0^2), \tag{46}$$

where the term $\mathcal{O}(n_0^2)$ represents the time complexity of computing the distance between two embeddings, and $n_0$ denotes the size of the reference space. In general, we would like to set $n_0$ to be the mean/median/maximum of the sizes of $K$ mm-spaces, and thus, this term can be ignored as compared to the first term.

*Remark* D.1. We generally impose a fixed limit on the maximum number of iterations to be used in the FW algorithm (e.g., in PythonOT (Flamary et al., 2021), it is set to 1000 by default). In practice, this maximum number of iterations is generally not achieved. Hence, un-rigorously speaking, we can think of the complexity of GW/PGW as being $\mathcal{O}(nm(n+m))$.

## E    PROOF OF PROPOSITION 2.1

### E.1    CLARIFICATION

Although statements (1) and (2) in this proposition are not explicitly introduced in (Beier et al., 2022), they are implicitly mentioned, for example, in Proposition III.1 of such article. Thus, we do not claim the proofs of statements (1) and (2) as contributions of this paper. Additionally, to the best of our knowledge, the statement (3) has not yet been studied. We present the related proof as a complement to the work in (Beier et al., 2022).

### E.2    CONJECTURE AND UNDERSTANDING

Regarding statement (3), we conjecture that this conclusion can be extended to the case where $g_X$ and $g_Y$ are squared Euclidean distances. However, this statement may not hold for other gauge mappings. In our understanding, for general gauge mappings, achieving a similar result to the statement (3) would require defining a generalized barycentric projection mapping that is dependent on the specific gauge mapping.

The formulation of a new barycentric projection mapping based on the chosen gauge mapping, as well as a generalized version of statement (3), is left for future work.

### E.3    PROOF OF PROPOSITION 2.1: PART (1)

**Proof in a particular case.** We first show the result in a simplified case. In particular, suppose $\gamma^*$ satisfies the Monge mapping, i.e. $\gamma^* = (\mathrm{id}\times T)_{\#}\mu$, where $T : X \to Y$, and $\gamma^0 = (\mathrm{id}\times\mathrm{id})_{\#}\mu \in \Gamma^*(\mathbb{X},\mathbb{X})$. Then $k_{\gamma^0}\equiv 0$ and thus

$$\begin{aligned}LGW(\mathbb{X},\mathbb{Y};\mathbb{X},\gamma^0,\gamma^*) &= \|k_{\gamma^*}-k_{\gamma^0}\|_{\mu^{\otimes 2}}^2 = \|k_{\gamma^*}\|_{\mu^{\otimes 2}}^2\\ &= \int_{(X\times Y)^{\otimes 2}}|g_X(x,x')-g_Y(T(x),T(x'))|^2 d\mu^{\otimes 2}\\ &= GW(\mathbb{X},\mathbb{Y}).\end{aligned}$$

**Proof in the general case.** Consider $\gamma^0 \in \Gamma^*(\mathbb{X}, \mathbb{X})$, and $\gamma \in \Gamma(\gamma^0, \gamma^*)$.

We claim the following lemma

**Lemma E.1.** *Choose $\gamma^0 \in \Gamma^*(\mathbb{X}, \mathbb{X})$, $\gamma^* \in \mathcal{M}_+(\mathbb{X}, \mathbb{Y})$, for each $\gamma \in \Gamma(\gamma^0, \gamma^*; \mu)$, we have the following identity:*

$$\int_{(X \times X \times Y)^{\otimes 2}} |g_X(x^2, x^{2\prime}) - g_Y(y, y')|^2 d(\gamma)^{\otimes 2}((x^1, x^{1\prime}), (x^2, x^{2\prime}), (y, y'))$$

$$= \int_{(X \times Y)^{\otimes 2}} |g_X(x, x') - g_Y(y, y')|^2 d(\gamma^*)^{\otimes 2}((x, x'), (y, y')) \tag{47}$$

*Proof of Lemma (E.1).* Since

$$GW(\mathbb{X}, \mathbb{X}) = \int_{(X \times X)^{\otimes 2}} |g_X(x^1, x^{1\prime}) - g_X(x^2, x^{2\prime})|^2 d\gamma^0 = 0,$$

we have

$$g_X(x^1, x^{1\prime}) - g_X(x^2, x^{2\prime}) = 0, \quad \gamma^0 - a.s.$$

For convenience, let $X^1 = X^2 = X$ be independent copies of $X$. We have:

$$\int_{(X \times X \times Y)^{\otimes 2}} |g_X(x^2, x^{2\prime}) - g_Y(y, y')|^2 d\gamma^{\otimes 2}((x^1, x^{1\prime}), (x^2, x^{2\prime}), (y, y'))$$

$$= \int_{(X \times X^0 \times Y)^{\otimes 2}} |g_X(x^2, x^{2\prime}) - g_X(x^1, x^{1\prime}) + g_X(x^1, x^{1\prime}) - g_Y(y, y')|^2 d\gamma^{\otimes 2}((x^1, x^{1\prime})(x^2, x^{2\prime}), (y, y'))$$

$$= \int_{(X^1 \times X^2)^{\otimes 2}} \left[ \int_{(Y)^{\otimes 2}} |g_X(x^2, x^{2\prime}) - g_X(x^1, x^{1\prime}) + g_X(x^1, x^{1\prime}) - g_Y(y, y')|^2 \right.$$

$$\left. d\gamma^{\otimes 2}_{Y|(X^1, X^2)}((y, y')|(x^1, x^{1\prime}), (x^2, x^{2\prime})) \right] d(\gamma^0)^{\otimes 2}((x^1, x^{1\prime}), (x^2, x^{2\prime})) \tag{48}$$

$$= \int_{(X^1 \times X^2)^{\otimes 2}} \left[ \int_{Y^{\otimes 2}} |g_X(x^1, x^{1\prime}) - g_Y(y, y')|^2 d\gamma^{\otimes 2}_{Y|X^1, X^2}((y, y')|(x^1, x^{1\prime}), (x^2, x^{2\prime})) \right]$$

$$d(\gamma^0)^{\otimes 2}((x^1, x^{1\prime}), (x^2, x^{2\prime}))$$

$$= \int_{(X^1 \times Y)^{\otimes 2}} |g_X(x^1, x^{1\prime}) - g_Y(y, y')|^2 d\gamma^{\otimes 2}((x^1, x^2, y), (x^{1\prime}, x^{2\prime}, y'))$$

$$= \int_{(X \times Y)^{\otimes 2}} |g_X(x, x') - g_Y(y, y')|^2 d(\gamma^*)^{\otimes 2}((x, y), (x', y')) \tag{49}$$

where in (48) $\gamma_{Y|X^1, X^2}(\cdot|x^1, x^2)$ is the disintegration of $\gamma$ with respect to $(\pi_{X^1, X^2})_\# \gamma = \gamma^0$. $\qquad \square$

Based on the identity (47) and the fact $\gamma^1$ is optimal for problem $GW(\mathbb{X}, \mathbb{Y})$, we have:

$$LGW(\mathbb{X}, \mathbb{Y}; \mathbb{X}, \gamma^0, \gamma^*) = \inf_{\gamma \in \Gamma(\gamma^0, \gamma^*)} \int_{(X \times X \times Y)^{\otimes 2}} |d_X(x^0, x^{0\prime}) - d_Y(y, y')|^2 d\gamma^{\otimes 2}((x, x'), (x^0, x^{0\prime}), (y, y'))$$

$$= \inf_{\gamma \in \Gamma(\gamma^0, \gamma^*)} GW(\mathbb{X}, \mathbb{Y})$$

$$= GW(\mathbb{X}, \mathbb{Y}),$$

and we have completed the proof.

### E.4 PROOF OF PROPOSITION 2.1: PART (2)

In this case, $\gamma^* = (id \times T)_\# \mu$, for each $x \in \text{Supp}(\mu)$, we have that the disintegration of $\gamma^*$ with respect to its first marginal $(\pi_X)_\# \gamma^* = \gamma^*_X$, given first component is $x$, is

$$\gamma^*_{Y|X}(\cdot_2|y') = \delta(\cdot_2 - T(x)),$$

where $\delta$ is the Dirac measure. Thus, for all $x \in \text{supp}(\mu)$, we have:

$$\mathcal{T}_{\gamma^*}(x) = \int_Y y d\gamma^*_{Y|X}(y|x) = \int_Y y \, d\delta(y - T(x)) = T(x).$$

where the last equality holds from the fact $T(x) \in \text{supp}(\nu) \subset Y$. Therefore, $\mathcal{T}_{\gamma^*} = T \ \mu$−a.s. Since $T_{\#}\mu = \nu$, we have $\widetilde{\nu}_{\gamma^*} := (\mathcal{T}_{\gamma^*})_{\#}\mu = T_{\#}\mu = \nu$.

### E.5 Proof of Proposition 2.1: Part (3)

#### E.5.1 Proof in the Discrete Case

We first demonstrate the proof in the following simplified discrete case: Suppose $g_X(x, x') = x^\top x'$ and $g_Y(y, y') = y^\top y'$, and consider

$$\mu = \sum_{i=1}^n p_i \delta_{x_i}, \qquad \nu = \sum_{j=1}^m q_j \delta_{y_j} \qquad \text{with } \sum_{i=1}^n p_i = 1 = \sum_{j=1}^m q_k.$$

In addition, we suppose that $X$, $Y$ are compact convex sets which contains 0 (besides containing $\{x_1, \ldots, x_n\}$ and $\{y_1, \ldots, y_m\}$, respectively).

Let $\gamma^* \in \Gamma^*(\mathbb{X}, \mathbb{Y})$ be an optimal transportation plan. Then, the corresponding barycentric projected measure is given by

$$\widetilde{\nu} = \widetilde{\nu}_{\gamma^*} := \sum_{i=1}^n p_i \delta_{\widetilde{y}_i}, \qquad \text{where } \widetilde{y}_i := \frac{1}{p_i} \sum_{j=1}^m \gamma^*_{ij} y_j.$$

By convexity, $\widetilde{y}_1, \ldots, \widetilde{y}_n \in Y$. Thus $\widetilde{\mathbb{Y}} := \widetilde{\mathbb{Y}}_{\gamma^*} = (X, g_Y, \widetilde{\nu})$.

Then, it induces a transportation plan $\widetilde{\gamma}^* := \text{diag}(p_1^0, \cdots, p_n^0) \in \mathbb{R}_+^{n \times n}$ with first and second marginals $\mu$ and $\widetilde{\nu}$, respectively. We will show that for any transportation plan $\widetilde{\gamma} \in \Gamma(\mu, \widetilde{\nu})$, it holds that

$$C(\widetilde{\gamma}^*; \mathbb{X}, \widetilde{\mathbb{Y}}) \leq C(\widetilde{\gamma}; \mathbb{X}, \widetilde{\mathbb{Y}}),$$

where

$$C(\gamma; \mathbb{X}, \mathbb{Y}) = \sum_{i,i'=1}^n \sum_{j,j'=1}^m (x_i^\top x_{i'} - y_j^\top y_{j'})^2 \gamma_{ij} \gamma_{i'j'},$$

in other words, we will show that $\widetilde{\gamma}^*$ is optimal for $\text{GW}(\mathbb{X}, \widetilde{\mathbb{Y}})$.

First, we notice that $(\widetilde{\gamma}^*)^{-1} = \text{diag}(1/p_1^0, \cdots, 1/p_n^0)$. Then, we set

$$\gamma := \widetilde{\gamma}(\widetilde{\gamma}^*)^{-1} \gamma^* \in \mathbb{R}_+^{n \times m}.$$

Secondly, we check that $\gamma \in \Gamma(\mu, \nu)$. In fact, let $p = (p_1, \ldots, p_n)$, $q = (q_1, \ldots, q_m)$ be the vectors of the weights of finite discrete measures $\mu$ and $\nu$, respectively, then, since $\gamma^* 1_m = p$ and $\tilde{\gamma}^T 1_n = q$, we have

$$\gamma 1_m = \widetilde{\gamma}(\widetilde{\gamma}^*)^{-1} \gamma^* 1_m = \widetilde{\gamma}(\widetilde{\gamma}^*)^{-1} p = \widetilde{\gamma} 1_n = q,$$
$$\gamma^\top 1_n = (\gamma^*)^\top (\widetilde{\gamma}^*)^{-1} \widetilde{\gamma}^\top 1_n = (\gamma^*)^\top (\widetilde{\gamma}^*)^{-1} q = (\gamma^*)^\top 1_m = p.$$

Lastly, we compute the costs as follows:

$$C(\gamma^*; \mathbb{X}, \mathbb{Y}) = \sum_{i,i'=1}^n (x_i^\top x_{i'})^2 p_i p_{i'} + \sum_{j,j'=1}^m (y_j^\top y_{j'})^2 q_j q_{j'} - 2 \sum_{i,i'=1}^n \sum_{j,j'=1}^m (x_i^\top x_{i'})(y_j^\top y_{j'}) \gamma^*_{ij} \gamma^*_{i'j'}$$

$$= J_1 - 2 \sum_{i,i'=1}^n \sum_{j,j'=1}^m (x_i^\top x_{i'})(y_j^\top y_{j'}) \gamma^*_{ij} \gamma^*_{i'j'}.$$

where $J_1 = \sum_{i,i'=1}^n (x_i^\top x_{i'})^2 p_i p_{i'} + \sum_{j,j'=1}^m (y_j^\top y_{j'})^2 q_j q_{j'}$ is independent of $\gamma^*$. By the multilinearity of the last term in the identity, we can show

$$
\begin{aligned}
C(\widetilde{\gamma}^*; \mathbb{X}, \widetilde{\mathbb{Y}}) &= J_2 - 2 \sum_{i,i'=1}^n \sum_{\ell,\ell'=1}^n (x_i^\top x_{i'})(\widetilde{y}_\ell^\top \widetilde{y}_{\ell'}) \widetilde{\gamma}_{i\ell}^* \widetilde{\gamma}_{i',\ell'}^* \\
&= J_2 - 2 \sum_{i,i'=1}^n \sum_{\ell,\ell'=1}^n x_i^\top x_{i'} \sum_{j,j'=1}^m \left( \frac{\gamma_{\ell j}^* y_j^\top}{p_\ell} \frac{\gamma_{\ell' j'}^* y_{j'}}{p_{\ell'}} \right) \widetilde{\gamma}_{i\ell} \widetilde{\gamma}_{i'\ell'} \\
&= J_2 - J_1 + C(\gamma^*; \mu, \nu^1),
\end{aligned}
\tag{50}
$$

where $J_2 = \sum_{i,i'=1}^n (x_i^\top x_{i'})^2 p_i p_{i'}^0 + \sum_{\ell,\ell'=1}^n (\widetilde{y}_\ell^\top \widetilde{y}_{\ell'})^2 p_\ell^0 p_{\ell'}^0$. Similarly, we can compute

$$
C(\widetilde{\gamma}; \mathbb{X}, \widetilde{\mathbb{Y}}) = J_2 - J_1 + C(\gamma; \mathbb{X}, \mathbb{Y})
\tag{51}
$$

The optimality of $\gamma^*$ together with identities (50) and (51) implies that $C(\widetilde{\gamma}^*; \mathbb{X}, \widetilde{\mathbb{Y}}) \leq C(\widetilde{\gamma}; \mathbb{X}, \widetilde{\mathbb{Y}})$.

### E.5.2 Proof in the General Case

Similar to the discrete case, given $\gamma^* \in \Gamma^*(\mathbb{X}, \mathbb{Y})$, the barycentric projection is given by

$$
\mathcal{T}_{\gamma^*}(x) = \int_Y y \, d\gamma_{Y|X}^*(y|x).
$$

We have that $\widetilde{\gamma}^* = (\mathrm{id} \times \mathcal{T}_{\gamma^*})_\# \mu$ is a joint measure with first and second marginals $\mu$ and $\widetilde{\nu} := \widetilde{\nu}_{\gamma^*} := (\mathcal{T}_{\gamma^*})_\# \mu$, respectively. The goal is to show that $\widetilde{\gamma}^*$ is optimal for $GW(\mathbb{X}, \widetilde{\mathbb{Y}})$ where $\widetilde{\mathbb{Y}} := \widetilde{\mathbb{Y}}_{\gamma^*} := (\mathcal{T}_{\gamma^*})_\# \mu$.

Let $\widetilde{\gamma} \in \Gamma(\mu, \widetilde{\nu}_{\gamma^*})$ be an arbitrary plan. Let $\widetilde{\gamma}(\cdot_2|x) := \widetilde{\gamma}_{Y|X}(\cdot_2|x)$ denote the disintegration of $\widetilde{\gamma}$ with respect to $\mu$, given that the first component is $x$, for each $x \in \mathrm{Supp}(\mu)$. Similarly, we adopt the notation $\gamma^*(\cdot_2|x)$ for each $x \in \mathrm{Supp}(\mu)$.

We define the joint measure

$$
\gamma := \gamma_{Y|X}^* \widetilde{\gamma}_{X|Y}^* \widetilde{\gamma}
\tag{52}
$$

on $X \times Y$. In particular, the above notation means that for each test function $\phi \in C_0(X \times Y)$, we have

$$
\begin{aligned}
\int_{X \times Y} \phi(x, y) d\gamma(x, y) &= \int_{X \times Y \times X \times Y} \phi(x, y) d\gamma^*(y|x^0) d\widetilde{\gamma}^*(x^0|\widetilde{y}) d\widetilde{\gamma}(x, \widetilde{y}) \\
&= \int_{X \times Y} \left( \int_X \left( \int_Y \phi(x, y) d\gamma^*(y|x^0) \right) d\widetilde{\gamma}^*(x^0|\widetilde{y}) \right) d\widetilde{\gamma}(x, \widetilde{y})
\end{aligned}
$$

For any test functions $\phi_X \in C_0(X), \phi_Y \in C_0(Y)$ we have

$$
\int_{X \times Y} \phi_X(x) d\gamma(x,y) = \int_{X \times Y \times X \times Y} \phi_X(x) d\gamma^*(y|x^0) d\widetilde{\gamma}^*(x^0|\widetilde{y}) d\widetilde{\gamma}(x, \widetilde{y})
$$

$$
= \int_{X \times Y} \phi_X(x) d\widetilde{\gamma}(x, \widetilde{y})
$$

$$
= \int_X \phi_X(x) d\mu(x)
$$

$$
\int_{X \times Y} \phi_Y(y) d\gamma(x,y) = \int_{X \times Y \times X \times Y} \phi_Y(y) d\gamma^*(y|x^0) d\widetilde{\gamma}^*(x^0|\widetilde{y}) d\widetilde{\gamma}(x, \widetilde{y})
$$

$$
= \int_{X \times Y \times X} \phi_Y(y) d\gamma^*(y|x^0) d\widetilde{\gamma}^*(x^0|\widetilde{y}) d\widetilde{\nu}(\widetilde{y})
$$

$$
= \int_{X \times Y} \phi_Y(y) d\gamma^*(y|x^0) d\mu(x^0)
$$

$$
= \int_{X \times Y} \phi_Y(y) d\gamma^*(x^0, y)
$$

$$
= \int_Y \phi_Y(y) d\nu(y)
$$

Thus $\gamma \in \Gamma(\mu, \nu)$.

In addition, we observe that we have the following property that holds for the case of considering inner products:

**Lemma E.2.** *Suppose $X \subset \mathbb{R}^{d_x}, Y \subset \mathbb{R}^{d_y}$ are finite-dimensional, convex compact sets. Let $\gamma \in \mathcal{M}_+(X \times Y)$ and define $\mathcal{T}_\gamma$ as in (9). In addition, let $g_Y(y, y') = \alpha \langle y, y' \rangle$, where $(y, y') \mapsto \langle y, y' \rangle := \sum_{i=1}^{d_1} y_i y_i'$ is an inner product (the standard one, for example). Then for each $x, x' \in X$, let $\widetilde{y} = \mathcal{T}_\gamma(x), \widetilde{y}' = \mathcal{T}_\gamma(x')$, we have:*

$$
g_Y(\widetilde{y}, \widetilde{y}') = \int_{Y \times Y} g_Y(y, y') d\gamma(y|x) d\gamma(y'|x'). \tag{53}
$$

*Proof of Lemma E.2.* For each $i \in [1 : d_y]$, we have:

$$
\widetilde{y}[i]\widetilde{y}'[i] = \int_X y[i] d\gamma(y|x) \int_X y'[i] d\gamma(y'|x')
$$

$$
= \int_X y[i] \int_{y'} y'[i] d\gamma(y|x) d\gamma(y'|x')
$$

$$
= \int_{Y \times Y} y[i] y'[i] d\gamma(y|x) d\gamma(y'|x')
$$

where the third equality follows from Fubini's theorem.

Thus,

$$
g_Y(\widetilde{y}, \widetilde{y}') = \alpha \sum_{i=1}^{d_y} \widetilde{y}[i]\widetilde{y}'[i]
$$

$$
= \alpha \int_{Y \times Y} y[i] y'[i] d\gamma(y|x) d\gamma(y'|x')
$$

$$
= \int_{Y \times Y} \alpha \sum_{i=1}^{d_y} y[i] y'[i] d\gamma(y|x) d\gamma(y'|x')
$$

$$
= \int_{Y \times Y} g_X(y, y') d\gamma(y|x) d\gamma(y'|x')
$$

$\square$

Now, we continue with the proof of Proposition 2.1, part (3).

We have

$$
\begin{aligned}
C(\gamma^*; \mathbb{X}, \mathbb{Y}) &= \int_{(X \times Y)^{\otimes 2}} |g_X(x, x') - g_Y(y, y')|^2 d\gamma^*(x, y) d\gamma^*(x', y') \\
&= \|g_X(x, x')\|^2_{L^2(\mu^{\otimes 2})} + \|g_Y(y, y')\|^2_{L^2(\nu^{\otimes 2})} - 2 \int_{(X \times X)^{\otimes 2}} g_X(x, x') g_Y(y, y') d\gamma^*(x, y) d\gamma^*(x', y') \\
&= J_1 - 2 \langle g_X g_Y, (\gamma^*)^{\otimes 2} \rangle
\end{aligned}
\tag{54}
$$

where $J_1 = \|g_X\|^2_{L^2(\mu^{\otimes 2})} + \|g_Y\|^2_{L^2(\nu^{\otimes 2})}$ is independent to $\gamma^*$. Similarly,

$$
\begin{aligned}
C(&\widetilde{\gamma}^*; \mathbb{X}, \widetilde{\mathbb{Y}}) \\
&= \int_{(X \times Y)^{\otimes 2}} |g_X(x, x') - g_Y(\widetilde{y}, \widetilde{y}')|^2 d\widetilde{\gamma}^*(x, y) d\widetilde{\gamma}^*(x', y') \\
&= \|g_X\|^2_{L^2(\mu^{\otimes 2})} + \|g_Y\|^2_{L^2(\widetilde{\nu}^{\otimes 2})} - 2 \int_{(X \times Y)^{\otimes 2}} g_X(x, x') g_Y(\widetilde{y}, \widetilde{y}') d\widetilde{\gamma}^*(x, \widetilde{y}) d\widetilde{\gamma}^*(x', \widetilde{y}') \\
&= J_2 - 2 \int_{X^{\otimes 2}} g_X(x, x') g_Y(\mathcal{T}_{\gamma^*}(x), \mathcal{T}_{\gamma^*}(x')) d\mu(x) d\mu(x') \tag{55} \\
&= J_2 - 2 \int_{X^{\otimes 2}} g_X(x, x') \left[ \int_{Y^{\otimes 2}} g_Y(y, y') d\gamma^*(y|x) d\gamma^*(y'|y') \right] d\mu(x) d\mu(x') \tag{56} \\
&= J_2 - 2 \int_{(X \times Y)^{\otimes 2}} g_X(x, x') g_Y(y, y') d\gamma^*(x, y) d\gamma^*(x', y') \\
&= J_2 - 2 \langle g_X g_Y, (\gamma^*)^{\otimes 2} \rangle \tag{57}
\end{aligned}
$$

where in (55), $J_2 = \|g_X(\cdot_1, \cdot_2)\|^2_{L^2(\mu^{\otimes 2})} + \|g_Y(\cdot_1, \cdot_2)\|^2_{L^2(\widetilde{\nu}^{\otimes 2})}$ (which is independent of $\widetilde{\gamma}^*$); and (56) follows from Lemma E.2. Combining (54) and (57), we have

$$
C(\widetilde{\gamma}^*; \mathbb{X}, \widetilde{\mathbb{Y}}) = C(\gamma^*; \mathbb{X}, \mathbb{Y}) - J_1 + J_2. \tag{58}
$$

Similarly, we can show that

$$
C(\widetilde{\gamma}; \mathbb{X}, \widetilde{\mathbb{Y}}) = C(\gamma; \mathbb{X}, \mathbb{Y}) - J_1 + J_2. \tag{59}
$$

Also, similarly to (54) and (57), we have

$$
C(\gamma; \mathbb{X}, \mathbb{Y}) = J_1 - 2 \langle g_X g_Y, \gamma^{\otimes 2} \rangle \tag{60}
$$

$$
C(\widetilde{\gamma}; \mathbb{X}, \widetilde{\mathbb{Y}}) = J_2 - 2 \langle g_X g_Y, \widetilde{\gamma}^{\otimes 2} \rangle \tag{61}
$$

It remains to show

$$
\langle g_X g_Y, \gamma^{\otimes 2} \rangle = \langle g_X g_Y, \widetilde{\gamma}^{\otimes 2} \rangle. \tag{62}
$$

Indeed,

$$
\begin{aligned}
\langle g_X g_Y, \widetilde{\gamma}^{\otimes 2}\rangle &= \int_{(X\times Y)^{\otimes 2}} g_X(x,x') g_Y(\widetilde{y},\widetilde{y}') d\widetilde{\gamma}(x,\widetilde{y}) d\widetilde{\gamma}(x',\widetilde{y}') \\
&= \int_{(X\times Y)^{\otimes 2}} g_X(x,x') g_Y(\widetilde{y},\widetilde{y}') d\widetilde{\gamma}(\widetilde{y}|x) d\widetilde{\gamma}(\widetilde{y}'|x'') d\mu(x) d\mu(x') \\
&= \int_{(X\times Y)^{\otimes 2}} g_X(x,x') \left[ \int_{X^{\otimes 2}} g_Y(\mathcal{T}_{\gamma^*}(x^0), \mathcal{T}_{\gamma^*}(x^{0'})) d\widetilde{\gamma}^*(x^0|\widetilde{y}) d\widetilde{\gamma}^*(x^{0'}|\widetilde{y}') \right] \\
&\quad d\widetilde{\gamma}(\widetilde{y}|x) d\widetilde{\gamma}(\widetilde{y}'|x') d\mu(x) d\mu(x') \qquad\qquad\qquad\qquad\qquad (63) \\
&= \int_{(X\times Y\times X)^{\otimes 2}} g_X(x,x') g_Y(\mathcal{T}_{\gamma^*}(x^0), \mathcal{T}_{\gamma^*}(x^{0'})) \\
&\quad d\widetilde{\gamma}^*(x^0|\widetilde{y}) d\widetilde{\gamma}^*(x^{0'}|\widetilde{y}') d\widetilde{\gamma}(\widetilde{y}|x) d\widetilde{\gamma}(\widetilde{y}'|x'') d\mu(x) d\mu(x') \\
&= \int_{(X\times Y\times X\times Y)^{\otimes 2}} g_X(x,x') \left[ \int_{Y^{\otimes 2}} g_Y(y,y') d\gamma^*(y|x^0) d\gamma^*(y'|x^{0'}) \right] \\
&\quad d\widetilde{\gamma}^*(x^0|\widetilde{y}) d\widetilde{\gamma}^*(x^{0'}|\widetilde{y}') d\widetilde{\gamma}(\widetilde{y}|x) d\widetilde{\gamma}(\widetilde{y}'|x') d\mu(x) d\mu(x') \qquad (64) \\
&= \int_{(X\times Y\times X\times Y)^{\otimes 2}} g_X(x,x') g_Y(y,y') \left( d\gamma^*(y|x^0) d\widetilde{\gamma}^*(x^0|\widetilde{y}) d\widetilde{\gamma}(\widetilde{y}|x) d\mu(x) \right) \\
&\quad \left( d\gamma^*(y'|x'^{0'}) d\widetilde{\gamma}^*(x^{0'}|\widetilde{y}') d\widetilde{\gamma}(\widetilde{y}'|x') d\mu(x') \right) \\
&= \int_{(X\times Y)^{\otimes 2}} g_X(x,x') g_Y(y,y') d\gamma(x,x) d\gamma(x',y') \\
&= \langle g_X g_Y, \gamma^{\otimes 2}\rangle
\end{aligned}
$$

where (63) holds since given $\widetilde{y}$, theprobability measure $\widetilde{\gamma}^*(\cdot_1|\widetilde{y})$ is only supported on $\{x^0 \in X : \widetilde{y} = \mathcal{T}_{\gamma^*}(x^0)\}$, similarly to measure $\gamma(\cdot_1|\widetilde{y}')$; (64) follows from Lemma E.2 (that is, from equality (53)).

Combining (59), (58), (62) and the fact $C(\gamma^*; \mathbb{X}, \mathbb{Y}) \leq C(\gamma; \mathbb{X}, \mathbb{Y})$, we obtain that $C(\widetilde{\gamma}^*; \mathbb{X}, \mathbb{Y}) \leq C(\widetilde{\gamma}; \mathbb{X}, \mathbb{Y})$ which completes the proof.

## F   Proof of Proposition 3.1

Consider gm-spaces $\mathbb{X}, \mathbb{Y}^1, \mathbb{Y}^2$, and select $\gamma^1 \in \Gamma^*_{\leq,\lambda}(\mathbb{X}, \mathbb{Y}^1), \gamma^2 \in \Gamma^*_{\leq,\lambda}(\mathbb{X}, \mathbb{Y}^2)$. Inspired by the LGW distance (38), we propose the LPGW distance (conditioned on $\gamma^1, \gamma^2$) (15) in the general case:

$$
\begin{aligned}
LPGW_\lambda(\mathbb{Y}^1, \mathbb{Y}^2; \mathbb{X}, \gamma^1, \gamma^2) := \inf_{\gamma \in \Gamma_\leq(\gamma^1,\gamma^2;\mu)} &\int_{(X\times Y^1\times Y^2)^{\otimes 2}} |g_{Y^1}(y^1,y^{1'}) - g_{Y^2}(y^2,y^{2'})|^2 d\gamma^{\otimes 2} \\
&+ \lambda(|\nu^1|^2 + |\nu^2|^2 - 2|\gamma|^2). \qquad (65)
\end{aligned}
$$

where $\Gamma_\leq(\gamma^1,\gamma^2;\mu) := \{\gamma \in \mathcal{M}_+(X\times Y^1\times Y^2) : (\pi_{X,Y^1})_\#\gamma \leq \gamma^1, (\pi_{X,Y^2})_\#\gamma \leq \gamma^2\}$.

Next, we discuss the proof of Proposition 3.1.

### F.1   Proof of Proposition 3.1: Part (1)

In this section, we discuss the proof of statement (1) of Proposition 3.1; namely, we prove the existence of a minimizer to the LPGW problem.

First, we introduce a series of lemmas.

**Lemma F.1.** *Given Radon measures $\mu, \nu^1, \nu^2$, and $\gamma^1 \in \Gamma_\leq(\mu, \nu^1), \gamma^2 \in \Gamma_\leq(\mu, \nu^2)$, then*

$$
\Gamma_\leq(\gamma^1, \gamma^2; \mu)
$$

*is sequentially compact set.*

*Proof.* The main idea is similar to the proof in, e.g., Lemma B.2 (Liu et al., 2023). In particular, consider a sequence $(\gamma^n) \subset \Gamma_{\leq,X}(\gamma_X, \gamma_Y)$. It is straightforward to verify that this sequence is bounded above in total variation (since $\gamma^1$ and $\gamma^2$ are finite measures) and that it is also a tight sequence. This verification is even simpler because $X, Y^1$, and $Y^2$ are compact. Thus, by Prokhorov's theorem for signed measures, the closure of $\Gamma_{\leq}(\gamma^1, \gamma^2)$ (in the weak topology) is weakly sequentially compact in $\mathcal{M}(X \times Y^1 \times Y^2)$.

It remains to show $\Gamma_{\leq}(\gamma^1, \gamma^2; \mu)$ is closed.

Suppose $\gamma^n \overset{w*}{\to} \gamma \in \mathcal{M}(X \times Y^1 \times Y^2)$, for each nonnegative test function $\phi \in C(X \times Y^1)$, we have

$$\lim_{n \to \infty} \int_{X \times Y^1} \phi(x, y^1) d(\pi_{X,Y^1})_\# \gamma^n(x, y^1) = \lim_{n \to \infty} \int_{X \times Y^1 \times Y^2} \phi(x, y^1) d\gamma^n(x, y^1, y^2)$$

$$= \int_{X \times Y^1 \times Y^2} \phi(x, y^1) d\gamma(x, y^1, y^2)$$

$$= \int_{X \times Y^1} \phi(x, y^1) d(\pi_{X,Y^1})_\# \gamma(x, y^1)$$

That is, $(\pi_{X,Y^1})_\# \gamma^n \overset{w*}{\to} (\pi_{X,Y^1})_\# \gamma$. By Lemma B.1 in (Liu et al., 2023), we have $(\pi_{X,Y^1})_\# \gamma \leq \gamma^1$. Similarly we have $(\pi_{X,Y^2})_\# \gamma \leq \gamma^2$.

Thus, $\gamma \in \Gamma_{\leq}(\gamma^1, \gamma^2; \mu)$ and we have completed the proof. $\square$

**Lemma F.2.** *Suppose $g_X, g_Y$ are continuous functions (see assumption (3)), we claim that the mapping*

$$(X \times Y)^{\otimes 2} \ni ((x, y), (x', y')) \mapsto |g_X(x, x') - g_Y(y, y')|^2 \in \mathbb{R}$$

*is a Lipschitz function with respect to metric*

$$D_{X \times Y}((x, y), (x', y')) = d_X(x, x') + d_Y(y, y').$$

*Proof of Lemma F.2.* Let the mapping in the statement of the lemma be denoted by $\Phi$. For $x_1, x_1', x_2, x_2' \in X, y_1, y_1', y_2, y_2' \in Y$ we have

$$|\Phi((x_1, y_1), (x_1', x_1')) - \Phi((x_2, y_2), (x_2', y_2'))|$$

$$= ||g_X(x_1, x_1') - g_Y(y_1, y_1')|^2 - |g_X(x_2, x_2') - g_Y(y_2, y_2')|^2|$$

$$\leq ||g_X(x_1, x_1') - g_Y(y_1, y_1')|^2 - |g_X(x_2, x_2') - g_Y(y_1, y_1')|^2|$$

$$+ ||g_X(x_2, x_2') - g_Y(y_1, y_1')|^2 - |g_X(x_2, x_2') - g_Y(y_2, y_2')|^2|$$

$$\leq K_1(|g_X(x_1, x_1') - g_X(x_2, x_2')| + |g_Y(y_1, y_1') - g_Y(y_2, y_2')|) \tag{66}$$

$$= K_1 K_2 |d_X(x_1, x_2) + d_X(x_1', x_2') + d_Y(y_1, y_2) + d_Y(y_1', y_2')| \tag{67}$$

where (66) follows from the fact that the function $r_1 \mapsto |r_1 - r_2|^2$ is Lipschitz on a compact set (see, e.g., Lemma C.1 in (Bai et al., 2024)), and $K_1 \geq 0$ is its Lipschitz constant. Then, (67) follows from assumption (3) for $g_X$ and $g_Y$, with $K_2 \geq 0$ being the maximum of their Lipschitz constants. $\square$

Consider $(\gamma^n) \subset \Gamma_{\leq}(\gamma^1, \gamma^2; \mu)$, such that

$$\langle |g_{Y^1} - g_{Y^2}|^2 - 2\lambda, (\gamma^n)^{\otimes 2}\rangle \to \inf_{\gamma \in \Gamma_{\leq}(\gamma^1, \gamma^2; \mu)} \langle |g_{Y^1} - g_{Y^2}|^2 - 2\lambda, \gamma^{\otimes 2}\rangle.$$

By compactness of $\Gamma_{\leq}(\gamma^1, \gamma^2; \mu)$ (see Lemma F.1), we have that there exists $\gamma^* \in \Gamma_{\leq}(\gamma^1, \gamma^2; \mu)$ which is sub-sequence limit of $\gamma^n$ (in weak convergence).

By Lemma F.2 and Lemma C.2.(3) in (Bai et al., 2024), we have

$$\langle |g_{Y^1} - g_{Y^2}|^2 - 2\lambda, (\gamma^*)^{\otimes 2}\rangle = \inf_{\gamma \in \Gamma_{\leq}(\gamma^1, \gamma^2; \mu)} \langle |g_{Y^1} - g_{Y^2}|^2 - 2\lambda, \gamma^{\otimes 2}\rangle.$$

Thus, $\gamma^*$ is a minimizer, and we have completed the proof.

### F.2 PROOF OF PROPOSITION 3.1: PART (2)

*Proof.* Under the Monge mapping assumptions, we have

$$\Gamma_{\le}(\gamma^1, \gamma^2; \mu) = \{(\mathrm{id} \times T^1 \times T^2)_{\#}\mu' : \mu' \le \gamma_X^1 \wedge \gamma_X^2\}.$$

Thus, (65) becomes

$LPGW(\mathbb{Y}^1, \mathbb{Y}^2; \mathbb{X}, \gamma^1, \gamma^2)$

$= \inf\limits_{\mu' \le \gamma_x^1 \wedge \gamma_X^2} \int_{X^{\otimes 2}} |g_{Y^1}(T^1(x), T^1(x')) - g_{Y^2}(T^2(x), T^2(x'))|^2 + \lambda(|\nu^1|^2 + |\nu^2|^2 - 2|(\mathrm{id} \times T^1 \times T^2)_{\#}\mu'|^2)$

$= \inf\limits_{\mu' \le \gamma_X^1 \wedge \gamma_X^2} \int_{X^{\otimes 2}} |g_{Y^1}(T^1(x), T^1(x')) - g_{Y^2}(T^2(x), T^2(x'))|^2 + \lambda(|\nu^1|^2 + |\nu^2|^2 - 2|\mu'|^2)$

$= (14).$

and we have proven the statement. □

### F.3 PROOF OF PROPOSITION 3.1: PART (3)

Since $X, Y^1, Y^2$ are compact, by continuity (see (3)), we have that $g_X, g_{Y^1}, g_{Y^2}$ are bounded functions.

By Lemma E.2 in (Bai et al., 2024), when $2\lambda > \max(|g_{Y^1}(y^1, y^{1\prime}) - g_{Y^2}(y^2, y^{2\prime})|^2)$, pick any optimal

$$\gamma \in \Gamma_{\le}(\gamma^1, \gamma^2; \mu) = \{(\mathrm{id} \times T^1 \times T^2)_{\#}\mu' : \mu' \le \gamma_X^1 \wedge \gamma_X^2\},$$

we have $|\gamma| = \min(|\gamma^1|, |\gamma^2|) = \min(|\gamma_X^1|, |\gamma_X^2|)$. Thus $\gamma = (\mathrm{id} \times T^1 \times T^2)_{\#}\gamma_X^1 \wedge \gamma_X^2$. Plug in this optimal $\gamma$ into the LPGW problem (14) (or (15)), we obtain

$LPGW(\mathbb{Y}^1, \mathbb{Y}^2; \mathbb{X}, \gamma^1, \gamma^2)$

$= \langle |g_{Y^1} - g_{Y^2}|^2, (\gamma_X^1 \wedge \gamma_X^2)^{\otimes 2}\rangle + \lambda(|\nu^1|^2 + |\nu^2|^2 - 2|\gamma_X^1 \wedge \gamma_X^2|^2)$

$= \langle |k_{\gamma^1} - k_{\gamma^2}|^2, (\gamma_X^1 \wedge \gamma_X^2)^{\otimes 2}\rangle + \lambda(|\nu^1|^2 + |\nu^2|^2 - 2|\gamma_X^1 \wedge \gamma_X^2|^2)$

and we complete the proof.

### F.4 PROOF OF PROPOSITION 3.1: PART (4)

First, under the Monge mapping assumption, an optimal PGW plan is of the form $\gamma^* = (\mathrm{id} \times T)_{\#}\gamma_X^*$ for some $\gamma_X^* \le \mu$, we have

$\|k_{\gamma^*}\|_{L^2((\gamma_X^*)^{\otimes 2})}^2 + \lambda(|\mu|^2 - |\gamma_X^*|^2 + |\gamma_c|)$

$= \int_{X^{\otimes 2}} |g_X(x, x') - g_Y(T(x), T(x'))|^2 d(\gamma^*)^{\otimes 2} + \lambda(|\mu^{\otimes 2} - (\gamma_X^*)^{\otimes 2}| + |\nu^{\otimes 2} - (\gamma_Y^*)^{\otimes 2}|)$

$= PGW_\lambda(\mathbb{X}, \mathbb{Y}).$

In the general case, we first introduce the following lemma.

**Lemma F.3.** *If $\gamma^0 \in \Gamma_{\le,\lambda}^*(\mathbb{X}, \mathbb{X})$, then $(\pi_X)_{\#}\gamma^0 = \mu, (\pi_Y)_{\#}\gamma^0 = \mu$.*

*Proof of Lemma F.3.* Pick any $\gamma \in \Gamma_{\le}(\mathbb{X}, \mathbb{X})$, suppose $|\gamma| < |\mu|$, we have

$$\begin{aligned} C(\gamma; \mathbb{X}, \mathbb{X}, \lambda) &= \int_{X^{\otimes 2}} |g_X(x, x') - g_X(x^0, x^{0\prime})| d\gamma^{\otimes 2} + \lambda(2|\mu|^2 - 2|\gamma|^2) \\ &\ge \lambda(2|\mu|^2 - 2|\gamma|^2) \\ &> 0. \end{aligned} \tag{68}$$

However, $PGW_\lambda(\mathbb{X}, \mathbb{X}) = 0$. Thus, $\gamma$ is not optimal. □

By the above lemma, we have $|\gamma^*| \leq |\gamma^0|$. For each $\gamma \in \Gamma_{\leq}(\gamma^0, \gamma^*; \mu)$, from (Bai et al., 2024), there exists $\gamma' \in \Gamma_{\leq}(\gamma^0, \gamma^*)$ such that $(\pi_{X,Y})_\#\gamma' = \gamma^*$ and $\gamma \leq \gamma'$. Thus we have $|\gamma'| = |\gamma^*|$.

Let

$$C(\gamma; \mathbb{X}, \mathbb{X}, \mathbb{Y}, \lambda) := \int_{(X \times X \times Y)^{\otimes 2}} |g_X(x, x') - g_Y(y, y')|^2 d\gamma^{\otimes 2}((x, x'), (x, x'), (y, y'))$$
$$+ \lambda(|\mu|^2 + |\nu|^2 - 2|\gamma|^2)$$

and similarly we define $C(\gamma'; \mathbb{X}, \mathbb{X}, \mathbb{Y}, \lambda)$. Then

$$C(\gamma'; \mathbb{X}, \mathbb{X}, \mathbb{Y}, \lambda) - C(\gamma; \mathbb{X}, \mathbb{X}, \mathbb{Y}, \lambda)$$
$$= \int_{(X \times X \times Y)^{\otimes 2}} \left[ |g_X(x, x') - g_Y(y, y')|^2 - 2\lambda \right] d(\gamma'^{\otimes 2} - \gamma^{\otimes 2})((x, x'), (x, x'), (y, y'))$$
$$\leq 0.$$

In addition,

$$C(\gamma'; \mathbb{X}, \mathbb{X}, \mathbb{Y}, \lambda)$$
$$= \int_{(X \times X \times Y)^{\otimes 2}} |g_X(x, x') - g_Y(y, y')|^2 d\gamma'^{\otimes 2}((x, x'), (x, x'), (y, y')) + \lambda(|\mu|^2 + |\nu|^2 - 2|\gamma'|^2)$$
$$= \int_{(X \times Y)^{\otimes 2}} |g_X(x, x') - g_Y(y, y')|^2 d(\gamma^*)^{\otimes 2}((x, x'), (y, y')) + \lambda(|\mu|^2 + |\nu|^2 - 2|\gamma^*|^2)$$
$$= PGW_\lambda(\mathbb{X}, \mathbb{Y})$$

where the second equality holds by (49) and the fact $|\gamma'| = |\gamma^*|$. Thus, we have

$$LPGW_\lambda(\mathbb{X}, \mathbb{Y}; \mathbb{X}, \gamma^0, \gamma^*) \leq C(\gamma'; \mathbb{X}, \mathbb{X}, \mathbb{Y}, \lambda) = PGW_\lambda(\mathbb{X}, \mathbb{Y}).$$

Another direction is trivial since for each $\gamma \in \Gamma_{\leq}(\gamma^0, \gamma^*; \mu)$, $(\pi_{2,3})_\#\gamma := (\pi_{X,Y})_\#\gamma \in \Gamma_{\leq}(\mu, \nu)$. Thus

$$PGW_\lambda(\mathbb{X}, \mathbb{Y}; \gamma) \leq C(\gamma; \mathbb{X}, \mathbb{X}, \mathbb{Y}, \lambda) \leq LPGW_\lambda(\mathbb{X}, \mathbb{Y}; \mathbb{X}, \gamma^0, \gamma^*).$$

and we have completed the proof.

### F.5  PROOF OF PROPOSITION 3.1: PART (5)

This section discusses the proof of statement (5) of Proposition 3.1. That is, we discuss the metric properties of the proposed LPGW distance.

#### F.5.1  FORMAL STATEMENT

In this section, we set $g_Y = d_Y^s$. Let

$$\mathcal{G} := \{\mathbb{Y} = (Y, g_Y, \nu)\}$$

where $Y \subset \mathbb{R}^d$ for some $d \in \mathbb{N}$ such that $Y$ is nonempty convex compact set; $\nu \in \mathcal{M}_+(Y)$ with $\nu^{\otimes 2}(|g_Y|) < \infty$.

Then $\forall \mathbb{Y}^1, \mathbb{Y}^2 \in \mathcal{G}$, we define $\mathbb{Y}^1 \sim \mathbb{Y}^2$ iff $GW(\mathbb{Y}^1, \mathbb{Y}^2) = 0$. By (Mémoli, 2011), in this case, optimal transportation plan $GW(\mathbb{Y}^1, \mathbb{Y}^2)$ is induced by the Monge mapping $T$. Furthermore, $T$ is bijection $\nu^1$-a.s. and is called "isomorphism" since

$$d_{Y^1}(y^1, y^{1'}) = d_{Y^2}(T(y^1), T(y^{1'})), \nu^1 - a.s.$$

Next, given $\mathcal{G}' \subset \mathcal{G}$ we introduce the following assumptions:

**Assumption F.4.** For each $\mathbb{Y} \in \mathcal{G}'$, the problem $PGW_\lambda(\mathbb{X}, \mathbb{Y})$ admits a unique solution.

**Assumption F.5.** For subset $\mathcal{G}'$, define $K_1, K_2$ by

$$K_1 = \sup\{|\nu|, \mathbb{Y} = (Y, g_Y, \nu) \in \mathcal{G}'\},$$
$$K_2 = \sup\{\max_{y,y' \in \text{supp}(\nu)} |g_Y(y,y')|^2 : \mathbb{Y} = (X, g_Y, \nu) \in \mathcal{G}')\}$$

We suppose $K_1, K_2 < \infty$.

**Assumption F.6.** $g_X, g_{Y^1}, g_{Y^2}$ are induced by metrics. In particular, $g_X(\cdot, \cdot) = d_X^q(\cdot, \cdot)$, $g_{Y^1}(\cdot, \cdot) = d_{Y^1}^q(\cdot, \cdot)$, $g_{Y^2}(\cdot, \cdot) = d_{Y^2}^q(\cdot, \cdot)$, where $d_X$ is a metric defined in $X$, $q \geq 1$, similar to $d_{Y^1}, d_{Y^2}$.

*Remark* F.7. The uniqueness of optimal transportation plan assumption (F.4) is also introduced by (Nenna & Pass, 2023) to prove the triangle inequality of linear optimal transport. The assumption (F.5) discusses the bounded total mass and transportation cost.

**Proposition F.8** (Metric property of LPGW). *Fix $\mathbb{X} \in \mathcal{G}$ and for each $\mathbb{Y}^1$, we select $\gamma^1 \in \Gamma^*_{\leq,\lambda}(\mathbb{X}, \mathbb{Y}^1), \gamma^2 \in \Gamma^*_{\leq,\lambda}(\mathbb{X}, \mathbb{Y}^2)$. In addition, set $g_X = d_X^s, g_Y = d_Y^s$ where $s \geq 1$. Then we have:*

(1) *The LPGW distances (15),(16) are nonnegative:*

$$LPGW(\mathbb{Y}^1, \mathbb{Y}^2; \mathbb{X}, \gamma^1, \gamma^2, \lambda), LPGW_\lambda(\mathbb{Y}^1, \mathbb{Y}^2; \mathbb{X}) \geq 0 \qquad (69)$$

(2) *The LPGW distances (15),(16) are symmetric:*

$$LPGW(\mathbb{Y}^1, \mathbb{Y}^2; \mathbb{X}, \gamma^1, \gamma^2, \lambda) = LPGW(\mathbb{Y}^2, \mathbb{Y}^1 \mathbb{X}, \gamma^2, \gamma^1, \lambda)$$
$$LPGW(\mathbb{Y}^2, \mathbb{Y}^1; \mathbb{X}, \lambda) = LPGW_\lambda(\mathbb{Y}^1, \mathbb{Y}^2; \mathbb{X}). \qquad (70)$$

(3) *The LPGW distances (15) satisfy triangle inequality.*

*In addition, under assumption (F.4), (16) satisfies the triangle inequality.*

(4) *Under assumption (F.6), if $LPGW_\lambda(\mathbb{Y}^1, \mathbb{Y}^2; \mathbb{X}, \gamma^1, \gamma^2) = 0$, we have $\mathbb{Y}^1 \sim \mathbb{Y}^2$. Similarly to $LPGW_\lambda(\mathbb{Y}^1, \mathbb{Y}^2; \mathbb{X})$.*

(5) *Under assumption (F.5), suppose $|\mu| \geq K_1$, and $2\lambda \geq K_2 + \sup_{x,x' \in supp(\mu)} |g_X(x, x')|^2$, and $\mathbb{Y}^1 \sim \mathbb{Y}^2$, where $K_1, K_2$ are defined in (F.5). Then there exists $\gamma^1, \gamma^2$ such that*

$$LPGW(\mathbb{Y}^1, \mathbb{Y}^2; \mathbb{X}, \gamma^1, \gamma^2, \lambda) = LPGW_\lambda(\mathbb{Y}^1, \mathbb{Y}^2; \mathbb{X}) = 0.$$

*Therefore, under the assumptions (F.4), (F.5), and (F.6), we have that (16) defines a metric in $\mathcal{G}'/\sim$.*

Note, the above statement implies the metric property of LGW by setting $\lambda \to \infty$. We refer to the next section for details.

### F.5.2 Proof of Proposition F.8: Parts (1), (2), (4), (5)

*Proof.* Choose $\mathbb{Y}^1, \mathbb{Y}^2, \mathbb{Y}^3 \in \mathcal{G}$.

(1), (2) It is straightforward to verify the nonnegativity (1) and symmetry (2).

(4) In this setting, we have:

$$0 = LPGW_\lambda(\mathbb{Y}^1, \mathbb{Y}^2; \mathbb{X}, \gamma^1, \gamma^2)$$
$$= \inf_{\gamma \in \Gamma_{\leq}(\gamma^1, \gamma^2; \mu)} \int_{(X \times Y^1 \times Y^2)^{\otimes 2}} |g_{Y^1}(y^1, y^{1\prime}) - g_{Y^2}(y^2, y^{2\prime})|^2 d\gamma^{\otimes 2} + \lambda(|\nu^1|^2 + |\nu^2| - 2|\gamma|^2)$$

Pick one corresponding minimizer $\gamma$, we obtain

$$\int_{(X \times Y^1 \times Y^2)^{\otimes 2}} |g_{Y^1}(y^1, y^{1\prime}) - g_{Y^2}(y^2, y^{2\prime})|^2 d\gamma^{\otimes 2} = 0,$$

and
$$|\nu^1|^2 + |\nu^2|^2 - 2|\gamma|^2 = 0.$$
Thus $\gamma \in \Gamma(\nu^1, \nu^2)$ and therefore
$$GW(\mathbb{Y}^1, \mathbb{Y}^2) = 0.$$
That is $\mathbb{Y}^1 \sim \mathbb{Y}^2$.

By a similar process, if $LPGW_\lambda(\mathbb{Y}^1, \mathbb{Y}^2; \mathbb{X}) = 0$, we have $\mathbb{Y}^1 \sim \mathbb{Y}^2$.

(5) Since $\mathbb{Y}^1 \sim \mathbb{Y}^2$, from triangle inequality of $PGW_\lambda(\cdot, \cdot)$, we have
$$PGW_\lambda(\mathbb{X}, \mathbb{Y}^1) = PGW_\lambda(\mathbb{X}, \mathbb{Y}^2).$$
In addition, there exists a bijection mapping $T : \operatorname{supp}(\nu^1) \to \operatorname{supp}(\nu^2)$ such that $g_{Y^1}(x_1, x_1') = g_{X_2}(T(x_1), T(x_1')), \nu^1-$a.s. such that $T_\# \nu^1 = \nu^2$.

Pick $\gamma^1 \in \Gamma^*_{\leq, \lambda}(\mathbb{X}, \mathbb{Y}^1)$. Let $\gamma^2 := (\mathrm{id} \times T)_\# \gamma^1$, that is, for each test function $\phi \in C_0(X \times Y^2)$,
$$\gamma^2(\phi) := \int_{X \times Y^1} \phi(x, T(y^1)) d\gamma^1(x, y^1).$$
We claim $\gamma^2$ is optimal for $PGW_\lambda(\mathbb{X}^1, \mathbb{Y}^2)$.
$$(\pi_1)_\# \gamma^2 = (\pi_1)_\# \gamma^1 \leq \mu,$$
$$(\pi_2)_\# \gamma^2 = T_\#((\pi_2)_\# \gamma^1) \leq \nu^2 \qquad \text{since } (\pi_Y)_\# \gamma^1 \leq \nu^1.$$
Thus, $\gamma^2 \in \Gamma_\leq(\mu, \nu^2)$. In addition,

$C(\gamma^2; \mathbb{X}, \mathbb{Y}^2)$
$$= \int_{(X \times Y^1)^{\otimes 2}} |g_X(x, x') - g_{Y^2}(T(y^1), T(y^1)')|^2 d(\gamma^1)^{\otimes 2} + \lambda(|\mu|^2 + |\nu^2|^2 - 2|\gamma^2|^2)$$
$$= \int_{(X \times Y^1)^{\otimes 2}} |g_X(x, x') - g_{Y^2}(T(y^1), T(y^1)')|^2 d(\gamma^1)^{\otimes 2}(x, x'|y^1, y^{1'}) d(\nu^1)^{\otimes 2}(y^1, y^{1'})$$
$$\quad + \lambda(|\mu|^2 + |\nu^1| - 2|\gamma^1|^1)$$
$$= \int_{(X \times Y^1)^{\otimes 2}} |g_X(x, x') - g_{Y^1}(y^1, y^{1'})|^2 d(\gamma^1)^{\otimes 2}(x, x'|y^1, y^{1'}) d(\nu^1)^{\otimes 2}(y^1, y^{1'})$$
$$\quad + \lambda(|\mu|^2 - |\gamma^1|^2) \tag{71}$$
$$= PGW_\lambda(\mathbb{X}, \mathbb{Y}^1) = PGW_\lambda(\mathbb{X}, \mathbb{Y}^2).$$
and thus $\gamma^2$ is optimal in $PGW_\lambda(\mathbb{X}, \mathbb{Y}^2)$.

Since $2\lambda$ is sufficiently large, by lemma E.2 in (Bai et al., 2024), there exists optimal $\gamma^1$ such that:
$$|\gamma^1| = \min(|\mu|, |\nu^1|) = |\nu^1|.$$
Thus we have $|\gamma^2| = |\gamma^1| = |\nu^1| = |\nu^2|$. Plug $\gamma^1, \gamma^2$ into (15), we have:
$$LPGW(\mathbb{Y}^1, \mathbb{Y}^2; \mathbb{X}, \gamma^1, \gamma^2)$$
$$= \int g_{Y^1}(y^1, y^{1'}) - g_{Y^2}(T(y^1), T(y^{1'})) d(\nu^1)^{\otimes 2}(y^1, y^{1'})$$
$$= 0. \tag{72}$$
$\square$

### F.5.3 Proof of Proposition F.8: Part (3)

**Notation Setup and Summary** Choose $\gamma^1 \in \Gamma^*_{\leq, \lambda}(\mathbb{X}, \mathbb{Y}^1), \gamma^2 \in \Gamma^*_{\leq, \lambda}(\mathbb{X}, \mathbb{Y}^2), \gamma^3 \in \Gamma^*_{\leq, \lambda}(\mathbb{X}, \mathbb{Y}^3)$, the goal is to show triangle inequality in this section:
$$LPGW_\lambda(\mathbb{Y}^1, \mathbb{Y}^2; \mathbb{X}, \gamma^1, \gamma^2) \leq LPGW_\lambda(\mathbb{Y}^1, \mathbb{Y}^3; \mathbb{X}, \gamma^1, \gamma^3) + LPGW_\lambda(\mathbb{Y}^2, \mathbb{Y}^3; \mathbb{X}, \gamma^2, \gamma^3). \tag{73}$$

**Related Lemmas**

**Lemma F.9** (Varient Gluing lemma). *Suppose $\gamma^1 \in \Gamma(\mu, \nu^1), \gamma^2 \in \Gamma(\mu, \nu^2), \gamma^3 \in \Gamma(\mu, \nu^3)$ are probability measures. Pick $\gamma^{1,2} \in \Gamma(\gamma^1, \gamma^2; \mu), \gamma^{1,3} \in \Gamma(\gamma^1, \gamma^3; \mu), \gamma^{2,3'} \in \Gamma(\gamma^1, \gamma^{3'}; \mu)$, there exists $\gamma \in \mathcal{P}(X \times Y^1 \times Y^2 \times Y^3)$ such that*

$$\begin{cases} (\pi_{X,Y^1,Y^3})_{\#}\gamma = \gamma^{1,3}, \\ (\pi_{X,Y^2,Y^3})_{\#}\gamma = \gamma^{2,3}. \end{cases} \tag{74}$$

Note, if we set $X = \{0\}$ and $\mu = \delta_0$, the above lemma becomes the classical Gluing lemma (see e.g. Lemma 5.5 (Santambrogio, 2015)). For this, we call it the **variant gluing lemma**.

*Proof of Lemma F.9.* For convenience, let $X', Y^{3'}$ be independent copy of $X, Y^3$. By gluing lemma, there exists $\gamma' \in \mathcal{P}((X \times Y^1 \times Y^2) \times X')$ such that

$$\begin{cases} (\pi_{X,Y^1,Y^2})_{\#}\gamma = \gamma^{1,2}, \\ (\pi_{X,X'})_{\#}\gamma = (\mathrm{id} \times \mathrm{id})_{\#}\mu. \end{cases} \tag{75}$$

Apply gluing lemma again between $\gamma'$ and $\gamma^{1,3}$, we can find $\gamma'' \in \mathcal{P}((X \times Y^1 \times Y^3) \times (X' \times Y^2 \times Y^{3'}))$ such that

$$\begin{cases} (\pi_{X,Y^1,Y^3,X'})_{\#}\gamma = \gamma', \\ (\pi_{X',Y^2,Y^3})_{\#}\gamma = \gamma^{2,3}. \end{cases}$$

Applying $\gamma = (\pi_{X,Y^1,Y^2,Y^3})_{\#}\gamma''$ we complete the proof.

Alternatively, we can set

$$\gamma = \gamma^{1,3}_{Y^1,Y^3|X}\gamma^{2,3}_{Y^2,Y^3|X}\mu(x)$$

and it is straightforward to verify $\gamma$ satisfies (74). $\qquad\square$

**Lemma F.10** (Triangle inequality for LGW). *Suppose $\mathbb{X}, \mathbb{Y}^1, \mathbb{Y}^2, \mathbb{Y}^3$ are under the balanced GW setting, choose $\gamma^1 \in \Gamma^*(\mu, \nu^1), \gamma^2 \in \Gamma^*(\mu, \nu^2), \gamma^3 \in \Gamma^*(\mu, \nu^3)$, we have*

$$LGW(\mathbb{Y}^1, \mathbb{Y}^2; \mathbb{X}, \gamma^1, \gamma^2) \leq LGW(\mathbb{Y}^1, \mathbb{Y}^3; \mathbb{X}, \gamma^1, \gamma^3) + LGW(\mathbb{Y}^2, \mathbb{Y}^3; \mathbb{X}, \gamma^2, \gamma^3) \tag{76}$$

*Proof of lemma F.10.* Pick the $\gamma$ from lemma F.9, we have

$$(\pi_{X,Y^1})_{\#}\gamma_{X,Y^1} = (\pi_{X,Y^1})_{\#}\gamma^{1,3} = \gamma^1,$$
$$(\pi_{X,Y^2})_{\#}\gamma_{X,Y^2} = (\pi_{X,Y^2})_{\#}\gamma^{2,3} = \gamma^2.$$

Thus $(\pi_{X,Y^1,Y^2})_{\#}\gamma \in \Gamma(\gamma^1, \gamma^2; \mu)$. Then we have

$LGW(\mathbb{Y}^1, \mathbb{Y}^2; \mathbb{X}, \gamma^1, \gamma^2)$

$= \left\langle |g_{Y^1} - g_{Y^2}|^2, (\gamma^{1,2})^{\otimes 2} \right\rangle^{1/2}$

$\leq \left\langle |g_{Y^1} - g_{Y^2}|^2, ((\pi_{X,Y^1,Y^2})_{\#}\gamma)^{\otimes 2} \right\rangle^{1/2} \tag{77}$

$= \left( \int_{X \times Y \times Y^2 \times Y^3} |g_{Y^1}(y^1, y^{1'}) - g_{Y^2}(y^2, y^{2'})|^2 d\gamma^{\otimes 2}((x,x'), (y^1, y^{1'}), (y^2, y^{2'}), (y^3, y^{3'})) \right)^{1/2}$

$\leq \left( \int_{X \times Y \times Y^2 \times Y^3} \left( |g_{Y^1}(y^1, y^{1'}) - g_{Y^3}(y^3, y^{3'})| + |g_{Y^2}(y^2, y^{2'}) - g_{Y^3}(y^3, y^{3'})| \right)^2 d\gamma^{\otimes 2} \right)^{1/2}$

$\leq \left( \int_{X \times Y \times Y^2 \times Y^3} \left( |g_{Y^1}(y^1, y^{1'}) - g_{Y^3}(y^3, y^{3'})| \right)^2 d\gamma^{\otimes 2} \right)^{1/2}$

$\quad + \left( \int_{X \times Y \times Y^2 \times Y^3} \left( |g_{Y^2}(y^2, y^{2'}) - g_{Y^3}(y^3, y^{3'})| \right)^2 d\gamma^{\otimes 2} \right)^{1/2} \tag{78}$

$= LGW(\mathbb{Y}^1, \mathbb{Y}^3; \mathbb{X}, \gamma^1, \gamma^3) + LGW(\mathbb{Y}^2, \mathbb{Y}^3; \mathbb{X}, \gamma^2, \gamma^3)$

where (77) follows from the fact $\gamma^{1,2}$ is optimal in $\Gamma(\gamma^1, \gamma^2; \mu)$; (78) holds by the Minkowski inequality. In the uniqueness assumption (F.4), we have

$$LGW(\mathbb{Y}^1, \mathbb{Y}^2; \mathbb{X}) \leq LGW(\mathbb{Y}^1, \mathbb{Y}^3; \mathbb{X}) + LGW(\mathbb{Y}^2, \mathbb{Y}^3; \mathbb{X}).$$

$\square$

**Convert LPGW to LGW**   The main idea of this step is similar to the proof of Proposition 3.3 in (Bai et al., 2024).

**Notations Setup**

We can redefine $\Gamma_{\leq}(\gamma^1, \gamma^2; \mu)$ as

$$\Gamma_{\leq}(\gamma^1, \gamma^2; \mu) = \big\{ \gamma \in \mathcal{M}_+(X \times Y^1 \times X \times Y^2) : (\pi_{1,2})_\# \gamma \leq \gamma^1, (\pi_{3,4})_\# \gamma \leq \gamma^2,$$
$$(\pi_{1,3})_\# \gamma = (\mathrm{id} \times \mathrm{id})_\# \mu', \mu' \leq \mu \big\}. \tag{79}$$

We define auxiliary points $\hat{\infty}_0, \hat{\infty}_1, \hat{\infty}_2, \hat{\infty}_3$ and then define

$$\hat{X} = X \cup \{\hat{\infty}_0\},$$
$$\hat{Y}^1 = Y^1 \cup \{\hat{\infty}_1, \hat{\infty}_2, \hat{\infty}_3\},$$
$$\hat{Y}^2 = Y^2 \cup \{\hat{\infty}_1, \hat{\infty}_2, \hat{\infty}_3\},$$
$$\hat{Y}^3 = Y^3 \cup \{\hat{\infty}_1, \hat{\infty}_2, \hat{\infty}_3\}.$$

and

$$\hat{\gamma}^1 = \gamma^1 + |\gamma^2| \delta_{\hat{\infty}_0, \hat{\infty}_2} + |\gamma^3| \delta_{\hat{\infty}_0, \hat{\infty}_3} \in \mathcal{M}_+(\hat{X} \times \hat{Y}^1),$$
$$\hat{\gamma}^2 = \gamma^2 + |\gamma^1| \delta_{\hat{\infty}_0, \hat{\infty}_1} + |\gamma^3| \delta_{\hat{\infty}_0, \hat{\infty}_3} \in \mathcal{M}_+(\hat{X} \times \hat{Y}^2),$$
$$\hat{\gamma}^3 = \gamma^3 + |\gamma^1| \delta_{\hat{\infty}_0, \hat{\infty}_1} + |\gamma^2| \delta_{\hat{\infty}_0, \hat{\infty}_2} \in \mathcal{M}_+(\hat{X} \times \hat{Y}^3).$$

Define

$$g_{\hat{Y}^1}(y^1, y^{1\prime}) = \begin{cases} g_{Y^1}(y^1, y^{1\prime}) & \text{if } y^1, y^1 \in Y^1, \\ \infty & \text{elsewhere} \end{cases}$$

and $g_{\hat{Y}^2}, g_{\hat{Y}^3}$ are defined similarly.

Finally, define

$$D_\lambda(r_1, r_2) = \begin{cases} |r_1 - r_2| & \text{if } r_1, r_2 \in \mathbb{R}, \\ \sqrt{\lambda} & \text{elsewhere.} \end{cases}.$$

Let $\hat{X}' = X' \cup \{\hat{\infty}_0\}$ to be a copy of set $\hat{X}$. Then, we can set

$$\hat{\Gamma}(\hat{\gamma}^1, \hat{\gamma}^2; \mu) := \big\{ \hat{\gamma} \in \mathcal{M}_+(\hat{X} \times \hat{Y}^1 \times \hat{X} \times \hat{Y}^2), (\pi_{1,2})_\# \hat{\gamma} = \hat{\gamma}^1, (\pi_{3,4})_\# \hat{\gamma} = \hat{\gamma}^2, \tag{80}$$
$$(\pi_{1,3})_\# \hat{\gamma}\,|_{X^{\otimes 2}} = (\mathrm{id} \times \mathrm{id})_\# \mu', \mu' \leq \mu \big\}. \tag{81}$$

Consider the mapping:

$$\mathcal{F} : \Gamma_{\leq}(\gamma^1, \gamma^2; \mu) \mapsto \hat{\Gamma}(\hat{\gamma}^1, \hat{\gamma}^2; \mu)$$
$$\gamma \mapsto \hat{\gamma} := \gamma + (\gamma^1 - (\pi_{1,2})_\# \gamma) \otimes \delta_{(\hat{\infty}_0, \hat{\infty}_1)} + \delta_{(\hat{\infty}_0, \hat{\infty}_2)} \otimes (\gamma^2 - (\pi_{3,4})_\# \gamma) + |\gamma| \delta_{(\hat{\infty}_0, \hat{\infty}_2), (\infty_0, \hat{\infty}_1)}$$
$$+ |\gamma^3| \delta_{(\hat{\infty}_0, \hat{\infty}_3), (\hat{\infty}_0, \hat{\infty}_3)}. \tag{82}$$

By the following lemma, we show that $\mathcal{F}$ defines an equivalent relation between the two sets.

**Lemma F.11.** *The mapping $\mathcal{F}$ defined in (82) is well-defined. In addition, it is a bijection if we set the identity $\hat{\infty}_0 = \hat{\infty}_1 = \hat{\infty}_2 = \hat{\infty}_3$.*

*Proof of Lemma F.11.* By (Bai et al., 2022, Proposition B.1) we have for each $\gamma \in \Gamma(\gamma^1, \gamma^2; \mu)$,

$$(\pi_{1,2})_\# \hat{\gamma} = \hat{\gamma}^1, (\pi_{3,4})_\# \hat{\gamma} = \hat{\gamma}^2.$$

In addition,

$$(\pi_{1,3})_\# \hat{\gamma} = (\pi_{1,3})_\# \bar{\gamma} + ((\pi_1)_\# \gamma^1 - (\pi_1)_\# \gamma) \otimes \delta_{\hat{\infty}_0} + \delta_{\hat{\infty}_0} \otimes ((\pi_1)_\# \gamma^2 - (\pi_3)_\# \gamma) + (|\bar{\gamma}| + |\gamma^3|) \delta_{\hat{\infty}_0, \hat{\infty}_0}.$$

Thus,

$$(\pi_{1,3})_\# \hat{\gamma} \mid_{X^{\otimes 2}} = (\pi_{1,3})_\# \gamma = (\mathrm{id} \times \mathrm{id})_\# (\pi_1)_\# \gamma,$$

where $(\pi_1)_\# \gamma \leq (\pi_1)_\# \gamma^1 \leq \mu$, and we have $\mathcal{F}$ is well-defined.

It remains to show $\mathcal{F}$ is a bijection. For each $\hat{\gamma} \in \hat{\Gamma}_{\leq}(\hat{\gamma}^1, \hat{\gamma}^2)$ and we define mapping

$$\mathcal{F}^{-1} : \widehat{\Gamma}(\hat{\gamma}^1, \hat{\gamma}^2; \mu) \to \Gamma_{\leq}(\gamma^1, \gamma^2; \mu) \tag{83}$$
$$\hat{\gamma} \mapsto \hat{\gamma} \mid_{X \times Y^1 \times X \times Y^2}$$

It remains to show $\mathcal{F}^{-1}$ is inverse of $\mathcal{F}$.

First, we claim $\mathcal{F}^{-1}$ is well-defined. Pick $\hat{\gamma} \in \hat{\Gamma}(\hat{\gamma}^1, \hat{\gamma}^2)$, by Lemma (Bai et al., 2022, Lemma B.1.), we have $(\pi_{X,Y^1})_\# \hat{\gamma} \leq \gamma^1, (\pi_{3,4})_\# \hat{\gamma} \leq \gamma^2$. In addition,

$$(\pi_{1,3})_\# \gamma = (\pi_{1,3})_\# \hat{\gamma} \mid_{X^{\otimes 2}} = (\mathrm{id} \times \mathrm{id})_\# \mu',$$

where $\mu' \leq \mu$. Thus, $\mathcal{F}^{-1}(\hat{\gamma}) \in \Gamma_{\leq}(\gamma^1, \gamma^2; \mu)$.

In addition, it is straightforward to verify $\mathcal{F}^{-1}(\mathcal{F}(\gamma)) = \gamma, \forall \bar{\gamma} \in \Gamma_{\leq}(\gamma^1, \gamma^2; \mu)$, and then we complete the proof. $\square$

It is straightforward to verify:

$$\langle D_\lambda^2(g_{\hat{Y}^1}, g_{\hat{Y}^2}, \hat{\gamma}^{\otimes 2}) \rangle = \langle |g_{Y^1} - g_{Y^2}|^2, \gamma^{\otimes 2} \rangle + \lambda(|\gamma^1|^2 + |\gamma^2|^2 - |\gamma|^2)$$

Combine it with the above lemma, take the infimum over $\Gamma_{\leq}(\gamma^1, \gamma^2; \mu)$ (equivalently, over $\hat{\Gamma}(\hat{\gamma}^1, \hat{\gamma}^2; \mu)$), we obtain:

$$\inf_{\hat{\gamma} \in \widehat{\Gamma}(\hat{\gamma}^1, \hat{\gamma}^2; \mu)} \langle D_\lambda^2(g_{\hat{Y}^1}, g_{\hat{Y}^2}, \hat{\gamma}^{\otimes 2}) \rangle = \inf_{\gamma \in \Gamma_{\leq}(\gamma^1, \gamma^2; \mu)} \langle |g_{Y^1} - g_{Y^2}|^2, \gamma^{\otimes 2} \rangle + \lambda(|\gamma^1|^2 + |\gamma^2|^2 - |\gamma|^2)$$
$$= LPGW(\mathbb{Y}^1, \mathbb{Y}^2; \mathbb{X}, \gamma^1, \gamma^2)$$

where the second equality holds from the fact $|\nu^1| = |\nu^2| = |\nu^3| = |\gamma^1| = |\gamma^2| = |\gamma^3|$. Similar identity holds for $LPGW(\mathbb{Y}^1, \mathbb{Y}^3; \mathbb{X}, \gamma^1, \gamma^3), LPGW(\mathbb{Y}^2, \mathbb{Y}^3; \mathbb{X}, \gamma^2, \gamma^3)$.

**Verifying Inequality**   It remains to show

$$\inf_{\hat{\gamma} \in \widehat{\Gamma}(\hat{\gamma}^1, \hat{\gamma}^2; \mu)} \langle D_\lambda^2(g_{\hat{Y}^1}, g_{\hat{Y}^2}, \hat{\gamma}^{\otimes 2}) \rangle^{1/2} \leq \inf_{\hat{\gamma} \in \widehat{\Gamma}(\hat{\gamma}^1, \hat{\gamma}^3; \mu)} \langle D_\lambda^2(g_{\hat{Y}^1}, g_{\hat{Y}^3}, \hat{\gamma}^{\otimes 2}) \rangle^{1/2} + \inf_{\hat{\gamma} \in \widehat{\Gamma}(\hat{\gamma}^2, \hat{\gamma}^3; \mu)} \langle D_\lambda^2(g_{\hat{Y}^2}, g_{\hat{Y}^3}, \hat{\gamma}^{\otimes 2}) \rangle^{1/2}.$$

Pick $\gamma^{1,2} \in \Gamma_{\leq}(\gamma^1, \gamma^2; \mu)$ that is optimal for $\langle D_\lambda^2(g_{\hat{Y}^1}, g_{\hat{Y}^2}, \hat{\gamma}^{\otimes 2}) \rangle^{1/2}$. Similalry, pick optimal $\gamma^{1,3} \in \Gamma_{\leq}(\gamma^1, \gamma^3), \gamma^{2,3} \in \Gamma_{\leq}(\gamma^2, \gamma^3)$, we construct the corresponding optimal $\hat{\gamma}^{1,3} \in \Gamma(\hat{\gamma}^1, \hat{\gamma}^3; \mu), \bar{\hat{\gamma}}^{2,3} \in \Gamma(\hat{\gamma}^2, \hat{\gamma}^3; \mu)$.

Thus, by Gluing lemma, there exists $\gamma \in \mathcal{M}_+((X \times Y^1) \times (X \times Y^2) \times (X \times Y^3)))$ such that

$$(\pi_{(1,2),(5,6)})_\# \gamma = \hat{\gamma}^{1,3},$$
$$(\pi_{(3,4),(5,6)})_\# \gamma = \hat{\gamma}^{2,3}.$$

Now by the definition of $\hat{\Gamma}(\hat{\gamma}^1, \hat{\gamma}^3; \mu), \hat{\Gamma}(\hat{\gamma}^2, \hat{\gamma}^3; \mu)$, we have

$$(\pi_{1,5})_\# \gamma = (\mathrm{id} \times \mathrm{id})_\# \mu', \tag{84}$$
$$(\pi_{3,5})_\# \gamma = (\mathrm{id} \times \mathrm{id})_\# \mu'' \tag{85}$$

for some radon measures $\mu', \mu'' \leq \mu$. Thus, we have $\mu' = \mu'' \leq \mu$. Therefore,

$$(\pi_{1,3})_{\#}\gamma = (\mathrm{id} \times \mathrm{id})_{\#}\mu'.$$

In addition,

$$(\pi_{1,2})_{\#}\gamma = (\pi_{1,2})_{\#}\hat{\gamma}^{1,3} = \hat{\gamma}^1,$$
$$(\pi_{3,4})_{\#}\gamma = (\pi_{1,2})_{\#}\hat{\gamma}^{2,3} = \hat{\gamma}^2.$$

So $(\pi_{1,2,3,4})_{\#}\gamma \in \Gamma(\hat{\gamma}^1, \hat{\gamma}^2; \mu)$.

Therefore, we have

$$\langle D_\lambda^2(k_{Y^1}, k_{Y^2}), (\hat{\gamma}^{1,2})^{\otimes 2}\rangle^{1/2}$$
$$\leq \langle D_\lambda^2(k_{Y^1}, k_{Y^2}), \gamma^{\otimes 2}\rangle^{1/2}$$
$$\leq \langle D_\lambda^2(k_{Y^1}, k_{Y^3}), \gamma^{\otimes 2}\rangle^{1/2} + \langle D_\lambda^2(k_{Y^2}, k_{Y^3}), \gamma^{\otimes 2}\rangle^{1/2}$$

where the second inequality holds from (Bai et al., 2024, Eq.(57)).

## G  RELATION BETWEEN LPGW AND LGW

**Theorem G.1.** *Suppose $|\mu| = |\nu^1| = |\nu^2| = 1$ and $x, x$ is bounded, $g_{Y^1}, g_{Y^2}$ are continuous function.*

*Choose sequence $\lambda_n \to \infty$, if $n$ is sufficiently large, we have $\lambda_n$, then*

$$LPGW_{\lambda_n}(\mathbb{Y}^1, \mathbb{Y}^2; \mathbb{X}) \to LGW_{\lambda_n}(\mathbb{Y}^1, \mathbb{Y}^2; \mathbb{X}),$$
$$aLPGW_{\lambda_n}(\mathbb{Y}^1, \mathbb{Y}^2; \mathbb{X}) \to aLGW(\mathbb{Y}^1, \mathbb{Y}^2; \mathbb{X}).$$

*Proof of theorem G.1.* By the lemma F.1 in (Bai et al., 2024), when $n$ is sufficiently large, in particular, when

$$\lambda_n > M := \sup_{\substack{y^1, y^{1\prime} \in Y^1 \\ y^2, y^{2\prime} \in Y^2}} |g_{Y^1}(y^1, y^{1\prime}) - g_{Y^2}(y^1, y^{1\prime})|,$$

for each $\gamma^1 \in \Gamma_{\leq}^*(\mathbb{X}, \mathbb{Y}^1)$, $|\gamma^1| = \min(|\mu|, |\nu^1|) = 1$. That is

$$\Gamma_{\leq, \lambda_n}^*(\mathbb{X}, \mathbb{Y}^1) = \Gamma^*(\mathbb{X}, \mathbb{Y}^1).$$

Similarly, when $n$ is sufficiently large, we have , $\Gamma_{\leq, \lambda_n}^*(\mathbb{X}, \mathbb{Y}^2) = \Gamma^*(\mathbb{X}, \mathbb{Y}^2)$.

Pick $\gamma^1 \in \Gamma_{\leq}^*(\mathbb{X}, \mathbb{Y}^1) = \Gamma^*(\mathbb{X}, \mathbb{Y}^1), \gamma^2 \in \Gamma_{\leq}^*(\mathbb{X}, \mathbb{Y}^2) = \Gamma^*(\mathbb{X}, \mathbb{Y}^2)$, since $|\gamma^1| = |\mu| = |\nu^1| = |\nu^2| = 1$, the mass penalty term vanishes in (65), thus we have

$$LPGW_{\lambda_n}(\mathbb{Y}^1, \mathbb{Y}^2; \mathbb{X}, \gamma^1, \gamma^2) = LGW(\mathbb{Y}^1, \mathbb{Y}^2; \mathbb{X}, \gamma^1, \gamma^2).$$

Take the infimum over all $\gamma^1, \gamma^2$ and the take a limit for $n \to \infty$, we prove

$$LPGW_{\lambda_n}(\mathbb{Y}^1, \mathbb{Y}^2; \mathbb{X}) \to LGW(\mathbb{Y}^1, \mathbb{Y}^2; \mathbb{X}).$$

Similarly, in this case, $\gamma_c^1 = \gamma_c^2 = 0$

$$LPGW_{\lambda_n}(\widetilde{\mathbb{Y}}_{\gamma^1}, \widetilde{\mathbb{Y}}_{\gamma^2}; \mathbb{X}, \widetilde{\gamma}^1, \widetilde{\gamma}^2) + \lambda(|\gamma_c^1| + |\gamma_c^2|) = LGW(\widetilde{\mathbb{Y}}_{\gamma^1}, \widetilde{\mathbb{Y}}_{\gamma^2}; \mathbb{X}, \widetilde{\gamma}^1, \widetilde{\gamma}^2).$$

Take the infimum on both sides over $\gamma^1, \gamma^2$ and take $\lambda_n \to \infty$, we obtain

$$aLPGW(\mathbb{Y}^1, \mathbb{Y}^2; \mathbb{X}) \to aLGW(\mathbb{Y}^1, \mathbb{Y}^2; \mathbb{X}).$$

$\square$

# H   PROOF OF THEOREM 3.2

## H.1   PROOF OF THEOREM 3.2: PARTS (1), (3)

(1) Similar to proof in Theorem 2.1, in this case, we have $\mathcal{T}_{\gamma^*} = T$, $\gamma_X^* - a.s.$

(3) Based on statement (2) of Theorem 3.2 (see the proof in next section), we have $\widetilde{\gamma}^1, \widetilde{\gamma}^2$ are optimal solutions for $PGW_\lambda(\mathbb{X}, \widetilde{\mathbb{Y}}_{\gamma^1}), PGW_\lambda(\mathbb{X}, \widetilde{\mathbb{Y}}_{\gamma^2})$ respectively.

By (1), we have $T^1 = \mathcal{T}_{\gamma^1}, T^2 = \mathcal{T}_{\gamma^2}$ $\gamma_X^1 \wedge \gamma_Y^2 -$a.s. Thus, we have:

$$LPGW_\lambda(\mathbb{Y}^1, \mathbb{Y}^2; \mathbb{X}, \gamma^1, \gamma^2)$$

$$= \inf_{\mu' \leq \gamma_X^1 \wedge \gamma_X^2} \int_{X^{\otimes 2}} |d_{Y^1}(T^1(\cdot_1), T^1(\cdot_2)) - d_{Y^2}(T^1(\cdot_1), T^2(\cdot_2))|^2 d(\mu')^{\otimes 2} + \lambda(|\nu^1|^2 + |\nu^2|^2 - 2|\mu'|^2)$$

$$= \inf_{\mu' \leq \gamma_X^1 \wedge \gamma_X^2} \int_{X^{\otimes 2}} |d_{Y^1}(\mathcal{T}_{\gamma^1}(\cdot_1), \mathcal{T}_{\gamma^1}(\cdot_2)) - d_{Y^2}(\mathcal{T}_{\gamma^2}(\cdot_1), \mathcal{T}_{\gamma^2}(\cdot_1))|^2 d(\mu')^{\otimes 2} + \lambda(|\widetilde{\nu}_{\gamma^1}|^2 + |\widetilde{\nu}_{\gamma^2}|^2 - 2|\mu'|^2)$$

$$+ \lambda(|\nu^1|^2 - |\widetilde{\nu}_{\gamma^1}|^2 + |\nu^2|^2 - |\widetilde{\nu}_{\gamma^2}|^2)$$

$$= \inf_{\mu' \leq \gamma_X^1 \wedge \gamma_X^2} \left[ \int_{X^{\otimes 2}} |d_{Y^1}(\mathcal{T}_{\gamma^1}(\cdot_1), \mathcal{T}_{\gamma^1}(\cdot_2)) - d_{Y^2}(\mathcal{T}_{\gamma^2}(\cdot_1), \mathcal{T}_{\gamma^2}(\cdot_1))|^2 d(\mu')^{\otimes 2} + \lambda(|\widetilde{\nu}_{\gamma^1}|^2 + |\widetilde{\nu}_{\gamma^2}|^2 - 2|\mu'|^2) \right]$$

$$+ \lambda(|\gamma_c^1| + |\gamma_c^2|) \tag{86}$$

$$= LPGW_\lambda(\widetilde{\mathbb{Y}}_{\gamma^1}, \widetilde{\mathbb{Y}}_{\gamma^2}; \mathbb{X}, \gamma^1, \gamma^2) + \lambda(|\gamma_c|^1 + |\gamma_c|^2)$$

where (86) holds since

$$\widetilde{\nu}^1 = \mathcal{T}_{\gamma^1}\gamma_1^1 = T^1\gamma_1^1 = (\pi_Y)_\# \gamma^1 \leq \nu^1.$$

Thus $|\nu^1|^2 - |\widetilde{\nu}^1|^2 = |(\nu^1)^{\otimes 2} - ((\pi_Y)_\# \gamma^1)^{\otimes 2}| = |\gamma_c^1|$; and similarly we have $|\nu^2|^2 - |\widetilde{\nu}^2|^2 = |\gamma_c^2|$.

## H.2   PROOF OF THEOREM 3.2: PART (2)

### H.2.1   PROOF IN THE DISCRETE CASE.

**Notation setup**

We first demonstrate a simplified proof in the discrete case. Next, we will provide the proof of the statement in the general case.

Suppose

$$\mu = \sum_{i=1}^n p_i x_i, \qquad \nu = \sum_{j=1}^m q_j y_j,$$

$$g_X(x, x') = x^\top x', \qquad \forall x, x' \in X,$$

$$g_X(y, y') = y^\top y', \qquad \forall y, y' \in Y.$$

Choose $\gamma^* \in \Gamma_{\leq}^*(\mathbb{X}, \mathbb{Y})$, we obtain the corresponding barycentric projected measure:

$\widetilde{\nu} := \widetilde{\nu}_{\gamma^*} = \sum_{i=1}^n \widetilde{q}_i \widetilde{y}_i$ with

$$\widetilde{q}_i = \sum_{j=1}^n \gamma_{i,j}^*, \qquad \forall i \in [1:n], \tag{87}$$

$$\widetilde{y}_i = \begin{cases} \frac{1}{\widetilde{q}_i} \gamma_{ij}^* x_i & \text{if } \widetilde{q}_i > 0, \\ 0 & \text{elsewhere.} \end{cases} \tag{88}$$

Then $\widetilde{\mathbb{Y}}_{\gamma^*} = (Y, g_Y, \widetilde{\nu})$, $\widetilde{\gamma}^* = \mathrm{diag}(\widetilde{q}_1, \ldots \widetilde{q}_n)$. Our goal is to show $\widetilde{\gamma}^* = \mathrm{diag}(\widetilde{q}_1, \ldots, \widetilde{q}_n)$ is optimal in $PGW_\lambda(\mathbb{X}, \widetilde{\mathbb{Y}}_{\gamma^*})$.

Pick $\widetilde{\gamma} \in \Gamma_{\le}(p, \widetilde{q}) := \Gamma_{\le}(\mu, \widetilde{\nu})$, similar to section E.5, we set diagonal matrix $(\widetilde{\gamma}^*)^{-1} \in \mathbb{R}_+^{n \times n}$ as

$$(\widetilde{\gamma}^*)_{ii}^{-1} = \begin{cases} \frac{1}{\widetilde{q}_i} & \text{if } \widetilde{q}_i > 0, \\ 0 & \text{elsewhere.} \end{cases}$$

Let $\gamma := \widetilde{\gamma}(\widetilde{\gamma}^*)^{-1}\gamma^* \in \mathbb{R}_+^{n \times m}$. Let $\gamma_X := \gamma 1_m$ and $\gamma_Y := \gamma^\top 1_n$.

In addition, we set $D = \{i : \widetilde{q}_i > 0\}$, and $1_D \in \mathbb{R}^n$ with $1_D[i] = 1, \forall i \in D$ and $1_D[i] = 0$ elsewhere. Then we have:

$$\gamma 1_m = \widetilde{\gamma}(\widetilde{\gamma}^*)^{-1}\gamma^* 1_m = \widetilde{\gamma}(\widetilde{\gamma}^*)^{-1}\widetilde{q} = \widetilde{\gamma} 1_D = \widetilde{\gamma}_X \le p_0, \tag{89}$$

$$\gamma^\top 1_n = (\gamma^*)^\top (\widetilde{\gamma}^*)^{-1}\widetilde{\gamma}^\top 1_n \le (\gamma^*)^\top (\widetilde{\gamma}^*)^{-1}\widetilde{q} = (\gamma^*)^\top 1_D = \gamma_Y^* \le q. \tag{90}$$

Therefore, $\gamma_X \le \widetilde{\gamma}_X \le p$, $\gamma_Y \le \gamma_Y^* \le q$. Thus, $\gamma \in \Gamma_{\le}(p, q)$.

**Relation between $PGW_\lambda(\mathbb{X}, \mathbb{Y})$ and $PGW_\lambda(\mathbb{X}, \widetilde{\mathbb{Y}})$.**

Similar to (54),(57), we obtain

$$C(\gamma^*; \mathbb{X}, \mathbb{Y}, \lambda) = \langle (x^\top x')^2, (\gamma_X^*)^{\otimes 2} \rangle + \langle (y^\top y')^2, (\gamma_Y^*)^{\otimes 2} \rangle + \lambda(|p|^2 + |q|^2 - 2|\gamma^*|)$$
$$- 2 \sum_{i,i'=1}^n \sum_{j,j'=1}^m x_i^\top x_i' y_j^\top y_j' \gamma_{i,j}^* \gamma_{i',j'}^*$$

$$C(\widetilde{\gamma}^*; \mathbb{X}, \widetilde{\mathbb{Y}}, \lambda) = \langle (x^\top x')^2, (\widetilde{\gamma}_X^*)^{\otimes 2} \rangle + \langle (y^\top y')^2, (\widetilde{\gamma}_Y^*)^{\otimes 2} \rangle + \lambda(|p|^2 + |\widetilde{q}|^2 - 2|\widetilde{\gamma}^*|)$$
$$- 2 \sum_{i,i'=1}^n \sum_{j,j'=1}^m x_i^\top x_i' y_j^\top y_j \gamma_{i,j}^* \gamma_{i',j'}^*$$

where $(x^\top x')^2 := [(x_i^\top x_{i'})^2]_{i,i' \in [1:n]} \in \mathbb{R}^{n \times n}$, $(\gamma_X^*)^{\otimes 2} = \gamma_X^*(\gamma_X^*)^\top$, $\langle (x^\top x')^2, (\gamma_X^*)^{\otimes 2} \rangle := \sum_{i,i'=1}^n (x_i^\top x_{i'})^2 (\gamma_X^*)_i (\gamma_X^*)_{i'}$, is the element-wise dot product. All other notations are defined similarly.

Combined the above two equalities with the fact $|\gamma^*| = |\gamma_Y^*|$ and $\gamma_X^* = \widetilde{\gamma}_X^*$, we obtain:

$$C(\widetilde{\gamma}^*; \mu, \widetilde{\nu}, \lambda) = C(\gamma^*; \mathbb{X}, \mathbb{Y}, \lambda) + \langle (\widetilde{y}^\top \widetilde{y}')^2, (\widetilde{\gamma}_Y^*)^{\otimes 2} \rangle - \langle (y^\top y')^2, (\gamma_Y^*)^{\otimes 2} \rangle + \lambda(|q|^2 - |\widetilde{q}|^2). \tag{91}$$

Similarly, from the fact $\gamma_X = \widetilde{\gamma}_X$ (see (89)) we obtain

$$C(\widetilde{\gamma}; \mu, \widetilde{\nu}, \lambda) = C(\gamma; \mathbb{X}, \mathbb{Y}, \lambda) + \langle (\widetilde{y}^\top \widetilde{y}')^2, (\widetilde{\gamma}_Y)^{\otimes 2} \rangle - \langle (y^\top y')^2, (\gamma_Y)^{\otimes 2} \rangle + \lambda(|q|^2 - |\widetilde{q}|^2). \tag{92}$$

Thus

$$C(\widetilde{\gamma}^*; \mathbb{X}, \widetilde{\mathbb{Y}}, \lambda) - C(\widetilde{\gamma}; \mathbb{X}, \widetilde{\mathbb{Y}}, \lambda)$$
$$= C(\gamma^*; \mathbb{X}, \mathbb{Y}, \lambda) - C(\gamma; \mathbb{X}, \mathbb{Y}, \lambda) + \langle (\widetilde{y}^\top \widetilde{y}')^2, (\widetilde{\gamma}_Y^*)^{\otimes 2} - (\widetilde{\gamma}_Y)^{\otimes 2} \rangle - \langle (y^\top y')^2, (\gamma_Y^*)^{\otimes 2} - (\gamma_Y)^{\otimes 2} \rangle$$
$$\le \langle (\widetilde{y}^\top \widetilde{y}')^2, (\widetilde{\gamma}_Y^*)^{\otimes 2} - (\widetilde{\gamma}_Y)^{\otimes 2} \rangle - \langle (y^\top y')^2, (\gamma_Y^*)^{\otimes 2} - (\gamma_Y)^{\otimes 2} \rangle \tag{93}$$
$$= \sum_{i,i' \in D} (\widetilde{y}_i^\top \widetilde{y}_{i'})^2 ((\widetilde{\gamma}_Y^*)_i (\widetilde{\gamma}_Y^*)_{i'} - (\widetilde{\gamma}_Y)_i (\widetilde{\gamma}_Y)_{i'}) - \sum_{j,j'=1}^m (y_j^\top y_{j'})^2 ((\gamma_Y^*)_j (\gamma_Y^*)_{j'} - (\gamma_Y)_j (\gamma_Y)_{j'})$$
$$\le \sum_{i,i' \in D} \sum_{j,j'=1}^m \frac{\gamma_{i,j}^* \gamma_{i,j'}^*}{\widetilde{q}_i^1 \widetilde{q}_{i'}^1} ((x_i^\top x_{i'})^2) ((\widetilde{\gamma}_Y^*)_i (\widetilde{\gamma}_Y^*)_{i'} - (\widetilde{\gamma}_Y)_i (\widetilde{\gamma}_Y)_{i'})$$
$$- \sum_{j,j'=1}^m (y_j^\top y_{j'})^2 ((\gamma_Y^*)_j (\gamma_Y^*)_{j'} - (\gamma_Y)_j (\gamma_Y)_{j'}) \tag{94}$$
$$= \sum_{j,j'=1}^m (y_j^\top y_{j'})^2 \left( \sum_{i,i' \in D} \frac{\gamma_{i,j}^* \gamma_{i,j'}^*}{\widetilde{q}_i^1 \widetilde{q}_{i'}^1} ((\widetilde{\gamma}_Y^*)_i (\widetilde{\gamma}_Y^*)_{i'} - (\widetilde{\gamma}_Y)_i (\widetilde{\gamma}_Y)_{i'}) - ((\gamma_Y^*)_j (\gamma_Y^*)_{j'} - (\gamma_Y)_j (\gamma_Y)_{j'}) \right)$$
$$= 0 \tag{95}$$

where (93) holds since $\gamma^1$ is optimal in $PGW_\lambda(\mathbb{X}, \mathbb{Y})$; (94) follows from the facts $\gamma_Y \le \gamma_Y^*$ (see (90)), and the fact

$$\widetilde{y}_i^\top \widetilde{y}_i' = \sum_{j=1}^m \sum_{j'=1}^m \frac{\gamma_{i,j}\gamma_{i',j'} y_j^\top y_{j'}'}{\widetilde{q}_i \widetilde{q}_i'}, \forall i, i' \in D$$

and Jensen's inequality:

$$(\widetilde{y}_i^\top \widetilde{y}_i')^2 \le \sum_{j=1}^m \sum_{j'=1}^m \frac{\gamma_{i,j}\gamma_{i',j'} (y_j^\top y_i')^2}{\widetilde{q}_i \widetilde{q}_i'}.$$

The last equality (95) holds from the following:

For each $(j, j' \in [1:m])$, we have

$$\sum_{i,i'\in D} \frac{\gamma_{i,j}^* \gamma_{i,j'}^*}{\widetilde{q}_i^1 \widetilde{q}_{i'}^1} ((\widetilde{\gamma}_Y^1)_i (\widetilde{\gamma}_Y^*)_{i'} - (\widetilde{\gamma}_Y)_i (\widetilde{\gamma}_Y)_{i'}) - ((\gamma_Y^*)_j (\gamma_Y^*)_{j'} - (\gamma_Y)_j (\gamma_Y)_{j'})$$

$$= \underbrace{\left( \sum_{i,i'\in D} \frac{\gamma_{i,j}^* \gamma_{i,j'}^*}{\widetilde{q}_i^1 \widetilde{q}_{i'}^1} (\widetilde{\gamma}_Y^*)_i (\widetilde{\gamma}_Y^*)_{i'} \right) - (\gamma_Y^*)_j (\gamma_Y^*)_{j'}}_{A} - \underbrace{\left( \sum_{i,i'\in D} \frac{\gamma_{i,j}^* \gamma_{i,j'}^*}{\widetilde{q}_i \widetilde{q}_{i'}} (\widetilde{\gamma}_Y)_i (\widetilde{\gamma}_Y)_{i'} \right) - (\gamma_Y)_j (\gamma_Y)_{j'}}_{B},$$

where

$$A = \sum_{i,i'\in D} \frac{\gamma_{i,j}^* \gamma_{i',j'}^*}{\widetilde{q}_i^1 \widetilde{q}_{i'}^1} \widetilde{q}_i^1 \widetilde{q}_{i'}^1 - \widetilde{q}_j^1 \widetilde{q}_{j'}^1$$

$$= \sum_{i,i'\in D} \gamma_{i,j}^* \gamma_{i',j'}^* - \widetilde{q}_j^1 \widetilde{q}_{j'}^1$$

$$= \left( \sum_{i\in D} \gamma_{i,j}^* \right) \left( \sum_{i'\in D} \gamma_{i',j'}^* \right) - \widetilde{q}_j^1 \widetilde{q}_{j'}^1 = 0$$

$$B = \sum_{i,i'\in D} \frac{\gamma_{i,j}^* \gamma_{i,j'}^*}{\widetilde{q}_i \widetilde{q}_{i'}} \sum_{k,k'=1}^n \widetilde{\gamma}_{k,i} \widetilde{\gamma}_{k',i'} - \sum_{k,k'=1}^n \gamma_{k,j} \gamma_{k',j'}$$

$$= \sum_{i,i'\in D} \sum_{k,k'=1}^n \frac{\gamma_{i,j}^* \gamma_{i,j'}^*}{\widetilde{q}_i^1 \widetilde{q}_{i'}^1} \widetilde{\gamma}_{k,i} \widetilde{\gamma}_{k',i'} - \sum_{k,k'=1}^n (\sum_{i\in D} \frac{\widetilde{\gamma}_{k,i} \gamma_{i,j}^*}{\widetilde{q}_i})(\sum_{i\in D} \frac{\widetilde{\gamma}_{k',i'} \gamma_{i',j'}^*}{\widetilde{q}_{i'}})$$

$$= \sum_{i,i'\in D} \sum_{k,k'=1}^n \frac{\gamma_{i,j}^* \gamma_{i,j'}^*}{\widetilde{q}_i^1 \widetilde{q}_{i'}} \widetilde{\gamma}_{k,i} \widetilde{\gamma}_{k',i'} - \sum_{i,i'\in D} \sum_{k,k'=1}^n \frac{\widetilde{\gamma}_{k,i} \widetilde{\gamma}_{k',i'} \gamma_{i,j}^* \gamma_{i',j'}^*}{\widetilde{q}_i^1 \widetilde{q}_{i'}^1}$$

$$= 0$$

and thus we complete the proof.

### H.3    Proof for the general case

#### H.3.1    Notation setup and related lemma

Suppose

$$g_X(x, x') = \alpha_0 x^\top x', g_Y = \alpha x^\top y', \forall x, x' \in X, Y, y' \in X.$$

Choose $\gamma^* \in \Gamma_{\le,\lambda}^*(\mathbb{X}, \mathbb{Y})$. The barycentric projection mapping is

$$\mathcal{T}_{\gamma^*}(x) := \int y' d\gamma^*(y'|x'), \forall x \in \text{supp}(\mu).$$

We obtain: $\widetilde{\nu} := \widetilde{\nu}_{\gamma^*} = (\mathcal{T}_{\gamma^*})_{\#}\mu$, $\widetilde{\gamma}^* = (\text{id} \times \mathcal{T}_{\gamma^*})_{\#}\mu$ and we $\widetilde{\mathbb{Y}}_{\gamma^*} := (X, g_Y, \widetilde{\nu})$.

**Our goal is to show** $\widetilde{\nu}$ is optimal in $PGW_\lambda(\mathbb{X}, \widetilde{\mathbb{Y}})$. Equivalently, pick $\widetilde{\gamma} \in \Gamma_{\leq}(\mu, \widetilde{\nu})$, we need to show

$$C(\widetilde{\gamma}^*; \mathbb{X}, \mathbb{Y}, \lambda) \leq C(\widetilde{\gamma}; \mathbb{X}, \mathbb{Y}, \lambda).$$

Set $\gamma$ from (52):

$$\gamma = \gamma_{Y|X}^* \widetilde{\gamma}_{X|Y}^* \widetilde{\gamma}.$$

We have $\gamma$ satisfies the following two lemma:

**Lemma H.1.** *Choose* $\gamma^* \in \Gamma_{\leq}(\mu, \nu), \widetilde{\gamma} \in \Gamma_{\leq}(\mu, \widetilde{\nu})$. *Set* $\widetilde{\nu} = (\mathcal{T}_{\gamma^*})_\# \mu$, *then* $\gamma$ *defined in (52) satisfies the following:*

(a) $\gamma \in \Gamma_{\leq}((\pi_X)_\# \widetilde{\gamma}, (\pi_Y)_\# \gamma^*) \subset \Gamma_{\leq}(\mu, \nu^1)$, *furthermore:*

$$(\pi_X)_\# \gamma = (\pi_X)_\# \widetilde{\gamma} \tag{96}$$
$$(\pi_Y)_\# \gamma \leq (\pi_Y)_\# \gamma^* \tag{97}$$

(b) *If* $\widetilde{\gamma} = \widetilde{\gamma}^*$, *then* $\gamma = \gamma^*$.

(c) *Regarding the second marginal of* $\gamma$, *we have for each test function* $\phi_X \in C_0(X)$:

$$\int_X \phi_Y(y) d\gamma_Y(y) = \int_{Y \times X \times Y \times X} \phi_Y(y) d\widetilde{\gamma}^*(y|x^0) d\widetilde{\gamma}^*(x^0|\widetilde{y}) d\widetilde{\gamma}_Y(\widetilde{y}). \tag{98}$$

*Proof.*

(a) Note, in a discrete setting, this statement has been proved in (89),(90). Pick test functions $\phi_X \in C_0(X), \phi_Y \in C_0(Y)$ with $\phi_Y \geq 0$, we have:

$$\begin{aligned}
\langle \phi_X, \gamma \rangle &= \int_{X \times Y} \phi_X(x) d\gamma(x, y) \\
&= \int_{X \times Y} \int_X \int_Y \phi_X(x) d\gamma^*(y|x^0) d\widetilde{\gamma}^*(x^0|\widetilde{y}) d\widetilde{\gamma}(x, \widetilde{y}) \\
&= \int_{X \times Y} \phi_X(x) d\widetilde{\gamma}(x, \widetilde{y}) \\
&= \int_X \phi_X(x) d\widetilde{\gamma}_X(x) \\
\langle \phi_Y, \gamma_Y \rangle &= \int_{X \times Y} \phi_Y(y) d\gamma(x, y) \\
&= \int_{X \times Y \times X \times Y} \phi_Y(y) d\gamma^*(y|x^0) d\widetilde{\gamma}^*(x^0|\widetilde{y}) d\widetilde{\gamma}(x, \widetilde{y}) \\
&= \int_{X \times X} \int_X \int_Y \phi_Y(y) d\gamma^*(y|x^0) d\widetilde{\gamma}^*(x^0|\widetilde{y}) d\widetilde{\gamma}_Y(\widetilde{y}) \\
&\leq \int_{X \times Y} \int_X \int_Y \phi_Y(y) d\gamma^*(y|x^0) d\widetilde{\gamma}^*(x^0|\widetilde{y}) d\widetilde{\gamma}_Y^*(\widetilde{y}) \\
&= \int_{X \times Y} \phi_Y(y) d\gamma^*(y|x^0) d\widetilde{\gamma}_X^*(x^0) \\
&= \int_{X \times Y} \phi_Y(y) d\gamma^*(y|x^0) d\gamma_X^*(x^0) \\
&= \int_Y \phi_Y(y) d\gamma_Y^*(y)
\end{aligned}$$

where the inequality holds from the fact $\widetilde{\gamma}_Y \leq \widetilde{\gamma}_Y^*$.

(b) If $\widetilde{\gamma} = \widetilde{\gamma}^*$, pick $\phi \in C_0(X \times X)$, we have:

$$
\begin{aligned}
\langle \phi, \gamma \rangle &= \int_{X \times Y} \int_X \int_Y \phi(x, y) d\gamma^*(y|x^0) d\widetilde{\gamma}^*(x^0|\widetilde{y}) d\widetilde{\gamma}^*(\widetilde{y}|x) d\widetilde{\gamma}_X^*(x) \\
&= \int_{X \times Y} \int_X \int_Y \phi(x, y) d\gamma^*(y|x^0) d\widetilde{\gamma}_X^*(x) \\
&= \int_{X \times Y} \int_X \phi(x, y) d\gamma^*(y|x^0) \delta(x - x^0) d\gamma_X^*(x) \\
&= \int_{X \times Y} \phi(x, y) d\gamma^*(y|x) d\gamma_X^*(x) \\
&= \int_{X \times X} \phi(x, y) d\gamma^*(x, y)
\end{aligned}
$$

(c) It follows directly from the definition of $\gamma$.

$\square$

### H.3.2 Relation between $PGW_\lambda(\mathbb{X}, \mathbb{Y})$ and $PGW_\lambda(\mathbb{X}, \widetilde{\mathbb{Y}})$.

From (54),(57),(60),(61) and (62), we obtain:

$$
C(\widetilde{\gamma}^*; \mathbb{X}, \widetilde{\mathbb{Y}}, \lambda) = C(\gamma^*; \mathbb{X}, \mathbb{Y}, \lambda) + \langle g_Y^2, (\widetilde{\gamma}_Y^*)^{\otimes 2} \rangle - \langle g_Y^2, (\gamma_Y^*)^{\otimes 2} \rangle + \lambda(|\nu|^2 - |\widetilde{\nu}|^2) \tag{99}
$$

$$
C(\widetilde{\gamma}; \mathbb{X}, \widetilde{\mathbb{Y}}, \lambda) = C(\gamma; \mathbb{X}, \mathbb{Y}, \lambda) + \langle g_Y^2, (\widetilde{\gamma}_Y)^{\otimes 2} \rangle - \langle g_Y^2, (\gamma_Y)^{\otimes 2} \rangle + \lambda(|\nu|^2 - |\widetilde{\nu}|^2) \tag{100}
$$

From (99)-(100), we obtain

$$
\begin{aligned}
&C(\widetilde{\gamma}^*; \mathbb{X}, \widetilde{\mathbb{Y}}, \lambda) - C(\widetilde{\gamma}; \mathbb{X}, \widetilde{\mathbb{Y}}, \lambda) \\
&= (C(\gamma^*; \mathbb{X}, \mathbb{Y}, \lambda) - C(\gamma; \mathbb{X}, \mathbb{Y}, \lambda)) + ((\widetilde{\gamma}^*)^{\otimes 2} - \widetilde{\gamma}^{\otimes 2}))(g_Y^2(\widetilde{y}, \widetilde{y}')) - ((\gamma^*)^{\otimes 2} - \gamma^{\otimes 2})(g_Y^2(x, y')) \\
&\leq \underbrace{\langle g_Y^2, (\widetilde{\gamma}_Y^*)^{\otimes 2} - (\widetilde{\gamma}_Y)^{\otimes 2} \rangle}_{A} - \langle g_Y^2, (\gamma_Y^*)^{\otimes 2} - \gamma_Y^{\otimes 2} \rangle \tag{101}
\end{aligned}
$$

where (101) holds from the fact $\gamma^*$ is optimal in $PGW_\lambda(\mathbb{X}, \mathbb{Y})$.

In addition, pick $\widetilde{y}, \widetilde{y}' \in \text{supp}(\widetilde{\gamma}_Y^*)$, then for each $x \in \text{supp}(\widetilde{\gamma}^*(\cdot_1|y)), x' \in \text{supp}(\widetilde{\gamma}^*(\cdot_1|y'))$, we have:

$$
\widetilde{y} = \mathcal{T}_{\gamma^*}(x), \widetilde{y}' = \mathcal{T}_{\gamma^*}(x').
$$

Thus,

$$
A = \int_{(X \times Y)^{\otimes 2}} \int_{X^{\otimes 2}} g_Y^2(\mathcal{T}_{\gamma^*}(x^0), \mathcal{T}_{\gamma^*}(x^{0\prime})) d(\gamma_{X|Y}^{\otimes 2})((x^0, x^{0\prime})|\widetilde{(y}, \widetilde{y}')) d((\widetilde{\gamma}_Y^*)^{\otimes 2} - (\widetilde{\gamma}_Y)^{\otimes 2})((x, x'), (\widetilde{y}, \widetilde{y}'))
$$

From Lemma E.2 and Jensen's inequality

$$
\begin{aligned}
&g_Y^2(\mathcal{T}_{\gamma^*}(x^0), \mathcal{T}_{\gamma^*}(x^{0\prime})) \\
&= \left( \int_{Y^{\otimes 2}} g_Y(y, y') d\gamma^*(y|x^0) d\gamma^*(y'|x^{\prime 0\prime}) \right)^2 \\
&\leq \int_{Y^{\otimes 2}} g_Y^2(y, y') d\gamma^*(y|x^0) d\gamma^*(y'|x^{\prime 0\prime}).
\end{aligned}
$$

Combined it with the fact $\widetilde{\gamma}_Y^* \leq \widetilde{\gamma}_Y$ (see Lemma H.1 (a)), we obtain

$$
\begin{aligned}
A &\leq \int_{(X \times Y)^{\otimes 2}} \int_{X^{\otimes 2}} g_Y^2(y, y') d\gamma^*(y|x^0) d\gamma^*(y'|x^{\prime 0\prime}) d\widetilde{\gamma}^*(x^0|y) d\widetilde{\gamma}^*(x^{0\prime}|y') d\left( (\widetilde{\gamma}_Y^*)^{\otimes 2} - \widetilde{\gamma}_Y^{\otimes 2} \right) \left( (x, x'), (y, y') \right) \\
&= \langle g_Y^2, (\gamma^*)_{Y|X}^{\otimes 2} (\widetilde{\gamma}^*)_{X|Y}^{\otimes 2} ((\widetilde{\gamma}_Y^*)^{\otimes 2} - (\widetilde{\gamma}_Y)^{\otimes 2}) \rangle. \tag{102}
\end{aligned}
$$

Thus, we can continue bound (101):

$$(101) \le \langle g_Y^2, (\gamma^*)_{Y|X}^{\otimes 2}(\widetilde{\gamma}^*)_{X|Y}^{\otimes 2}((\widetilde{\gamma}_Y^*)^{\otimes 2} - (\widetilde{\gamma}_Y)^{\otimes 2})\rangle - \langle g_Y^2, (\gamma_Y^*)^{\otimes 2} - \gamma_Y^{\otimes 2}\rangle$$

$$= \underbrace{\langle g_Y, (\gamma^*)_{Y|X}^{\otimes 2}(\widetilde{\gamma}^*)_{X|Y}^{\otimes 2}(\widetilde{\gamma}_Y^*)^{\otimes 2} - (\gamma_Y^*)^{\otimes 2}\rangle}_{B_1}$$

$$- \underbrace{\langle g_Y, (\gamma^*)_{Y|X}^{\otimes 2}(\widetilde{\gamma}^*)_{X|Y}^{\otimes 2}(\widetilde{\gamma}_Y)^{\otimes 2} - (\gamma_Y)^{\otimes 2}\rangle}_{B_2}, \tag{103}$$

where $B_2 = 0$ from lemma H.1 (c) and $B_1 = B_2$ from lemma H.1(b). Therefore w, we complete the proof.

### H.4 PROOF OF PROPOSITION 3.3

The proposition is directly implied by definition (20) and Proposition 3.1 (3).

## I SPECIAL CASE: LINEAR MASS-CONSTRAINED PARTIAL GROMOV-WASSERSTEIN DISTANCE

In this section, we introduce the linearization technique for the "mass-constraint" partial Gromov-Wasserstein distance, which can be regarded as a special case of the proposed LPGW distance.

Mass-constraint Partial Gromov-Wasserstein distance is defined as:

$$MPGW_\rho(\mathbb{X}, \mathbb{Y}) = \inf_{\gamma \in \Gamma_{\le}^\eta(\mu,\nu)} \int_{(X \times Y)^{\otimes 2}} |g_X(x,x') - g_Y(y,y')|^2 d\gamma^\otimes, \tag{104}$$

where $\eta \in [0, \min(|\mu|, |\nu^1|)]$ is a fixed number.

$$\Gamma_{\le}^\eta(\mu,\nu) = \{\gamma \in \mathcal{M}_+(X \times Y) : (\pi_X)_\# \gamma \le \mu, (\pi_Y)_\# \gamma \le \nu, |\gamma| = \eta\},$$

We use $\Gamma_{\le}^\eta(\mathbb{X}, \mathbb{Y}^1)$ to denote the set of all optimal transportation plans.

Consider the following gm-spaces $\mathbb{X} = (X, g_X, \mu), \mathbb{Y}^1 = (Y^1, g_{Y^1}, \nu^1), \mathbb{Y}^2 = (Y^2, g_{Y^2}, \nu^2)$, given $\eta \in [0, \min(|\nu^1|, |\nu^2|)$ and we suppose $|\mu| = \eta$. Choose $\gamma^1 \in \Gamma_{\le}^\eta(\mathbb{X}, \mathbb{Y}^1), \gamma^2 \in \Gamma_{\le}^\eta(\mathbb{X}, \mathbb{Y}^2)$. Suppose the Monge mapping assumption holds and let $T^1, T^2$ be the corresponding transportation plan. We define

$$k_{\gamma^1} : (\cdot_1, \cdot_2) \mapsto d_X(\cdot_1, \cdot_2) - d_{Y^1}(T_1(\cdot_1), T_1(\cdot_2))$$

as linear MPGW embedding of $\mathbb{Y}^1$ given $\gamma^1$. $k_{\gamma^2}$ is defined similalry.

Thus, similar to (15) and (7), the linear M-PGW distance between $\mathbb{Y}^1, \mathbb{Y}^2$, given $\gamma^1, \gamma^2$ is defined as

$$LMPGW_\eta(\mathbb{Y}^1, \mathbb{Y}^2; \mathbb{X}, \gamma^1, \gamma^2) := \|k_{\gamma^1} - k_{\gamma^2}\|_{\mu^{\otimes 2}}^2. \tag{105}$$

and similar to (7),(15), the general case without Monge mapping assumption is

$$LMPGW_\eta(\mathbb{Y}^1, \mathbb{Y}; \mathbb{X}, \gamma^1, \gamma^2) := \inf_{\gamma \in \Gamma_{\le}^\eta(\gamma^1, \gamma^2; \mu)} \int_{(X \times Y)^{\otimes 2}} |g_X - g_Y|^2 d\gamma^{\otimes 2}, \tag{106}$$

where $\Gamma_{\le}^\rho(\gamma^1, \gamma^2; \mu) := \{\gamma \in \mathcal{M}_+(X \times Y^1 \times Y^2) : (\pi_{X,Y^1})_\# \gamma \le \gamma^1, (\pi_{X,Y^2})_\# \gamma \le \gamma^2, |\gamma| = \eta\}$. And similar to (10),(20), by applying barycentric projection $\mathcal{T}_{\gamma^1}, \mathcal{T}_{\gamma^2}$, we define the approximated linear-MPGW distance as follows:

$$aLMPGW_\eta(\mathbb{Y}^1, \mathbb{Y}^2; \mathbb{X}) := \inf_{\substack{\gamma^1 \in \Gamma_{\le}^\rho(\mathbb{X}, \mathbb{Y}^1) \\ \gamma^2 \in \Gamma_{\le}^{\tilde{\rho}}(\mathbb{X}, \mathbb{Y}^2)}} LMPGW(\widehat{\mathbb{Y}}^1, \widehat{\mathbb{Y}}^2; \mathbb{X})$$

$$= \inf_{\substack{\gamma^1 \in \Gamma_{\le}^\rho(\mathbb{X}, \mathbb{Y}^1) \\ \gamma^2 \in \Gamma_{\le}^{\tilde{\rho}}(\mathbb{X}, \mathbb{Y}^2)}} \int_{X^{\otimes 2}} |d_{Y^1}(\mathcal{T}_{\gamma^1}(x), \mathcal{T}_{\gamma^1}(x') - d_{Y^2}(\mathcal{T}_{\gamma^2}(x), \mathcal{T}_{\gamma^2}(x'))| d\mu^{\otimes 2}. \tag{107}$$

Similar to theorem 3.2, we have:

**Proposition I.1.** *Choose $\gamma^* \in \Gamma_{\leq}^{\rho,*}(\mathbb{X}, \mathbb{Y}), \gamma^1 \in \Gamma_{\leq}^{\rho,*}(\mathbb{X}, \mathbb{Y}^1), \gamma^2 \in \Gamma_{\leq}^{\rho,*}(\mathbb{X}, \mathbb{Y}^2)$, we have:*

(1) *linear MPGW embedding of $\mathbb{Y}^1$ can recover MPGW discrepancy between $\mathbb{X}$ and $\mathbb{Y}^1$, i.e.*
$$\|k_{\gamma^*}\|_{\mu^{\otimes 2}}^2 = MPGW_\eta(x).$$
*In general, we have:*
$$LMPGW_\eta(\mathbb{X}, \mathbb{Y}; \gamma^0, \gamma^*) := MPGW_\eta(\mathbb{X}, \mathbb{Y}), \forall \gamma^0 \in \Gamma_{\leq}^{\eta,*}(\mathbb{X}, \mathbb{X}).$$

(2) *If Monge mapping assumption hold for $\gamma^1, \gamma^2$, then (106),(105) coincide.*

(3) *$\hat{\gamma}^* = (\mathrm{id} \times \mathcal{T}_{\gamma^*})_\# \mu$ is optimal for $MPGW_\eta(\mathbb{X}, \widetilde{\mathbb{Y}}_{\gamma^*}) = GW(\mathbb{X}, \widetilde{\mathbb{Y}}_{\gamma^*})$.*

(4) *When $\lambda$ is sufficiently large, we have:*
$$LMPGW_\eta(\mathbb{Y}^1, \mathbb{Y}^2; \mathbb{X}) = LPGW_\lambda(\mathbb{Y}^1, \mathbb{Y}^2; \mathbb{X}) - \lambda(|\nu^1|^2 + |\nu^2|^2 - 2\eta^2).$$
$$aLMPGW_\eta(\mathbb{Y}^1, \mathbb{Y}^2; \mathbb{X}) = aLPGW_\lambda(\mathbb{Y}^1, \mathbb{Y}^2; \mathbb{X}) - \lambda(|\nu^1|^2 + |\nu^2|^2 - 2\eta^2).$$

(5) *If $|\mu| = |\nu^1| = |\nu^2| = \eta = 1$, $LMPGW_\eta, LGW$ coincide and $aLMPGW_\eta, aLGW$ coincide.*

*Proof.*

(1) If Monge mapping assumption holds for $\gamma^*$, i.e. $\gamma^* = (\mathrm{id} \times T)_\# \mu$, we have:
$$MPGW_\eta(\mathbb{X}, \mathbb{Y}) = \int_{(X \times Y)^{\otimes 2}} |g_X(x, x') - g_Y(y, y')|^2 d(\gamma^*)^{\otimes 2}$$
$$= \|k_{\gamma^*}\|_{\mu^{\otimes 2}}^2.$$

Without Monge mapping assumption, pick $\gamma^0 \in \Gamma_{\leq}^{\eta,*}(\mathbb{X}, \mathbb{X})$, since $|\gamma^0| = \eta = |\mu|$, we have $\gamma^0 \in \Gamma(\mu, \mu)$. Thus
$$\int_{(X \times X)^{\otimes 2}} |g_X(x, x') - g_X(x^0, x^{0\prime})|^2 d((\gamma^0)^2)^{\otimes 2} = MPGW_\eta(\mathbb{X}, \mathbb{X}) = 0 = GW(\mathbb{X}, \mathbb{X}).$$

Therefore $\gamma^0 \in \Gamma^*(\mathbb{X}, \mathbb{X})$.

Pick $\gamma \in \Gamma_{\leq}^{\eta}(\gamma^0, \gamma^*) = \Gamma(\gamma^0, \gamma^*)$, by (49), we have
$$\int_{(X \times X \times Y)^{\otimes 2}} |g_X(x, x') - g_Y(y, y')|^2 d\gamma^{\otimes 2}$$
$$= \int_{(X \times Y)^{\otimes 2}} |g_X(x, x') - g_Y(y, y')|^2 d(\gamma^*)^{\otimes 2}$$
$$= MPGW_\eta(\mathbb{X}, \mathbb{Y}^1; \mathbb{X}, \gamma^0, \gamma^*)$$

It holds for all $\gamma \in \Gamma_{\leq}^{\eta}(\gamma^0, \gamma^*)$, thus
$$LMPGW_\eta(\mathbb{X}, \mathbb{Y}; \mathbb{X}, \gamma^0, \gamma^*) = MPGW_\eta(\mathbb{X}, \mathbb{Y}^*).$$

(2) Under Monge mapping assumption, we have $\gamma^1 = (\mathrm{id} \times T^1)_\# \mu', \gamma^2 = (\mathrm{id} \times T^2)_\# \mu''$ for some mappings $T^1, T^2$, where $\mu', \mu'' \leq \mu$. Since $|\mu'| = |\mu''| = \eta = |\mu|$, then we have $\mu'' = \mu' = \mu$. Thus,
$$\Gamma_{\leq}(\gamma^1, \gamma^2; \mu) = \{(\mathrm{id} \times T^1 \times T^2)_\# \mu', \mu' \leq \mu, |\mu'| = \eta\}$$
$$= \{(\mathrm{id} \times T^1 \times T^2)_\# \mu\}$$

and we have (106),(105) coincide.

(3) Since $\eta = |\mu|$, we have $|\widetilde{\nu}_{\gamma^1}| = |\gamma^*| = \eta = |\mu|$, thus

$$\Gamma_{\leq}^{\eta}(\mu, \widetilde{\nu}_{\gamma^1}) = \Gamma(\mu, \widetilde{\nu}_{\gamma^1}).$$

Thus, $MPGW_\eta(\mathbb{X}, \widetilde{\mathbb{Y}}_{\gamma^*}) = GW(\mathbb{X}, \widetilde{\mathbb{Y}}_{\gamma^*})$. Since $\gamma^*$ is optimal in $MPGW_\eta(\mathbb{X}, \mathbb{Y}) = GW(\mathbb{X}, \mathbb{Y})$, from Proposition 2.1, we have $\widetilde{\gamma}_{\gamma^1}$ is optimal for $GW(\mathbb{X}, \mathbb{Y}_{\gamma^1}) = MPGW_\eta(\mathbb{X}, \widetilde{\mathbb{Y}}_{\gamma^1})$ and we complete the proof.

(4) Since $X, Y^1, Y^2$ are compact, by Lemma E.2 in (Bai et al., 2024) when $\lambda$ is sufficiently large, for each $\gamma^1 \in \Gamma_{\leq,\lambda}^*(\mathbb{X}, \mathbb{Y}^1), \gamma^2 \in \Gamma_{\leq,\lambda}^*(\mathbb{X}, \mathbb{Y}^2)$, we have $|\gamma^1| = |\gamma^2| = \eta = |\mu|$. By Proposition M.1. (Bai et al., 2024), we have $\gamma^1, \gamma^2$ are optimal for $MPGW_\eta(\mathbb{X}, \mathbb{Y}^1), MPGW_\eta(\mathbb{X}, \mathbb{Y}^2)$. That is

$$\Gamma_{\leq,\lambda}^*(\mathbb{X}, \mathbb{Y}^1) \subset \Gamma_{\leq}^{\eta,*}(\mathbb{X}, \mathbb{Y}^1), \Gamma_{\leq,\lambda}^*(\mathbb{X}, \mathbb{Y}^2) \subset \Gamma_{\leq}^{\eta,*}(\mathbb{X}, \mathbb{Y}^2).$$

For the other direction, pick $\gamma^1 \in \Gamma_{\leq}^{\eta,*}(\mathbb{X}, \mathbb{Y}^1)$, $\gamma' \in \Gamma_{\leq,\lambda}^*(\mathbb{X}, \mathbb{Y}^1)$. Thus $|\gamma'| = \eta$, and we have

$$\begin{aligned}
&C(\gamma^1; \mathbb{X}, \mathbb{Y}^1, \lambda) \\
&= \int_{(X \times Y^1)^{\otimes 2}} |g_X(x, x') - g_{Y^1}(y^1, y^{1'})|^2 d(\gamma^1)^{\otimes 2} + \gamma(|\mu|^2 + |\nu^1|^2 - 2\eta^2) \\
&\leq \int_{(X \times Y^1)^{\otimes 2}} |g_X(x, x') - g_{Y^1}(y^1, y^{1'})|^2 d(\gamma')^{\otimes 2} + \gamma(|\mu|^2 + |\nu^1|^2 - 2\eta^2) \\
&= C(\gamma'; \mathbb{X}, \mathbb{Y}^1, \lambda) \\
&= PGW_\lambda(\mathbb{X}, \mathbb{Y}^1)
\end{aligned}$$

Thus, $\gamma^1 \in \Gamma_{\leq,\lambda}^*(\mathbb{X}, \mathbb{Y}^1)$.

Therefore,

$$\Gamma_{\leq,\lambda}^*(\mathbb{X}, \mathbb{Y}^1) = \Gamma_{\leq}^{\eta,*}(\mathbb{X}, \mathbb{Y}^1), \Gamma_{\leq,\lambda}^*(\mathbb{X}, \mathbb{Y}^2) = \Gamma_{\leq}^{\eta,*}(\mathbb{X}, \mathbb{Y}^2).$$

Pick $\gamma^1 \in \Gamma_{\leq,\lambda}^*(\mathbb{X}, \mathbb{Y}^1) = \Gamma_{\leq}^{\eta,*}(\mathbb{X}, \mathbb{Y}^1), \gamma^2 \in \Gamma_{\leq,\lambda}^*(\mathbb{X}, \mathbb{Y}^2) = \Gamma_{\leq}^{\eta,*}(\mathbb{X}, \mathbb{Y}^2)$, we have

$$MLPGW_\eta(\mathbb{Y}^1, \mathbb{Y}^2; \mathbb{X}, \gamma^1, \gamma^2) = LPGW_\lambda(\mathbb{Y}^1, \mathbb{Y}^2; \mathbb{X}, \gamma^1, \gamma^2) - \lambda(|\nu^1| + |\nu^2| - 2\eta^2)$$

Take the infimum over all $\gamma^1, \gamma^2$, we obtain

$$MLPGW_\eta(\mathbb{Y}^1, \mathbb{Y}^2; \mathbb{X}) = LPGW_\lambda(\mathbb{Y}^1, \mathbb{Y}^2; \mathbb{X}) - \lambda(|\nu^1|^2 + |\nu^2|^2 - 2\eta^2)$$

Similarly, we have

$$aMLPGW_\eta(\mathbb{Y}^1, \mathbb{Y}^2; \mathbb{X}) = aLPGW_\lambda(\mathbb{Y}^1, \mathbb{Y}^2; \mathbb{X}) - \lambda(|\nu^1|^2 + |\nu^2|^2 - 2\eta^2) \quad (108)$$

(5) In this case, we have $\Gamma_{\leq}^{\eta}(\mu, \nu^1) = \Gamma(\mu, \nu^1), \Gamma_{\leq}^{\eta}(\mu, \nu^2 = \Gamma(\mu, \nu^2)$. Thus

$$\Gamma_{\leq}^{\eta,*}(\mathbb{X}, \mathbb{Y}^1) = \Gamma^*(\mathbb{X}, \mathbb{Y}^1), \Gamma_{\leq}^{\eta,*}(\mathbb{X}, \mathbb{Y}^2) = \Gamma^*(\mathbb{X}, \mathbb{Y}^2).$$

Thus $LMPGW, LGW$ coincide, $aLMPGW, aLGW$ coincide.

$\square$

## J  NUMERICAL IMPLEMENTATION OF LOT, LGW, LPGW DISTANCE.

**LOT distance**  In previous sections, we introduce LOT distance (31), aLOT distance (34) and its approximation formulation (35). Their relationship can be described as follows:

- LOT distance (31) is proposed to approximate OT distance.
- aLOT distance (34) is proposed to approxiamte LOT distance.

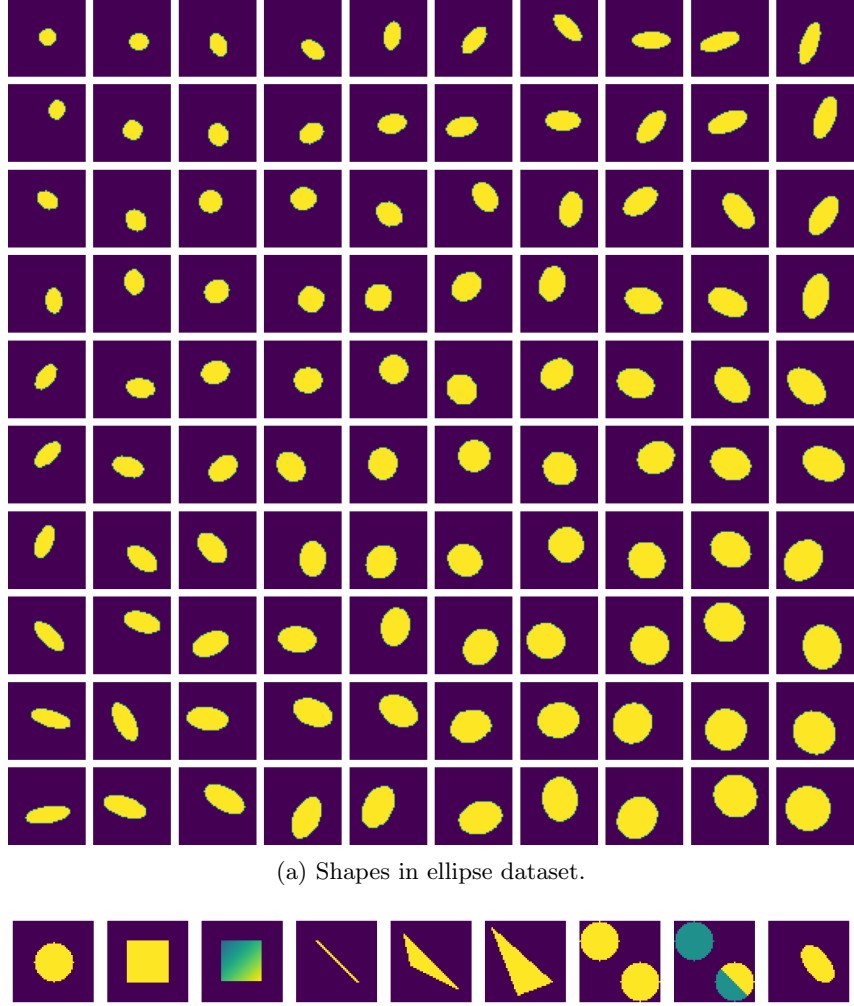

(a) Shapes in ellipse dataset.

(b) Reference spaces.

Figure 2: In the first figure, we visualize the ellipse dataset (Beier et al., 2022). For each ellipse shape $X \subset \mathbb{R}^{n \times 2}$, we normalize the scaling of the shape such that $\max_{i,i' \in [1:n]} \|X[i,:] - X[i',:]\| = 1$. The sizes of these shapes range from 90 to 650. In the second figure, we visualize the reference spaces. In each shape, the color represents the value of the probability mass at the corresponding location.

- Formulation (35) is proposed to approximate aLOT distance.

However, in practice, it is not a multi-layer approximation. The distance LOT (31) and the distance LOT (34) are only proposed for theoretical completeness and are not computationally feasible. In practice, formulation (35),

$$\|\mathcal{T}_{\gamma^1} - \mathcal{T}_{\gamma^2}\|_{L(\mu)},$$

is used to approximate $OT$ distance between $\nu^1$ and $\nu^2$ and in most of the reference which cite (Wang et al., 2013), (34) is refereed as LOT distance, the original formulation (31),(34) are not mentioned.

A similar convention is adapted to LGW and LPGW.

**LGW distance** The relation between LGW distance (39), aLGW distance (10) and its approximation formulation (11) can be described as follows:

- LGW distance (39) is proposed to approximate GW distance.
- aLGW distance (10) is proposed to approximate LGW distance.
- Formulation (11) is proposed to approximate aLGW distance.

Similarly, it is not a multi-layer approximation. LGW distance (39) and aLGW distance (??) are only proposed for theoretical completeness and are not computationally feasible. In practice, formulation (11),

$$|g_{Y^1}(\mathcal{T}_{\gamma^1}(\cdot_1), \mathcal{T}_{\gamma^2}(\cdot_2)) - g_{Y^2}(\mathcal{T}_{\gamma^2}(\cdot_1), \mathcal{T}_{\gamma^2}(\cdot_2))|^2_{L(\mu^{\otimes 2})},$$

is used to approximate $GW$ distance.

**LPGW distance** The relation between LPGW distance (16), aLGW distance (20) and its approximation formulation (22) can be described as follows:

- LPGW distance (16) is proposed to approximate PGW distance.
- aLGW distance (20) is proposed to approximate LPGW distance.
- Formulation (22) is proposed to approximate aLPGW distance.

Similarly, it is not a multi-layer approximation. LPGW distance (16) and aLGW distance (20) are only proposed for theoretical completeness and are not computationally feasible. In practice, formulation (22),

$$|g_{Y^1}(\mathcal{T}_{\gamma^1}(\cdot_1), \mathcal{T}_{\gamma^2}(\cdot_2)) - g_{Y^2}(\mathcal{T}_{\gamma^2}(\cdot_1), \mathcal{T}_{\gamma^2}(\cdot_2))|^2_{L((\gamma_X^1 \wedge \gamma_X^2)^{\otimes 2})} + \lambda(|\nu|^1 + |\nu^2|^2 - 2|\gamma_X^1 \wedge \gamma_X^2|^2)$$

is used to approximate PGW distance.

## K    DETAILS OF ELLIPTICAL DISKS EXPERIMENT

In this section, we present the details of the elliptical disks experiment.

**Dataset and numerical setting details.** The dataset we used is the ellipse dataset given by (Beier et al., 2022), which consists of 100 distinct ellipses. Each ellipse is represented as an $n \times 2$ matrix, where $n$ ranges from 90 to 600. For reference shapes, we selected 9 different 2D shapes, including disks, squares, triangles, and others. See Figure 2 for a visualization of the dataset and the reference spaces. In this experiment, each shape is modeled with an empirical measure $\sum_{i=1}^{n} \frac{1}{n} \delta_{x_i}$.

**Performance analysis**

We present the results in Table 4. As mentioned in the main text, LPGW is significantly faster than PGW, as it requires only $N = 100$ PGW computations, while PGW requires $\binom{N}{2}$ computations. Furthermore, we observe that for some reference spaces (e.g., $\mathbb{S}_5$, $\mathbb{S}_7$,

| $\lambda$ | points | PGW | $\mathbb{S}_1$ | $\mathbb{S}_2$ | $\mathbb{S}_3$ | $\mathbb{S}_4$ | $\mathbb{S}_5$ | $\mathbb{S}_6$ | $\mathbb{S}_7$ | $\mathbb{S}_8$ | $\mathbb{S}_9$ |
|---|---|---|---|---|---|---|---|---|---|---|---|
| | | | 441 | 676 | 625 | 52 | 289 | 545 | 882 | 882 | 317 |
| 0.05 | time (mins) | 45.34 | 0.86 | 3.8 | 3.08 | 0.66 | 0.63 | 1.43 | 1.89 | 2.1 | 0.69 |
| | MRE | — | 0.1982 | 0.1263 | 0.1428 | 0.6315 | 0.4732 | 0.1454 | **0.0394** | 0.0793 | **0.0245** |
| | PCC | — | 0.5774 | 0.5741 | 0.5884 | 0.5584 | 0.6514 | 0.7698 | **0.9303** | 0.8711 | **0.9944** |
| 0.08 | time (mins) | 43.86 | 1.02 | 3.33 | 3.55 | 0.34 | 0.99 | 1.25 | 1.29 | 1.55 | 1.22 |
| | MRE | — | 0.1941 | 0.1264 | 0.1431 | 0.2542 | 0.0705 | 0.0444 | **0.0205** | **0.0198** | 0.0245 |
| | PCC | — | 0.5781 | 0.5738 | 0.5881 | 0.8581 | 0.8741 | 0.993 | 0.9952 | **0.9954** | **0.9949** |
| 0.1 | time (mins) | 46.97 | 0.76 | 3.78 | 3.13 | 0.08 | 0.62 | 1.56 | 1.91 | 1.98 | 0.71 |
| | MRE | — | 0.1941 | 0.1264 | 0.1431 | 0.2542 | 0.0538 | 0.0444 | **0.0205** | **0.0198** | 0.0245 |
| | PCC | — | 0.5781 | 0.5738 | 0.5881 | 0.8581 | 0.9871 | 0.993 | **0.9952** | **0.9954** | 0.9949 |
| 0.5 | time (mins) | 44.77 | 0.74 | 3.68 | 3.02 | 0.08 | 0.61 | 1.51 | 1.89 | 1.98 | 0.69 |
| | MRE | — | 0.1932 | 0.1262 | 0.1428 | 0.2542 | 0.0538 | 0.0443 | **0.0205** | **0.0198** | 0.0246 |
| | PCC | — | 0.5779 | 0.5737 | 0.5879 | 0.8583 | 0.9871 | 0.993 | **0.9952** | **0.9953** | 0.9949 |

Table 4: In the first column, the values $0.05, 0.08, 0.1, 0.5$, represent the selected $\lambda$ values. For each $\lambda$, the first row shows the wall-clock time for PGW and LPGW. The second and third rows display the MRE (mean relative error) and PCC (Pearson correlation coefficient), respectively.

$\mathbb{S}_9$), the MRE is relatively lower. Moreover, for most reference spaces, including $\mathbb{S}_7$, $\mathbb{S}_8$, and $\mathbb{S}_9$, LPGW admits a PCC greater than 0.85. Finally, when $\lambda$ is larger, the PCC tends to be higher, and the MRE is lower across all reference spaces, as discussed further in the next section. These results highlight the importance of the choice of reference space, as is commonly the case for linear OT-based methods.

**Relative error analysis** Given $\nu^1, \nu^2, \ldots, \nu^K$ and reference measure $\mu$, the relative error is defined as:

$$\text{MRE} = \frac{1}{\binom{K}{2}} \sum_{i \neq j} \frac{|PGW(\nu^i, \nu^j) - LPGW(\nu^i, \nu^j; \mu)|}{PGW(\nu^i, \nu^j)}. \tag{109}$$

*Remark* K.1. For numerical stability, when $\lambda$ is small (i.e., $\lambda = 0.05, 0.08$), we remove the PGW/LPGW distance whenever $PGW \leq 1 \cdot 10^{-10}$ since $1 \cdot 10^{-10}$ is the tolerance in the PGW algorithm. In this case, $PGW \approx 1 \cdot 10^{-10}$ and $LPGW \approx 1 \cdot 10^{-11}$ which renders the relative error uninformative.

We decompose this error into the following four aspects:

- The transportation plan induced by LPGW may not necessarily be the optimal transportation plan for the PGW problem.
- In practice, we use the barycentric projected measure $\hat{\nu}^i$ to approximate $\nu^i$ for each $i$. These two measures can be distinct, especially when the optimal transportation plan is not generated by the Monge mapping.
- The solvers for both PGW and LPGW rely on the Frank-Wolfe algorithm. Due to the non-convexity of these problems, the Frank-Wolfe algorithm may yield a local minimizer instead of a global one. Therefore, the computed PGW transportation plan might not be optimal.
- In practice, we approximate the original LPGW distance (16) (or (10)) using the approximation formulation (22). This introduces a gap between the real LPGW distance and the approximation.

It is important to note that the first issue arises from a theoretical perspective, while the remaining aspects are due to the numerical implementation. The first two errors are also present in other linear OT-based methods (Wang et al., 2013; Cai et al., 2022; Bai et al., 2023; Beier et al., 2022). The third issue stems from the non-convexity of GW/PGW, affecting both LGW and LPGW similarly. The final error is specific to LPGW due to the approximation formulation used.

In practice, several methods can be employed to reduce these errors. For example, the dataset can be represented as empirical measures with equal mass at each point. This approach is effective for unbalanced linear OT techniques (e.g., (Bai et al., 2023; Cai et al., 2022)), as these methods do not require mass normalization. With high probability, this will result in a Monge mapping, reducing the second error. Additionally, to minimize the last error, as stated in Theorem 3.2, using a higher $\lambda$ leads to a lower error, and when $\lambda$ is sufficiently large, this error becomes zero.

**MDS visualization and analysis.** We visualize the multi-dimensional scaling (MDS) embeddings for both PGW and LPGW with respect to each reference space in Figure 3. We observe that when $\lambda$ is large (i.e. $\lambda = 0.1, 0.5$), LPGW with reference space $\mathbb{S}_5, \mathbb{S}_6, \mathbb{S}_7, \mathbb{S}_8, \mathbb{S}_9$ admit similar patterns. When $\lambda = 0.05$ or $\lambda = 0.08$, we observe $LPGW$ with reference $\mathbb{S}_7, \mathbb{S}_8, \mathbb{S}_9$ and PGW admit similar patterns. From this figure, we observe that $\mathbb{S}_7, \mathbb{S}_8, \mathbb{S}_9$ admit better performance than other reference spaces.

**Summary.** Although LPGW methods are subject to potential errors, the high PCC observed between LPGW and PGW (when, e.g., the reference space is $\mathbb{S}_7, \mathbb{S}_8, \mathbb{S}_9$), suggests that LPGW can serve as a good proxy for PGW, rather than simply as an approximation.

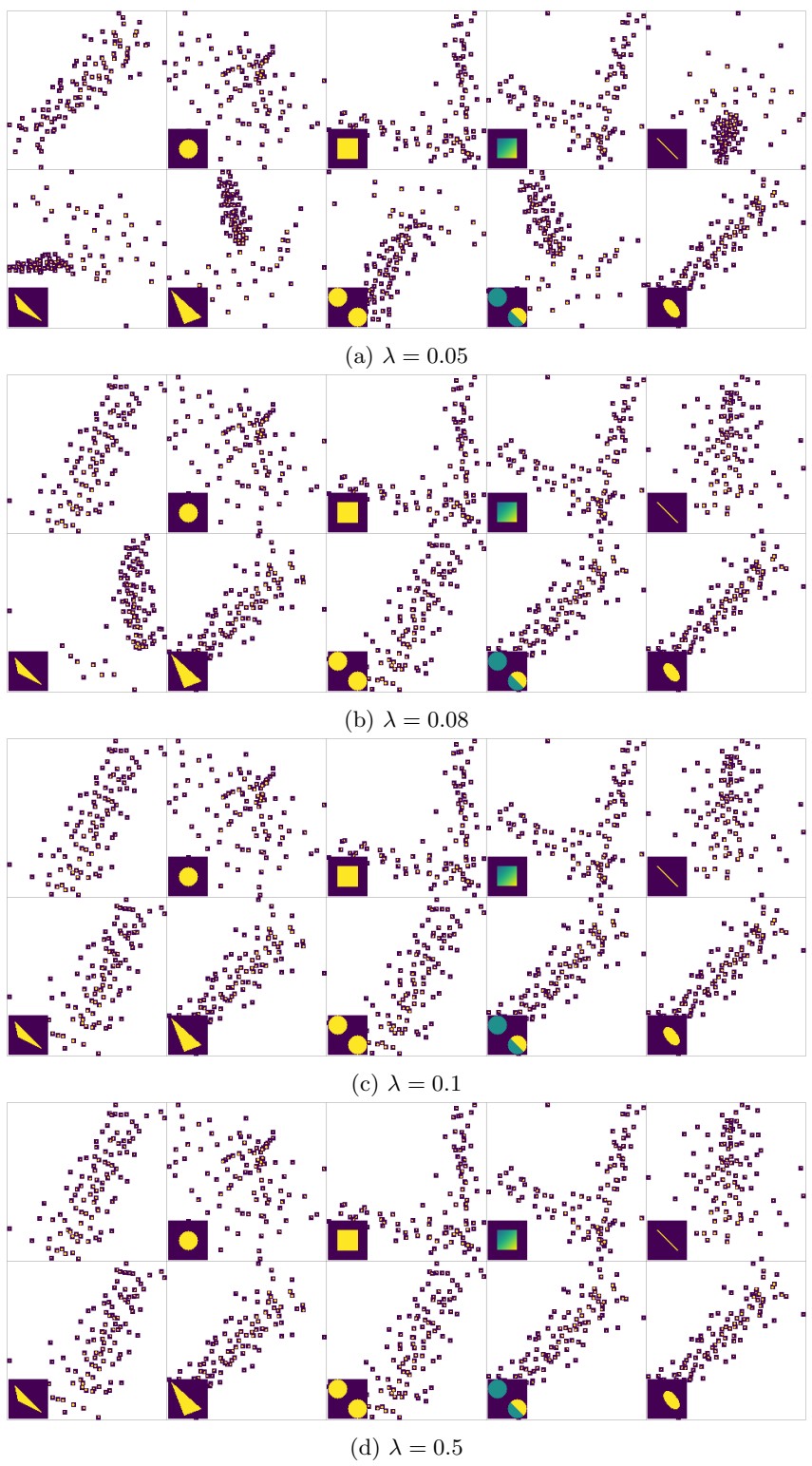

(a) $\lambda = 0.05$

(b) $\lambda = 0.08$

(c) $\lambda = 0.1$

(d) $\lambda = 0.5$

Figure 3: MDS visualization for $\lambda = 0.05, 0.08, 0.10, 0.50$. Each subfigure shows the MDS visualization for PGW and LPGW based on different reference spaces. In each figure, the first subfigure in the first row is the MDS visualization of PGW.

## L  DETAILS OF SHAPE RETRIEVAL EXPERIMENT

**Dataset details.**  We test two datasets in this experiment: a 2D dataset and a 3D dataset. The visualization of the two datasets is presented in Figure 4.

In addition, for each shape $X = \{x_1, \ldots x_n\}$, we normalize the scaling such that $\max_{i \neq j} \|x_i - x_j\| = 1$. For the 3D dataset, we apply the $k$-means clustering to reduce the size of each shape to 500, from its original size 2048.

**Numerical details.**  We represent the shapes in each dataset as mm-spaces $\mathbb{X}^i = \left(\mathbb{R}^d, \|\cdot\|_2, \mu^i = \sum_{k=1}^{n^i} \alpha^i \delta_{x_k^i}\right)$. We use $\alpha^i = \frac{1}{n^i}$ to compute the GW/LGW distances for the balanced mass constraint setting.

For the PGW/LPGW distances, we set $\alpha = \frac{1}{N}$, where $N$ is the median number of points across all shapes in the dataset.

For the SVM experiments, we use $\exp(-\sigma D)$ as the kernel for the SVM model.

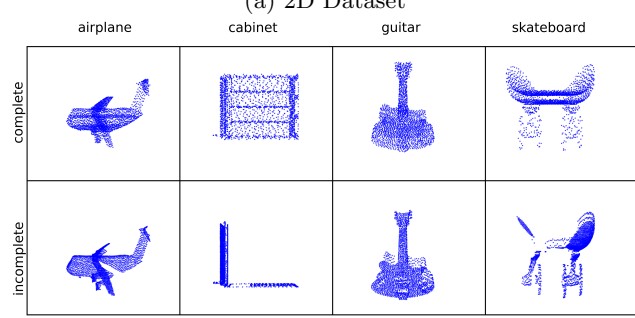

(a) 2D Dataset

(b) 3D Dataset

Figure 4: Datasets for shape retrieval experiment.

Here, we normalize the matrix $D$ and choose the best $\sigma \in \{0.001, 0.01, 0.1, 1, 5, 8, 10, 100\}$ for each method used in order to facilitate a fair comparison of the resulting performances. We note that the resulting kernel matrix is not necessarily positive semidefinite.

**Parameter selection for PGW and LPGW.**  The parameter $\lambda$ for PGW and LPGW is chosen as follows. For the 2D dataset, we set $\lambda$ such that $\lambda \leq \lambda_{max} = \frac{1}{2} \max_i (|C^i|^2) = 0.5$, since all shapes are normalized to a scale of 1. We perform a line search for $\lambda$ over the set $\{0.2, 0.3, 0.5\}$ and select $\lambda = 0.2$ for both PGW and LPGW.

For the 3D dataset, we randomly select 2-3 shapes and compute the PGW distance between these shapes and the reference space. We find that $\lambda_{max} \approx 0.15$. Thus, we conduct a line search for $\lambda$ over the range $\{0.02, 0.03, 0.05, 0.06, 0.070.08, 0.080.1, 0.15\}$ and choose $\lambda = 0.06$ for both PGW and LPGW.

**Reference space setting for LGW and LPGW.**  Similar to the classical OT method (Wang et al., 2013), the ideal reference space should represent the "center" of the tested measures. In practice, we typically use the GW barycenter (Peyré et al., 2016) or the PGW barycenter (Bai et al., 2024).

For each dataset, we randomly select one point cloud from each class. Specifically, for the 2D dataset, we select 8 shapes, and for the 3D dataset, we select 4 shapes. We then compute the GW/PGW barycenter between these selected shapes. The wall-clock time for this computation on the 2D dataset is about 10 seconds, and for the 3D dataset, it is about 5 minutes. For fairness in comparison, we apply the same reference space for both the LGW and LPGW methods in all experiments.

**Nearest neighbor classification.**  Once the pairwise distance matrices have been computed between the shapes using each method, we use the computed distances for a nearest-

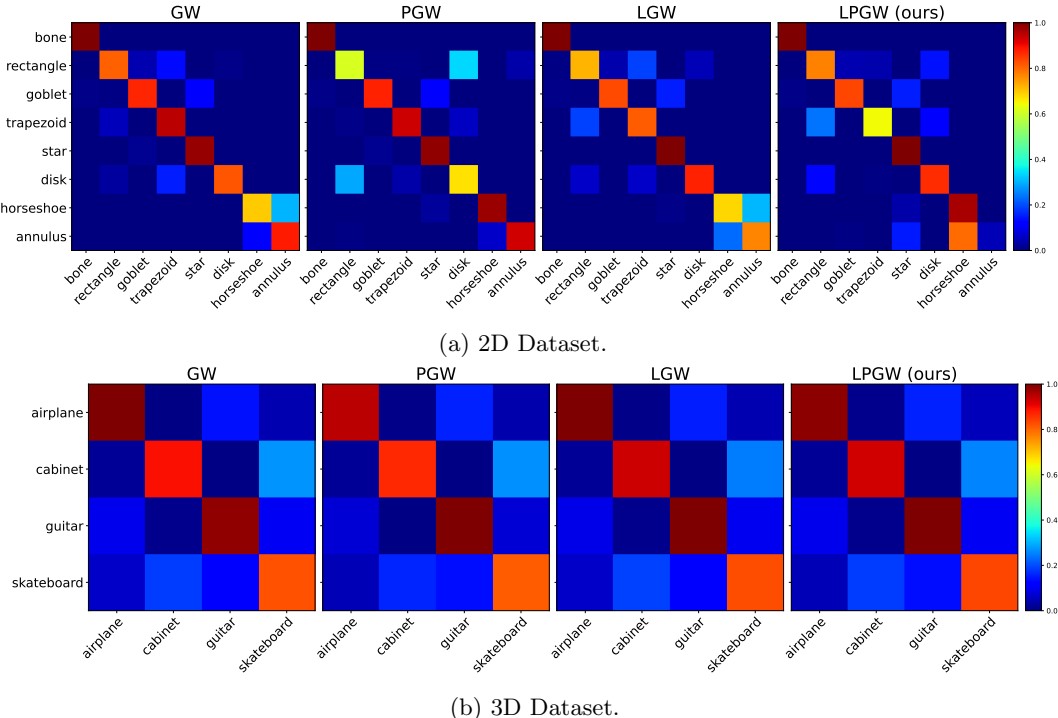

(a) 2D Dataset.

(b) 3D Dataset.

Figure 5: Confusion matrices computed from nearest neighbor classification experiments.

neighbor classification experiment. We choose a single representative at random from each class in the dataset and classify each shape according to its nearest representative. This is repeated over 10,000 iterations, and we generate a confusion matrix for each distance used.

**Performance analysis.** The pairwise distance matrices are visualized for each dataset in Figure 7, and the confusion matrices from the nearest neighbor classification experiment on each dataset are shown in Figure 5. Finally, the classification accuracy with the SVM experiments is reported in Table 2. The results indicate that the LPGW distance is able to obtain high performance across both data sets consistently.

From Figure 7, we observe that PGW and LPGW qualitatively admit a more reasonable similarity measure compared to the other considered methods. For example, on the 2D dataset, class "bone" and "rectangle" should have a relatively smaller distance than "bone" and "annulus". Ideally, a reasonable distance should satisfy the following: $0 < d(\text{bone}, \text{rectangle}) < d(\text{bone}, \text{anulus})$. However, we do not observe this relation in GW or LGW.

In the 3D dataset, the main challenge is the presence of incomplete shapes, which represent only a part of the corresponding complete shape and can have a large dissimilarity to each other. In this experiment, we observe that PGW/LPGW admits slightly better performance. We suspect due the the unbalanced setting of these two methods, these two methods are more robust to the incomplete shape classification.

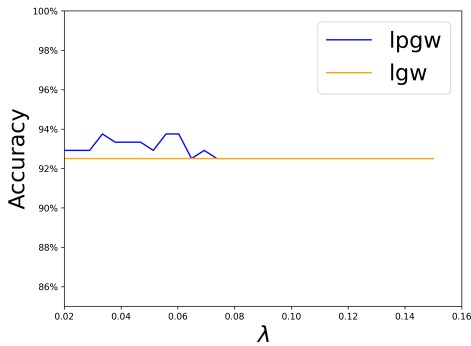

Figure 6: We visualize the accuracy of LPGW for different $\lambda$. For each $\lambda$, we select the best $\sigma$ to compute the accuracy. We observe that when $\lambda \geq 0.08$, LPGW and LGW have the same accuracy.

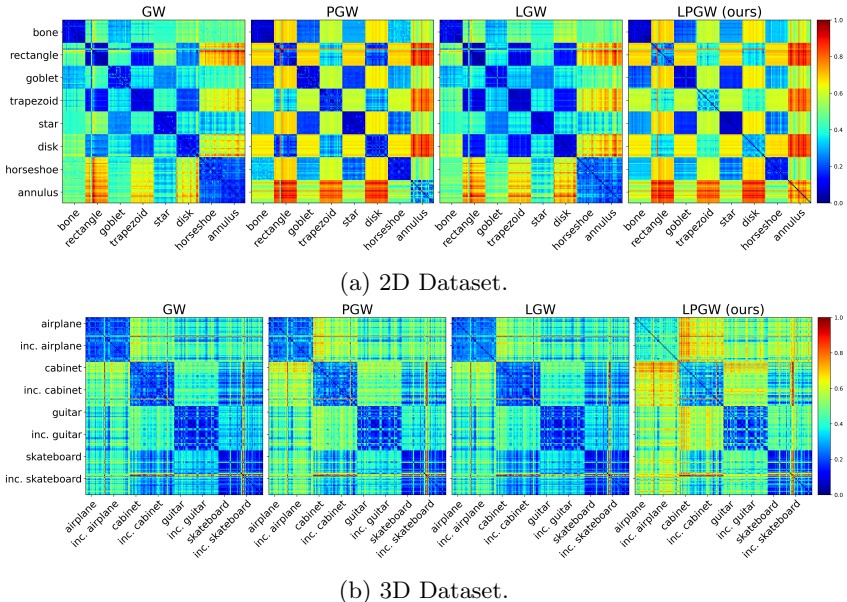

(a) 2D Dataset.

(b) 3D Dataset.

Figure 7: Visualization of pairwise distance matrices resulting from each of the considered methods: GW, PGW, LGW, and LPGW (ours). In (b), the label "inc. airplane" denotes the incomplete airplane shape class, and similarly for each of the other 3 such classes.

Note that in this dataset all shapes have the same size. Thus, when $\lambda$ is sufficiently large (i.e. $\lambda = 0.5$), LPGW can recover the performance of LGW. We refer to Figure 6 for visualization.

## M  Details of Learning with Transform-Based Embeddings Experiment

**Reference space.**  Similar to the shape retrieval experiment, for each class/digit, we select one shape from the training set and compute the GW barycenter based on the selected shapes. Note, in this step, the reference space we obtain is $(M_0 \in \mathbb{R}^{n_0 \times n_0}, p^0)$, where $n_0 \in \mathbb{N}$ is the size of the reference space, $M_0$ denotes the pairwise distance matrix, and $p^0$ is the PMF of the reference measure. Here, $n_0$, $p^0$ are inputs to the GW barycenter algorithm, and in this experiment, we set $n_0$ to be the mean of the sizes of the selected shapes, and $p^0$ is set to be $p^0 = \frac{1}{n_0} 1_{n_0}$.

However, $M_0$ cannot be applied as reference space for the LOT method. Thus, we apply the MDS method for $M_0$ and obtain the supported points $X^0 = \{x_1^0, \ldots x_{n_0}^0\}$ of the reference space, i.e.,

$$X = \arg\min_X \sum_{i=1}^{n} \|(M_0)_{i,j} - \|x_i - x_j\|\|^2, \text{where } x_i = X[i,:], \forall i.$$

We use $(M_0, p^0, \sum_{i=1}^{n} x_i)$ as the reference space for LOT/LGW/LPGW. The wall-clock time of barycenter computation is 26 seconds.

**Numerical details.**  Note, since the goal of this experiment is to test the performance of learning from embeddings under corruption by random rotation/flipping and noise, we remove digits $\{5, 9\}$ from the data since the rotated and flipped digit "2" is highly similar to "5". Similarly, rotated "6" is nearly identical to "9".

The classification model we selected is the logistic regression model as provided by the scikit-learn package. For the LOT and LGW methods, due to the balanced mass requirement, we

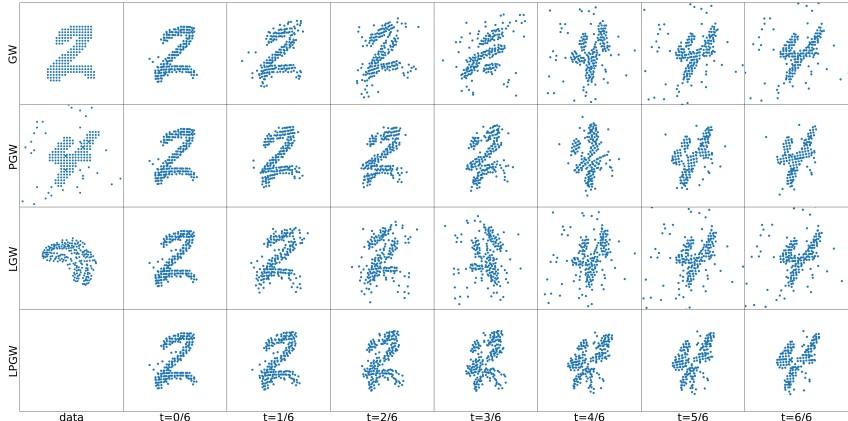

Figure 8: We visualize the interpolation between two shapes using GW, PGW, LGW, and LPGW. In the first column, the three shapes shown are the source shape, the target shape, and the reference space used in LGW and LPGW. The target shape, digit "4", is corrupted by the addition of noise points, with a total mass of $\eta = 0.3$.

normalize the PMF of all shapes in the testing dataset. For LPGW, this normalization is not applied.

**Parameter setting for LPGW.** Note that in this case, the training data is not corrupted. Thus, we can set $\lambda = \lambda_{max} = \frac{1}{2} \max_i C_i^2$ where $C_i$ is the cost matrix for shape $i$. In this experiment, we set $\lambda = 40$. In addition, to improve the computational speed of LGW/LPGW, the augmented training data obtained by rotation and flipping is NOT contained in these two methods since the rotated shape is equivalent to the original shape in the setting of GW/PGW.

**Performance analysis.** The LOT method achieves 51.3% accuracy when $\eta = 0$. This indicates that even with the addition of rotated/flipped data to the training set, the LOT embedding method struggles to classify the rotated digits with high accuracy. In contrast, LGW/LPGW achieve 82.5% accuracy, indicating that these two embedding techniques are more robust to corruption by rotation/flipping.

When $\eta \geq 0.1$, the accuracy of both LOT and LGW drops to 10-20%. However, we observe that the LPGW embedding maintains a strong accuracy between 70% and 85%. This demonstrates that the LPGW embedding is more robust to corrupted test data.

M.1    TOY EXAMPLE: POINT CLOUD INTERPOLATION

In this subsection, we select one shape from the training data and a noise-corrupted shape from the testing data (with rotation and flipping removed). We demonstrate the interpolation between these two shapes using the GW barycenter (Peyré et al., 2016), PGW barycenter (Bai et al., 2024), LGW geodesic (Beier et al., 2022), and our LPGW interpolation. The goal is to intuitively visualize the LGW/LPGW embeddings for the reader's understanding.

**Our method and baseline methods.** Let $\mathbb{X}$ be the reference mm-space applied in the classification experiment. Note, numerically, it can be described by $(C^0, X^0, p^0)$, where $p^0 \in \mathbb{R}_+^{n_0}$ is the PMF; $n_0$ is the size of $p^0$; $X^0 \in \mathbb{R}^{n_0 \times 2}$ is the set of 2D supported points; and $C^0 = [\|X_i^0 - X_j^0\|_2^2]_{i,j \in [1:n_0]}$ is the corresponding cost matrix.

That is, we can use $(C^0, p^0)$ to represent $\mathbb{X}$[3]. Similarly, we use $(C^1, p^1)$ and $(C^2, p^2)$ to represent the source and target mm-spaces (shapes).

---

[3]$X^0$ is not required as $C^0$ contains all the information of $X^0$ in the GW/PGW problem.

Now we introduce the **GW** barycenter method. For each time $t \in \{0, 1/6, \ldots, 6/6 = 1\}$, we solve the barycenter problem

$$C^* = \underset{C \in \mathbb{R}^{n_0 \times n_0}}{\arg\min} \left( (1-t) \, GW((C, p^0), (C^1, p^1)) + t \, GW((C, p^0), (C^2, p^2)) \right).$$

When $t = 0$, $(C^*, p^0)$ and $(C^1, p^1)$ represent similar shapes. In fact, if $p^0 = p^1$, the two shapes are identical. Similarly, if $t = 1$, $(C^*, p^1)$ is similar to the target $(C^2, p^2)$. For $t \in (0, 1)$, $(C^*, p^0)$ represents an interpolation shape between the source shape $(C^1, p^1)$ and the target shape $(C^2, p^2)$.

The **PGW** barycenter method is defined similarly.

In the **LGW** geodesic method, let $\sum_{i=1}^{n} \delta_{\hat{y}_i^1} p_i^0$ denote the barycentric projection obtained by $\gamma$, where $\gamma$ is the optimal transportation plan for the GW problem between $(C^0, p^0)$ and $(C^1, p^1)$. The numerical implementation of the LGW embedding is given by

$$E^1 = C^0 - [\|\hat{y}_i^1 - \hat{y}_j^1\|^2]_{i,j \in [1:n_0]} \in \mathbb{R}^{n_0 \times n_0}$$

(we refer to (5) for the original formulation). Similarly, we can define the LGW embedding for the target shape $(C^2, p^2)$, denoted as $E_2 \in \mathbb{R}^{n_0 \times n_0}$.

Then for each $t$, the interpolation/geodesic of LGW is defined by

$$C^0 + (1-t)E_1 + tE_2. \tag{110}$$

For the **LPGW** interpolation method, the numerical LPGW embeddings $E_1, E_2 \in \mathbb{R}^{n_0 \times n_0}$ are defined similarly, and we refer to (24) for details. We use (110) for the interpolation.

To visualize these interpolations, for each $C_t$, we adapt the MDS method, and the solution $X_t \in \mathbb{R}^{n_0 \times 2}$ is a point cloud. We visualize these point clouds in Figure 8.

**Performance comparison.** In this experiment, the size of these shapes is in the range of 200–250. GW/PGW requires 230–270 seconds, while LGW/LPGW requires 1–2 seconds. In Figure 8, we observe that the shapes generated by GW/LGW contain more noise points. Additionally, at times such as $t = 3/6$ and $t = 4/6$, the generated shapes are difficult to distinguish due to the noise points.

In contrast, the shapes generated by the PGW/LPGW methods contain significantly fewer noise points.

Note that at $t = 0$ and $t = 1$, the shapes generated by the LGW/LPGW methods can be treated as visualizations of the corresponding embeddings for the source shape $(C^1, p^1)$ and target shape $(C^2, p^2)$, respectively.

## M.2 OTHER BASELINES OT/GW/PGW

Note that OT, GW, and PGW methods cannot be directly applied to this experiment since we use a linear model (logistic regression) as the classifier. However, we can adapt the kernel-SVM methods described in the shape retrieval experiment for this classification task as well. The primary distinction between these three methods and LOT, LGW, and LPGW lies in their computational complexity.

Suppose $N_{\text{train}}$ and $N_{\text{test}}$ represent the sizes of the training and testing datasets, respectively. Let $n$ and $d$ denote the average size and dimension of all the point clouds or digits. Additionally, let $\mathcal{N}$ represent the average number of iterations required by the FW algorithms for GW and PGW.

The computational cost of these methods is summarized as follows:

In methods OT and LOT, the term $(n^3 + n^2 d)$ refers to the complexity of computing one OT distance, while $nd$ represents the complexity of computing one LOT distance given two embeddings.

| Method | Training Process | Testing Process |
|--------|------------------|-----------------|
| OT | $\mathcal{O}(N_{\text{train}}^2 \cdot (n^3 + n^2 d))$ | $\mathcal{O}(N_{\text{train}} N_{\text{test}}(n^3 + n^2 d))$ |
| LOT | $\mathcal{O}(N_{\text{train}} \cdot (n^3 + n^2 d))$ | $\mathcal{O}(N_{\text{test}} \cdot (n^3 + n^2 d) + N_{\text{test}} N_{\text{train}} \cdot (nd))$ |
| GW | $\mathcal{O}(N_{\text{train}}^2 \cdot (n^3 \mathcal{N} + n^2 d))$ | $\mathcal{O}(N_{\text{test}} N_{\text{train}} \cdot (n^3 \mathcal{N} + n^2 d))$ |
| LGW | $\mathcal{O}(N_{\text{train}}(n^3 \mathcal{N} + n^2 d))$ | $\mathcal{O}(N_{\text{test}}(n^3 \mathcal{N} + n^2 d) + N_{N_{\text{test}}\text{train}} \cdot (n^2))$ |
| PGW | $\mathcal{O}(N_{\text{train}}^2 \cdot (n^3 \mathcal{N} + n^2 d))$ | $\mathcal{O}(N_{\text{test}} N_{\text{train}} \cdot (n^3 \mathcal{N} + n^2 d))$ |
| LPGW | $\mathcal{O}(N_{\text{train}}(n^3 \mathcal{N} + n^2 d))$ | $\mathcal{O}(N_{\text{test}}(n^3 \mathcal{N} + n^2 d) + N_{\text{test}} N_{\text{train}}(n^2))$ |

Table 5: Computational complexity of different methods.

Similarly, in methods GW and LGW, the term $(n^3 \mathcal{N} + n^2 d)$ refers to the computational complexity of computing one GW distance, and the term $(n^2)$ represents the complexity of computing one LGW distance given two embeddings.

The complexities for PGW and LPGW follow a similar pattern.

In this experiment, $N_{\text{train}} = 8000$ and $N_{\text{test}} = 1000$. OT-based methods require approximately 50 hours (2–3 days) to train the model, while GW and PGW require about 400 hours (over 15 days). In contrast, LOT, LGW, and LPGW take only 3–5 minutes.

## N    COMPLEMENTAL RESULTS MNIST CLASSIFICATION EXPERIMENT

**Data reconstruction.** We visualize the reconstructed digits using LOT, LGW, and LPGW embeddings under two settings: $\eta = 0$ and $\eta = 0.2$ (See figure 1b). When the data is not corrupted by noise points ($\eta = 0$), the reconstructed digits from all methods closely resemble the original data. However, when 20% of noise points are added ($\eta = 0.2$), LPGW's reconstructed digits effectively exclude most of the noise points. In contrast, the embeddings produced by OT and LGW retain information from the noisy data. This demonstrates that the LPGW embedding leverages the partial matching property of PGW, ensuring that most of the noise points are excluded during the embedding process. This explains the robustness of LPGW embeddings to noise corruption.

**t-SNE.** From the figure 9, we can observe that when $\eta \geq 0.1$, LPGW embeddings show greater separability than those of LOT or LGW.

## O    COMPUTE RESOURCES

All experiments presented in this paper are conducted on a computational machine with an AMD EPYC 7713 64-Core Processor, $8 \times 32$GB DIMM DDR4, 3200 MHz, and an NVIDIA RTX A6000 GPU.

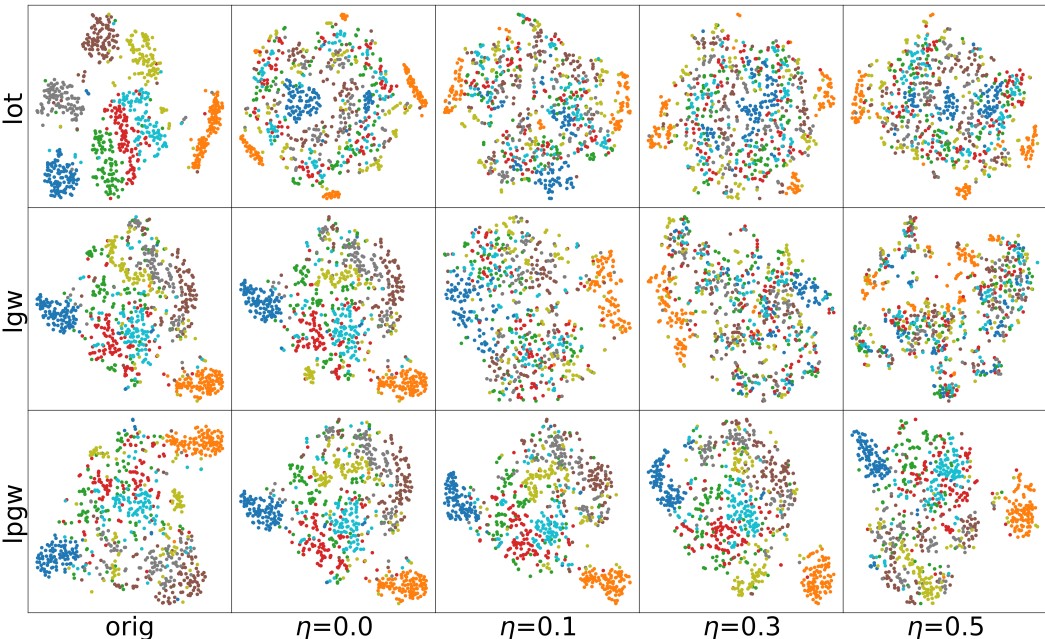

Figure 9: t-SNE visualization of embeddings. In the first column, the label "orig" indicates the experimental results for the original testing dataset without random rotations/flips or noise. In the remaining columns, random rotations/flips are applied to the testing data, and $\eta$ denotes the total mass of noise points.

