# OpenReview forum: "Linear Partial Gromov-Wasserstein Embedding"
_ICLR.cc/2025/Conference — ICLR 2025 Poster_

### Official Review · Reviewer_gpEb · 2024-10-26

**Soundness:** 3
**Presentation:** 3
**Contribution:** 4
**Rating:** 8
**Confidence:** 3

**Summary:**

The authors of this paper aim to improve on the partial Gromov-Wasserstein (PGW) method by creating linear partial Gromov-Wasserstein (LPGW) embedding. They begin the paper with a succinct coverage of optimal transport (OT). They further begin with covering the 2-Wasserstein distance, a common choice for OT-based methods, and proceed to lay out the fundamentals of the Gromov-Wasserstein (GW) distance. Next, the authors give a breakdown of the linear embedding technique, linear Gromov-Wasserstein (LGW). The mass constraint on OT methods motivated the creation of partial Gromov-Wasserstein, or PGW. Similar to the linearization of GW, the authors introduce the linearization of PGW through known techniques such as approximations; their work was further inspired by the need of having a faster solution for computing the distance between K different metric spaces. The authors claimed their algorithm can improve computing the PGW distances between K metric measure spaces from O(K^2) to O(K). While the authors have done a great job with the mathematical formulations, they have not provided a time-complexity analysis to back up their statement. Lastly, I believe the authors did not provide sufficient explanations for their experiments. For example, the simulation on the MNIST dataset, no clarifications are given on the details of training. For example, the authors state they will use a logistic regression for classification; this model is meant for binary classification and does not simply work on categorical cases. How exactly did they use logistic regression to predict classes when they range from 0-9? Furthermore, does this work with other models such as neural networks? It is always encouraged to be as specific as possible for reproducibility.

**Strengths:**

The paper mathematically backs up their introduction of the linear partial Gromov-Wasserstein embedding algorithm. They also clearly demonstrate how previous work led to their contributions by breaking down and introducing each step along the way.

**Weaknesses:**

The authors claim that given K metric measure spaces, their algorithm can compute the distance between them in O(K) time. This claim was not supported with a time-complexity analysis and therefore remains unsubstantiated. It would strengthen the paper significantly to add a dedicated section, if even a small one, to prove the time-complexity (perhaps space-complexity if there's a tradeoff).

**Questions:**

Can you provide a time-complexity analysis to support your claim on a linear computation of the distances between K metric measure spaces?

---

> ### Author Response · Authors · 2024-11-20
> **Time complexity**
>
> We appreciate the reviewer’s time and insightful feedback. It seems the
>
> **W1 and Q1: Complexity.** In Appendix C, we discuss the complexity with respect to $K$. Here’s a summary:
>
> Let $n$ be the size of the reference measure and $m$ the average size of all target measures.
>
> - Pairwise computation of PGW/GW distance for $K$ measures requires
> $$\binom{K}{2} \mathcal{O}\left(\frac{1}{\epsilon} nm(n + m)nm^2\right),$$
> where $\epsilon > 0$ is the accuracy tolerance.
>
> - The time complexity for LGW/LPGW is
> $$\binom{K}{2} \mathcal{O}(n^2) + K \mathcal{O}\left(\frac{1}{\epsilon} nm(n + m)n^2 m^2\right).$$
>
> Note that if we apply the Sinkhorn algorithm in the solvers for GW/PGW, the term $nm(n + m)$ can be improved to $\frac{1}{\epsilon} nm \ln(n + m)$.
>
> In general, $n, m \gg K$ (i.e., the size of each dataset is much larger than the number of datasets). Therefore, in (a), the term $\binom{K}{2}(n^2)$ is negligible, and the dominant term is $K \mathcal{O}\left(\frac{1}{\epsilon} nm(n + m)n^2 m^2\right)$, which is linear with respect to $K$.

---

> > ### Author Response · Authors · 2024-11-20
> > **Logistic regression, other models**
> >
> > > How exactly did they use logistic regression to predict classes when they range from 0-9?
> >
> > For this experiment, we use the [scikit-learn logistic regression](https://scikit-learn.org/1.5/modules/generated/sklearn.linear_model.LogisticRegression.html), which automatically performs multi-class classification (with multinomial loss). Since we use this model out-of-the-box with all of its default settings, we originally did not include further details about its usage. We intend to make all of our source code publicly available to ensure that these experiments are reproducible.
> >
> > > Furthermore, does this work with other models such as neural networks?
> >
> > Yes, one could use any model in this experiment that uses the LOT/LGW/LPGW embeddings to perform classification, including neural networks. We chose to use logistic regression for simplicity.

---

### Official Review · Reviewer_ueVA · 2024-10-30

**Soundness:** 3
**Presentation:** 2
**Contribution:** 3
**Rating:** 6
**Confidence:** 3

**Summary:**

This paper introduces the Linear Partial Gromov-Wasserstein (LPGW) embedding, a computationally efficient approximation of the Partial Gromov-Wasserstein (PGW) problem, which relaxes mass constraints for comparing measures across different metric spaces. By reducing the pairwise complexity from $\mathcal{O}(K^2)$ to $\mathcal{O}(K)$, LPGW preserves the partial matching capabilities of PGW while enhancing scalability. The authors provide theoretical guarantees and validate LPGW through experiments in shape retrieval and transport-based learning tasks.

**Strengths:**

1. The paper is well-written and easy to follow.

2. The authors introduce the Linear Partial Gromov-Wasserstein (LPGW) embedding, which addresses the computational challenges of PGW. They also provide a formal proof of the equivalence between LPGW and PGW under the PGW-Monge mapping assumption, contributing to the theoretical foundation of the method.

3. The paper presents a method to approximate LPGW efficiently, reducing computational overhead. Furthermore, the authors conduct extensive numerical experiments to demonstrate the effectiveness and computational advantages of LPGW.

**Weaknesses:**

1. The equivalence between LGW and GW (Proposition 2.1) and between LPGW and PGW (Proposition 3.1) critically relies on the existence of a Monge mapping. However, proving the existence of such a mapping for the GW problem requires satisfying strict conditions [A]. This raises concerns about the robustness of these approximations in practical scenarios where Monge mappings may not exist. The paper would benefit from
    - providing theoretical bounds on the approximation error when a Monge mapping does not exist;
    - including empirical experiments comparing performance on datasets known to violate Monge mapping conditions;
    - discussing potential modifications to make the method more robust when Monge mappings do not exist.

2. The computational implementation of LPGW relies on two layers of approximations. First, the authors approximate LPGW using aLPGW, as introduced in Equation (18). Then, Equation (20) further approximates the aLPGW formulation. However, the paper lacks both theoretical guarantees and empirical evaluation of the error introduced by these sequential approximations. It would be beneficial if the authors could
    - provide theoretical error bounds for each approximation step;
    - include ablation studies quantifying the empirical error introduced at each approximation stage;
    - discuss the tradeoffs between computational efficiency and approximation accuracy.

3. The paper primarily compares LPGW with PGW and LGW. However, other GW approximation methods could offer meaningful insights into the strengths and weaknesses of LPGW. Notably, comparisons with Low-Rank GW [B] and Sliced GW [C] could demonstrate whether LPGW provides any unique computational or approximation advantages.


[A] Dumont, Théo, Théo Lacombe, and François-Xavier Vialard. "On the existence of Monge maps for the Gromov–Wasserstein problem." *Foundations of Computational Mathematics* (2024): 1-48.

[B] Scetbon, Meyer, Gabriel Peyré, and Marco Cuturi. "Linear-time gromov wasserstein distances using low rank couplings and costs." *International Conference on Machine Learning*. PMLR, 2022.

[C] Titouan, Vayer, et al. "Sliced gromov-wasserstein." *Advances in Neural Information Processing Systems* 32 (2019).

**Questions:**

1. Could the authors extend the approximation results for LGW and LPGW to scenarios where the Monge mapping may not exist?

2. Could the authors provide empirical or theoretical results on the error introduced by the two layers of approximation? Specifically, how much error is accumulated when transitioning from LPGW to aLPGW (Equation (18)) and then from aLPGW to the final approximation (Equation (20))?

3. Proposition 3.1 (3) provides a result when the cost function $g_Y$​ is an inner product. Could this result be extended to the case of squared Euclidean distance with additional assumptions? This would be relevant since [A] also explores the existence of Monge mappings under squared Euclidean costs.

4. In the elliptical disks experiment, the choice of reference space has a significant impact on performance. Could the authors provide practical guidance or heuristics for selecting reference spaces to optimize the performance of LPGW?

---

> ### Author Response · Authors · 2024-11-20
> **Weaknesses 1,2,3**
>
> We would like to express our gratitude for the thorough and insightful feedback provided in the review. In what follows, we offer our explanations to the reviewer's questions and concerns. We welcome follow-up inquiries and would be happy to engage in more in-depth discussions.
>
>
> **W1: Monge Mapping Assumption.**
> > Providing theoretical bounds on the approximation error when a Monge mapping does not exist.
>
> To the best of our knowledge, theoretical bounds for LGW and LPGW have not yet been established. This bound will likely depend heavily on the reference measure. Empirically, we observe that increasing the size of the reference measure (i.e., increasing "resolution") helps decrease the error.
>
> > Including empirical experiments comparing performance on datasets known to violate the Monge mapping conditions.
>
> In Table 2, we present the relative error and Pearson correlation coefficient for this purpose. Note that in this experiment, **we do not have a Monge mapping for any of the embeddings since all reference measures and target measures have distinct sizes, and some reference measures are non-uniform.** We observe that, when the reference measure is appropriate (e.g., $\mathbb{S}_7, \mathbb{S}_8, \mathbb{S}_9$), the relative error is low (0.02) and the PCC is high (0.99).
>
> > Discussing potential modifications to make the method more robust when Monge mappings do not exist.
>
> In our understanding, the existence of a Monge mapping is not essential since this situation can occur with the LOT, LGW, and LHK methods, and in these cases, barycentric projection is generally used. In fact, LPGW (or aLPGW) is viewed as a proxy for the original PGW distance rather than an exact approximation.
>
> Our approach for improving accuracy and preserving information in the original target measure includes the following strategies, as discussed in lines 2217-2252. To summarize, accuracy can be enhanced by:
>
> 1. Choosing a good reference measure:
>    - The size of the reference measure acts as the "resolution" of the embedding, so larger sizes are preferred. Typically, we set this size to be either the maximum or mean/median of the sizes of all target datasets.
>
>    - Similar to linear OT and linear GW, the reference measure should ideally be set to the GW/PGW barycenter, as explained in lines 2364-2366, 2433-2338, and 2476-2478.
>
> 2. Setting the mass of each point in the reference and target measures to be equal if feasible.
>
>
>
> ------------
>
> **W2: Computational Implementation.** We appreciate the reviewer’s observation and suggestion. To clarify the "two-layer approximation":
>
> - (20) is an approximation of (19), aLPGW.
> - aLPGW (19) approximates LPGW (15).
> - LPGW (15) approximates PGW (11).
>
> This relationship explains the derivation of (20). However, it is not a multi-layers approximation. (15)(19) are only proposed for theoretical completeness and is not computational feasible.
>
>
> In practice, we directly approximate PGW with (20), rather than through (19) or (15), which were proposed for theoretical completeness, similar to LOT and LGW.
>
> The above approximation is similar to LOT/LGW. That is LOT, aLOT, LGW, aLGW are only proposed for theoretical completeness. The practical formulations are
> $$aLOT(\mu,\nu)\approx \|\mathcal{T}_{\gamma^1}-\mathcal{T}_{\gamma^2}\|_{L(\mu)}^2$$
>
> $$aLGW(\mu,\nu)\approx \|\mathcal{T}_{\gamma^1}-\mathcal{T}_{\gamma^2}\|_{L(\mu^{\otimes2})}^2.$$
>
>
>
> > Provide theoretical error bounds.
>
> Developing a theoretical error bound is part of our future work. To our knowledge, there is a gap in this area for both LGW and LPGW.
>
> > Discuss the trade-offs between computational efficiency.
>
> Similar to LOT and LGW, both (15) and (19) are computationally inefficient (in fact, there is not an existing solver for (15) or (19)). These formulations are presented primarily for theoretical completeness and to potentially support future developments (e.g., LPGW geodesics). Thus, only (20) is computationally efficient.
>
> ------------
>
> **W3: Comparisons with Existing Methods.** LPGW is proposed as a proxy for Partial GW distance, and, to the best of our knowledge, low-rank GW and sliced-GW have not been extended to the partial GW setting.
>
> Additionally, sliced-GW [C] solves a variant of the GW problem:
>   $$\sum_{l=1}^L GW((\theta_l)_\#\mathbb{X}, (\theta_l)_\#\mathbb{Y}).$$
>
> Apart from the difference between GW and PGW, there are other issues:
>    - The optimal value in the formulation above represents the average transport cost based on 1D projections, which can differ significantly from the transport cost in the original GW/PGW problem.
>    - Theorem 3.1 in [C] requires the Monge mapping assumption. However, in our ellipse experiment (Section 4.1) and MNIST classification experiment (Section 4.3), a Monge mapping does not exist and so the closed form in their main theorem cannot be applied.

---

> ### Author Response · Authors · 2024-11-20
> **Question 1,2,3,4**
>
> **Q1: Approximation Results without Monge Mapping Assumption.** In the general case without Monge mapping, LPGW is defined in (55) and LGW in (36). In the approximation error experiment (Section 4.1, Table 2), **a Monge mapping does not exist for any embedding** since all the reference measures and target measures have distinct sizes and some of the reference measures are non-uniform. See the [repo](https://anonymous.4open.science/r/Linearized_Partial_Gromov_Wasserstein-F43E/README.md), file `ellipse/ellipses.ipynb` for details.
>
> ------------
>
> **Q2: Approximation Errors.** As explained above, LPGW (15) and aLPGW (18) are proposed for theoretical completeness and are computationally infeasible. Thus, we use (20) to approximate PGW. This situation is similar for LOT and LGW.
>
> In the LOT paper, both LOT Eq. (27) and aLOT Eq. (32) are inefficient to solve. The only feasible implementation is the approximation of (32):
> $$\|\mathcal{T}_{\gamma^1} - \mathcal{T}_{\gamma^2}\| \tag{a}$$
>
> Strictly speaking, (a) is neither aLOT nor LOT. However, (a) is used directly as the representation of LOT in much of the LOT literature. In LGW/LPGW, the situation is the same.
>
> ------------
>
> **Q3: Extensions of Cost Function.** To the best of our knowledge, the optimality of barycentric projection has been proven only for LOT and linear partial OT. We plan to extend this result to Euclidean cost in future work. In GW/PGW, we just proposed the proof for inner product. In the future, we will extend this result to more general cost.
>
> In our understanding, to extend the result to general cost, the definition of barycentric projection may need to be modified.
>
> ------------
>
> **Q4: Elliptical Disks Experiment.** A suitable reference measure can be chosen as follows:
>
> - The reference measure size acts as the "resolution" of the embedding, so a larger size is preferred. Typically, we set this to be the maximum/mean/median of the sizes of all target datasets.
>
> - Similar to linear OT and linear GW, the reference measure should ideally be the GW/PGW barycenter, as explained in lines 2364-2366, 2433-2338, and 2476-2478. Corresponding barycenter algorithms can be found in [Peyre et al., 2016](https://proceedings.mlr.press/v48/peyre16.pdf), [Bai et al., 2024](https://arxiv.org/abs/2402.03664).

---

> > ### Comment · Reviewer_ueVA · 2024-12-03
> >
> > I thank the authors for their detailed response.  I will keep my score unchanged.

---

### Official Review · Reviewer_wRQx · 2024-11-04

**Soundness:** 3
**Presentation:** 3
**Contribution:** 3
**Rating:** 6
**Confidence:** 4

**Summary:**

The paper proposes a modification of a variant of the Gromov-Wasserstein (GW) distance, known as the Partial GW (PGW) problem,  which enables to compare gauged measure (gm) spaces whose measures have different total masses. The modification consists of applying the linearization strategy proposed in (Beier et al., 2022) to the PGW problem, yielding the so-called Linear PGW discrepancy (LPGW). It is shown that if the PGW problem admits a solution that is induced by a deterministic map, then the LPGW problem recovers the PGW problem. For the purposes of computation, an approximated LPGW problem is considered. The paper concludes with some numerical experiments which exhibit the empirical performance of the proposed approach.

**Strengths:**

The paper is, in my opinion well-written, and the theoretical findings appear correct. In the experiments, the LPGW algorithm compares favorably to the other approaches considered.

**Weaknesses:**

I believe that the paper is of a limited novelty. Indeed, both the linearized GW problem and the partial GW problem have appeared in previous works. Combining these two approaches to obtain a linearized partial GW problem appears to be incremental progress on the important question of efficient computation for the GW problem.

Furthermore, the theoretical properties of the LPGW problem appear quite limited. Many of the results require the existence of a Monge map for the PGW problem, but sufficient conditions for these assumptions to hold are not discussed in the main text. This assumption is quite strong and, as such, these results appear to be of a limited applicability.

The approximated LPGW is, in practice approximated as (20). This appears to be a very coarse approximation in general; no justifications are provided for when this might be a reasonable approximation.

Finally, the experiments do not present any significantly new applications. The experiments in 4.1 and 4.2 are essentially the same as those from (Beier et al., 2022) the experiment in 4.3 is similar to that  from S. Nieter, R. Cummings, Z. Goldfeld, Outlier-Roubust Optimal Transport: Duality, Structure, and Statistical Analysis, AISTATS 2022. The main distinction is that a random rotation is performed on the dataset in the current paper, but this appears a bit contrived in the setting of MNIST.

**Questions:**

1. Many results in the paper depend on the assumption that a Monge map exists. Can the authors comment on sufficient conditions for such a result to hold?

2. Proposition 3.3 holds when $\lambda$ is sufficiently large. Can the authors clarify this condition further in the main text? Moreover, if $\lambda$ is too large, the regularizer dominates so there appears to be some tradeoff in terms of minimizing the first part of the objective.

3. In the numerical experiments, in the experiments on MNIST, it seems that PGW would offer the best comparison (vs LOT and LGW which do not account for noise). Could these results be included?

4. I believe that the robustness to noise is the most interesting feature of the PGW formulation. I would recommend that this be addressed more clearly in the text.

---

> ### Author Response · Authors · 2024-11-20
> **Weakness 1, novelty**
>
> We would like to thank the reviewer for their feedback. We hope the following adequately addresses the reviewer's concerns and would be more than happy to engage in further discussions.
>
> **W1: Novelty.** First, we would like to clarify a potential misunderstanding of the LPGW formulation and our main contributions. The following will be emphasised in the main text and abstract.
>
> From a theoretical perspective, our proposed LPGW formulation consists of two parts: the **LPGW embedding** and the **LPGW distance**.
> - The LPGW embedding is the primary contribution of this paper, and we numerically demonstrate that this embedding is robust to rotation/flipping/noise corruption, compared with the LOT embedding or the LGW embedding. In addition, the embedding can be used to reconstruct (denoised) data (see [the following figure](https://anonymous.4open.science/r/Linearized_Partial_Gromov_Wasserstein-F43E/mnist2d/results_visual/embedding.png), which we plan to add to the paper). Meanwhile, the embedding is orthogonal to PGW, as OT/GW/PGW does not provide such an embedding.
>
> - The LPGW distance is a byproduct of the LPGW embedding. This distance can be treated as a proxy/approximation of PGW, and it significantly improves the computational complexity.
>
> We would like to clarify that the LPGW distance is not the primary contribution of our work; rather, it is a byproduct of the proposed LPGW embedding. Specifically, our main contribution lies beyond providing an approximation for the PGW distance, as the reviewer's comment might suggest.
>
> >Combining these two approaches to obtain a linearized partial GW problem appears to be incremental progress on the important question of efficient computation for the GW problem.
>
>
> The authors respectfully disagree with the opinion that the proposed LPGW is merely a combination of PGW and LGW.
>
> In the classical OT/GW setting, the problem admits a dynamic formulation, where the embedding is defined through the logarithm mapping. This mapping incorporates both the velocity of mass transportation and the derivative related to mass creation and destruction (noting that in balanced OT/GW, the derivative for mass creation/destruction is zero). However, in UGW, including PGW, a dynamic formulation has not yet been explored. Incorporating this dynamic information to define the embedding and establishing a similarity measure between embeddings that can recover LGW and PGW in specific settings remains a significant challenge in the field.
>
> From a computational perspective, we demonstrate that for $K$ different metric measure spaces, computing their pairwise PGW distances requires solving the PGW problem $\mathcal{O}(K^2)$ times. In contrast, computing the LPGW distances pairwise requires only $\mathcal{O}(K)$ distance computations, representing a dramatic improvement in computational efficiency.
>
> In addition, our work completes the theoretical analysis of linear GW in the [original paper](https://arxiv.org/abs/2112.11964) regarding barycentric projection (see Proposition 2.1 (3)), which is essential for explaining why the barycentric projection mapping can be used in the approximate aLGW formulation (e.g., (10)). We then extend this result to the PGW setting (see Proposition 3.2).

---

> ### Author Response · Authors · 2024-11-20
> **Weakness 2, Question 1, Monge mapping**
>
> > Many of the results require the existence of a Monge map for the PGW problem.
>
> To clarify, similar to LOT/LGW, **none of the formulations for the LPGW embedding technique (e.g., LPGW distance, LPGW embedding, aLPGW distance, etc.) rely on the Monge mapping assumption**. The formulation of the LPGW distance (e.g., Eq. (14)), which requires a Monge mapping, is presented in the main text only as a simplified formulation for the reader’s convenience. However, when a Monge mapping does not exist, these concepts can still be well-defined. We would be happy to discuss this in further detail if the reviewer so pleases.
>
>
> The theoretical results requiring the Monge mapping assumption are:
> - Proposition 3.1 (2), (3), (4)
> - Theorem 3.2 (3)
>
> Proposition 3.1 (2) characterizes the relationship between the barycentric projection and the Monge mapping, and we felt its inclusion important for the reader to understand the barycentric projection.
>
> Proposition 3.1 (3), (4), and Theorem 3.2 (3) address the gap between the proposed (approximated) LPGW distance and the PGW distance. In Proposition 3.1 (3), we clarify the conditions under which the LPGW distance can exactly recover the PGW distance. Proposition 3.1 (4) shows when the numerical implementation (Eq. (20)) can recover PGW, and Theorem 3.2 (3) states the conditions for aLPGW to recover LPGW.
>
> **Conditions for the existence of Monge mapping**: In the continuous measure setting and the empirical discrete measure setting (i.e., where the mass of each point in $\mu$ and $\nu$ is the same), GW admits a Monge mapping (e.g., [Dumont et al., 2024](https://arxiv.org/pdf/2210.11945), Proposition 1.16, Theorem 2.1; [Courty et al., 2019](https://arxiv.org/pdf/1905.10124), Theorem 3.2). However, it is unclear whether this result extends to unbalanced GW (including PGW), as these formulations were proposed only recently. Further characterization of the conditions under which the Monge mapping assumption hold is an open problem left for future work.
>
> **Note:** Empirically, we observe that both GW and PGW admit Monge mappings when data are in an empirical distribution.
>
> > These results appear to be of limited applicability.
>
> We respectfully disagree with this point. Based on our understanding, the only effect of the absence of Monge mappings in PGW is an increased gap between the LPGW distance and PGW distance (see lines 2231-2234 in the appendix). In a machine learning setting, when target datasets have different sizes or the reference measure has non-uniform distribution, then the Monge mapping assumption may not hold in all linear OT methods, including [LOT](https://link.springer.com/article/10.1007/s11263-012-0566-z), [Linear HK](https://arxiv.org/abs/2102.08807), [Linear OPT](https://arxiv.org/abs/2302.03232), and [LGW](https://arxiv.org/abs/2112.11964). However, these methods are still viable and of broad importance.
>
> (We would like to briefly remark, however, that if we manually set the mass for each point in all datasets to be the same, the Monge mapping assumption will still hold in the linear unbalanced OT setting, including our LPGW.)
>
> Nonetheless, similar to Linear OT/Linear GW, we present the LPGW distance as a proxy for PGW rather than as an exact approximation. In fact, in our shape retrieval experiment, the PGW distance performs worse than LPGW distance in measuring similarity in the 2D dataset (see Table 2).

---

> ### Author Response · Authors · 2024-11-20
> **Weakness 3, approximation**
>
> **W3: Approximation of LPGW.** Equation (20) is the empirical implementation of the LPGW distance, formulated analogously to the empirical implementation of the LGW distance (see line 215).
>
> In Table 1, we show the relative error gap (MRE) and Pearson correlation (PCC) between (20) and PGW. We find that when the reference measure is well-chosen (i.e., $\mathbb{S}_7, \mathbb{S}_8, \mathbb{S}_9$), (20) and PGW exhibit low MRE (0.02) and high PCC (0.99). **Note that in this experiment, we do not have a Monge mapping for any embedding, since all the reference measures and target measures have distinct sizes and some of the reference measures are non-uniform. See the [repo](https://anonymous.4open.science/r/Linearized_Partial_Gromov_Wasserstein-F43E/ellipses/ellipses.ipynb), file `ellipse/ellipses.ipynb` for details.**
>
> Using approximation formulations to replace the original linear OT/GW/PGW is common in other linear OT-based techniques as well.
>
> For example, in the LOT method, LOT (Eq. (27)) is proposed to approxiamte OT. aLOT (Eq. (32)) is then proposed to approximate LOT.
>
> Furthermore, the aLOT (Eq. (32)) is approximated by:
> $$\|\mathcal{T}_{\gamma^1}-\mathcal{T}_{\gamma^2}\|^2\tag{b}.$$
>
> Note, to clarify it, it is not a multi-layers approximation. LOT (27) and aLOT (32) are only proposed for theoretical completeness and are not computational feasible.
>
> In practice, (b) is used to approximate OT, though (b) is neither LOT (27) nor aLOT (32).  However, in most of the existing literature that cites [LOT](https://link.springer.com/article/10.1007/s11263-012-0566-z), (b) is generally treated as the LOT distance and (27), (32) are not presented.
>
> Our approximation (20) follows analogously from the previous works.

---

> > ### Author Response · Authors · 2024-11-20
> > **Weakness 4: Experiments.**
> >
> > **W4: Experiments.** Given that LPGW is an extension of LGW to the partial GW setting, our first aim is to demonstrate the performance of LPGW in existing experiments, such as the shape retrieval experiment in the [LGW paper](https://arxiv.org/pdf/2112.11964). In this experiment, we demonstrate that when the tested shapes have different numbers of points, LPGW achieves better accuracy with faster speed. Our aim here is not to invent completely new expiriments; rather, we deliberately choose existing experiments for better comparison.
> >
> > > The experiment in 4.3 is similar to that from S. Nieter, R. Cummings, Z. Goldfeld, Outlier-Roubust Optimal Transport: Duality, Structure, and Statistical Analysis, AISTATS 2022.
> >
> > The authors respectively disagree with this opinion. In the MNIST classification experiment (Section 4.3), the experiment setup differs from the generative model (GANs) experiment in [Nieter et al., 2023](https://arxiv.org/pdf/2111.01361). Furthermore, the "outlier-robust Wasserstein distance" (Eq 1) which is used in [Nieter et al., 2023](https://arxiv.org/pdf/2111.01361) can be treated as a variant of partial OT, and is not robust to rotation and flipping.
> >
> >
> > The problem setup of Experiment 4.3 is about the classification of data (e.g. MNIST digits) when the data in the testing set is corrupted by   rotation/flipping/random noise. The flipping/rotation corruption is not considered in [Nieter et al., 2023](https://arxiv.org/pdf/2111.01361).
> >
> > > This appears a bit contrived in the setting of MNIST.
> >
> > To the best of our knowledge, besides noise corruption, classifying rotated/flipped digits is also a common problem in machine learning research, as seen in, e.g. [Zhou et al, 2017](https://arxiv.org/abs/1701.01833). We believe the inclusion of random rotations/flipping in our work is important to demonstrate the robustness of our method; this was likely not included in [Nieter et al., 2023](https://arxiv.org/pdf/2111.01361) and the method presented there is not robust to rotation/flipping.

---

> > > ### Author Response · Authors · 2024-11-20
> > > **Questions 2,3,4**
> > >
> > > **Q2: Role of $\lambda$ in Proposition 3.3.** The authors apologize for this unclear explanation in Proposition 3.3. We will update the statement in proposition 3.1 (3) and 3.3. In particular, "lambda is sufficiently large" will be replaced by the following:
> > >
> > > $$2\lambda \ge \max_{y_i^1,y_{i'}^1\in Y^1,y^2_j,y^2_{j'}\in Y^2}|g_{Y^1}(y^1_{i'},y_i^1)-g_{Y^2}(y_{j}^2,y_{j'}^2)|^2.\tag{c}$$
> > >
> > >
> > > > Moreover, if $\lambda$ is too large, the regularizer dominates so there appears to be some tradeoff in terms of minimizing the first part of the objective.
> > >
> > > When $\lambda$ is greater than the above threoushold, the maximum amount of mass will be transported, meaning that problem (19) has a closed-form solution. This ensures that all transportation plans for problems $PGW(\mathbb{X}, \mathbb{Y}^1)$, $PGW(\mathbb{X}, \mathbb{Y}^2)$, and (19) remain unchanged once $\lambda$ exceeds this threshold. Intuitively, this implies no further increase in $\lambda$ is needed. Thus, the largest value for $\lambda$ in our experiments is based on this bound.
> > >
> > >
> > > ------------
> > >
> > > **Q3: MNIST Experiment.** There appears to be a potential misunderstanding regarding the experiment presented in Section 4.3. In this experiment, we train a logistic regression model using the embeddings from LOT/LGW/LPGW. Since OT/GW/PGW are not embedding techniques and do not provide embeddings, they cannot be used in this experiment.
> > >
> > > If the goal is to use OT/GW/PGW to train a classifier, we can employ models such as a kernel-SVM, as demonstrated in the shape retrieval experiment (Section 4.2).
> > >
> > > The main difference between LPGW and PGW lies in their computational complexity.
> > > In summary, LPGW requires approximately 3–5 minutes to train the model, whereas PGW demands ~400 hours (over 15 days). Due to the time constraints of the review period, we were unable to provide classification results using PGW. However, the code is available in the [repository](https://anonymous.4open.science/r/Linearized_Partial_Gromov_Wasserstein-F43E/README.md), `mnist2d/classification.ipynb`.
> > >
> > > The following is a summary of the difference in complexity:
> > >
> > > Suppose the size of training dataset and testing dataset are $N_{train},N_{test}$ respectively. In this experiment $N_{train}=4000, N_{test}=1000$.
> > >
> > > - Training a kernel-SVM model for GW/PGW requires
> > > $\mathcal{O}(N^2_{train}T)$ where $T$ is the complexity for one GW/PGW computation. The testing step requires $\mathcal{O}(N_{test}N_{train}T)$ complexity.
> > >
> > > - For the LPGW model, the training process requres only $\mathcal{O}(N_{train}T)$ and the testing step requires $\mathcal{O}(N_{test}T)$.
> > >
> > > This means that the larger the (training and testing) dataset, the more pronounced the speed advantage of LPGW compared to PGW.
> > >
> > > ------------
> > >
> > > **Q4: Robustness to Noise.** The authors suspect the reviewer's question is about LPGW, not PGW.
> > >
> > > We will add a section to explain the intuitive noise-robustness feature of LPGW.
> > >
> > > Note, the robustness to noise is property of PGW due to its partial matching property. In PGW, $\lambda$ plays a role as threshold, when the "transportation distance" for a certain pair is greater then $2\lambda$, this pair is unlikely to be transported. When the noise points has larger distance to the clean data, these points are unlike to be transported/paired.
> > >
> > > By extension, LPGW utilizes this property to contruct the embedding. That is, with high probability, outliers/noise will not be incoporated when we construct the LPGW embedding. Thus the affect of noise in downstream machine learning tasks (e.g. classification) will be limited.
> > >
> > > We also refer to [the following figure](https://anonymous.4open.science/r/Linearized_Partial_Gromov_Wasserstein-F43E/mnist2d/results_visual/embedding.png) for a visulization of the LPGW embedding. We can observe that, when a digit is corrupted with noise, the reconstructed figure based on the LPGW embedding does not contain most of the noise points.

---

> > > > ### Comment · Reviewer_wRQx · 2024-11-24
> > > >
> > > > I thank the authors for their detailed response to my questions. I agree with many of the points raised and hence have updated my score to reflect this. I recommend that the authors clarify these points further in the main text.

---

> ### Author Response · Authors · 2024-11-25
> **comment**
>
> The authors sincerely thank the reviewer for their valuable comments and constructive discussion. The paper will be updated to include the detailed discussion on Monge mapping, approximation error and MNIST experiments. Additionally, the statement of Proposition 3.3 and the citation issue raised by the reviewer has been addressed/updated.

---

### Official Review · Reviewer_K3go · 2024-11-04

**Soundness:** 3
**Presentation:** 2
**Contribution:** 2
**Rating:** 6
**Confidence:** 4

**Summary:**

This paper studies the linear partial Gromov-Wasserstein (GW) alignment problem. Specifically the paper extends ideas from linearized GW and partial GW, and combines them to LPGW which inherits the merits of both method and is applicable to wider tasks. To formulate, the paper gives detailed account for how to incorporate general measure instead of probability measures, and how to coordinate different marginals. Extensive numerical experiments are presented, showcasing better performance.

**Strengths:**

The paper is well motivated and the study of the proposed formulation is very thorough, covering both general and special cases under suitable assumptions. The proposed algorithm combines the advantages from both partial GW and linearized GW, and extensive experiments supports this extension.

**Weaknesses:**

1. The paper, though overall well motivated, is mostly combining existing methods and theoretical novelty is limited, with discussions following from the the original ideas and definitions of linearized/partial GW/OT. A further question: what if TV is replaced by a general divergence for unbalanced formulation as in [1]?


2. Discussion of related works: the term "linearization" is from the tangent structure of gauged GW space as is first pointed out in [2]. Mapping based unbalanced GW formulations are also studied in [3],[4].


3. Minor comment:
If the barycentric LGW is the version to use as in the second equality of (10), then this should be stated in the paper.
Notation $\gamma^1\wedge\gamma^2$ on line 260 seems to be incorrect and should be $\inf_A \gamma^1(A\cap E)+\gamma^2(A^C\cap E)$.

[1] Séjourné, Thibault, François-Xavier Vialard, and Gabriel Peyré. "The unbalanced gromov wasserstein distance: Conic formulation and relaxation." Advances in Neural Information Processing Systems 34 (2021): 8766-8779.

[2] Sturm, Karl-Theodor. The space of spaces: curvature bounds and gradient flows on the space of metric measure spaces. Vol. 290. No. 1443. American Mathematical Society, 2023.

[3] Hur, YoonHaeng, Wenxuan Guo, and Tengyuan Liang. "Reversible Gromov–Monge sampler for simulation-based inference." SIAM Journal on Mathematics of Data Science 6.2 (2024): 283-310.

[4] Zhang, Zhengxin, et al. "Cycle consistent probability divergences across different spaces." International Conference on Artificial Intelligence and Statistics. PMLR, 2022.

**Questions:**

See above.

---

> ### Author Response · Authors · 2024-11-20
> **Weaknesses 1, 2, 3**
>
> **W1: Novelty of this paper**:
> > The paper is mostly combining existing methods and theoretical novelty is limited, with discussions following from the the original ideas and definitions of linearized/partial GW/OT.
>
> While the general principle of linear partial GW is inspired by linear GW/OT, our proposed formulation of the LPGW embedding is not as simple as combining existing techniques.
>
> Notably, for prior linear GW/OT works in the balanced setting, the linear embedding represents the displacement between the reference measure and the given target measures based on the optimal transport plans.
>
> In our work, it is essential to incorporate information about mass creation and destruction into the embedding. Formulating such an embedding within the partial GW setting is highly non-trivial. As such, we feel that describing our approach as simply a combination of existing methods does not fully capture its unique contributions.
>
> In the classical OT/GW/UOT framework, the problem admits a dynamic formulation. Consequently, the linear embedding can be interpreted as a logarithm mapping, capturing the velocity of particles involved in mass transportation as well as the derivatives governing mass creation and destruction. However, the dynamic formulation for UGW/PGW has not yet been thoroughly explored. Incorporating this dynamic information to define an embedding and establishing a similarity measure between two embeddings that can recover LGW and PGW is a significant challenge.
>
> In contrast, these challenges do not arise in the balanced OT/GW setting, where the formulation is inherently simpler.
>
> We kindly encourage the reviewer to reconsider this perspective.
>
> > What if TV is replaced by a general divergence for unbalanced formulation as in [1]?
>
> As explained above, defining a new embedding in the unbalanced GW setting based on general divergence penalty is non-trivial. To the best of our knowledge, linear unbalanced GW has not been studied in any setting, regardless of whether the penalty is TV or another divergence (e.g., KL divergence). Even in classical OT setting, the unblanced linear OT embedding has only been studied with the KL penalty or the TV penalty (see e.g., [Linear HK](https://arxiv.org/abs/2102.08807) and [Linear OPT](https://arxiv.org/abs/2302.03232)); a formulation using general divergences has not yet been proposed.
>
>
> **W2: Additional References.** We appreciate the reviewer’s observation. We would like to remark that [2,3] are already cited in our work (note that they do not directly address unbalanced GW however). We will add [4] to our related works section. (See the updated pdf.)
>
> ------------
>
> **W3: Barycentric LGW & Notation.** LGW’s theoretical formulation is (8), and its approximation is (10). Equation (10) can be further simplified as:
> $$\|g_{Y^1}(\mathcal{T}_{\gamma^1}(\cdot_1),\mathcal{T}_{\gamma^1}(\cdot_2))-g_{Y^2}(\mathcal{T}_{\gamma^2}(\cdot_1),\mathcal{T}_{\gamma^2}(\cdot_2))\|^2_{L(\mu^{\otimes2})} \tag{a}$$
> (We've added a label to this equation in the paper.)
>
> In all experiments, the numerical implementation of LGW is based on (a) rather than (8) or (10). The code in [Github](https://github.com/Gorgotha/LGW) also uses (a), similar to LOT and LPGW. We will add a sentence to clarify this point.
>
> > Notation $\gamma^1 \wedge \gamma^2$ on line 260 seems to be incorrect and should be $\inf_A \gamma^1(A\cap E)+\gamma^2(A^C\cap E)$.
>
> We thank the reviewer for this comment and have corrected the typo.

---

> ### Author Response · Authors · 2024-11-25
> **comment**
>
> The authors sincerely thank the reviewer for their valuable comments and thoughtful discussion. The additional reference [4] has been added, and the typo in line 260 has been corrected.
>
> The dynamic formulation of PGW is part of our future research direction. We aim to explore how our new formulation aligns with the tangent space and GW geodesics discussed in [Sturm](https://arxiv.org/abs/1208.0434), [Beier et al., 2021](https://arxiv.org/abs/2112.11964), and [Beier et al., 2024](https://arxiv.org/abs/2403.08612).
>
> However, to the best of our knowledge, there is currently no dynamic formulation for GW (specifically, the related continuity equation) in the general gauge measure space setting. Dynamic GW in a special setting has been studied very recently (see [Zhang et al., 2024](https://arxiv.org/pdf/2407.11800?)).
>
> In conclusion, we agree with the reviewer on the importance of having a dynamic formulation for PGW, and it will be part of our future work.

---

### Official Review · Reviewer_HYNr · 2024-11-25

**Soundness:** 4
**Presentation:** 4
**Contribution:** 3
**Rating:** 8
**Confidence:** 1

**Summary:**

The authors propose the linear partial Gromov-Wasserstein (LPGW) embedding, a linearization technique for the PGW problem. Theoretically, they prove that LPGW admits a metric with certain assumptions, and numerically, they demonstrate that the utility of the proposed LPGW method in shape retrieval and learning with transform-based embedding tasks. They illustrate that the LPGW- based method can preserve the partial matching property of PGW, while significantly improving computational efficiency.

**Strengths:**

1. The paper is well organized and easy to follow.
2. The paper contains many details, which is very helpful for understanding.
3. The experiment is promising, justifying theoretic part.

**Weaknesses:**

I am not an expert in this area, I don't find weaknesses from my perspective.

**Questions:**

I am not an expert in this area.

---

### Author Response · Authors · 2024-11-20
**Response to common questions and concerns**

# Main Contribution
From a theoretical perspective, our proposed LPGW formulation consists of two parts: the **LPGW embedding** and the **LPGW distance**.
- The LPGW embedding is the primary contribution of this paper, and we numerically demonstrate that this embedding is robust to rotation/flipping/noise corruption, compared with the LOT embedding or the LGW embedding. In addition, the embedding can be used to reconstruct (denoised) data (see [the following figure](https://anonymous.4open.science/r/Linearized_Partial_Gromov_Wasserstein-F43E/mnist2d/results_visual/embedding.png)). Meanwhile, OT/GW/PGW does not provide such an embedding.

- The LPGW distance is a byproduct of the LPGW embedding. This distance can be treated as a proxy/approximation of PGW, and it significantly improves the computational complexity.

We would like to clarify that the LPGW distance is only a byproduct of the proposed embedding. **Specifically, the main contribution lies beyond providing an approximation for the PGW distance, as the reviewer's comment might suggest.**

# Novelty of the paper
While the general principle of linear partial GW is inspired by linear GW/OT, our proposed formulation of the LPGW embedding is not as simple as combining existing techniques.

Notably, for prior linear GW/OT in the balanced setting, the linear embedding represents the displacement between the reference measure and the given target measures based on the optimal transport plans.

In our work, it is essential to incorporate information about mass creation, transportation and destruction into the embedding. Formulating such an embedding within the partial GW setting is highly non-trivial. As such, we feel that describing our approach as simply a combination of existing methods does not fully capture its unique contributions.

In the classical OT/GW/UOT framework, the problem admits a dynamic formulation. Consequently, the linear embedding can be interpreted as a logarithm mapping, capturing the velocity of particles involved in mass transportation as well as the derivatives governing mass creation and destruction. However, the dynamic formulation for UGW/PGW has not yet been thoroughly explored. Incorporating this dynamic information to define an embedding and establishing a similarity measure between two embeddings that can recover LGW and PGW is a significant challenge.

In contrast, these challenges do not arise in the balanced OT/GW setting, where the formulation is inherently simpler.

# Monge mapping assumption (We refer Appendix C in the updated pdf for details.)
To clarify, similar to LOT/LGW, **none of the formulations for the LPGW embedding technique (e.g., LPGW distance, LPGW embedding, aLPGW distance, etc.) rely on the Monge mapping assumption**. The formulation of the LPGW distance (e.g., Eq. (14)), which requires a Monge mapping, is presented in the main text only as a simplified formulation for the reader’s convenience. However, when a Monge mapping does not exist, these concepts can still be well-defined. We would be happy to discuss this in further detail if the reviewer so pleases. Lack of Monge mapping may affect the gap between PGW and LPGW, and there is no other affections.

# Approximation error  (We refer appendix J in the updated pdf for details)
The relation between PGW, LPGW, aLPGW can be described as follows:
- (20) is an approximation of (19), aLPGW.
- aLPGW (19) approximates LPGW (15).
- LPGW (15) approximates PGW (11).

However, it is **not a multi-layers approximation**. (15)(19) are only proposed for theoretical completeness and is not computational feasible.

Using approximation formulations to replace the original linear OT/GW/PGW is common in other linear OT-based techniques as well.
For example, in the LOT method, LOT (Eq. (27)) is proposed to approxiamte OT. aLOT (Eq. (32)) is then proposed to approximate LOT.
Furthermore, the aLOT (Eq. (32)) is approximated by:
$$\|T_{\gamma^1}-T_{\gamma^2}\|^2\tag{b}.$$
In practice, (b) is used to approximate OT, though (b) is neither LOT (27) nor aLOT (32).  However, in most of the existing literature that cites [LOT](https://link.springer.com/article/10.1007/s11263-012-0566-z), (b) is generally treated as the LOT distance and (27), (32) are not presented.
Our approximation follows analogously from the previous works.

In Table 2, we present the relative error and Pearson correlation coefficient for this purpose. **we do not have a Monge mapping in this experiment.** We observe that, when the reference measure is appropriate (e.g., $\mathbb{S}_7, \mathbb{S}_8, \mathbb{S}_9$), the relative error is low (0.02) and the PCC is high (0.99). Applying GW/PGW barycenter can help to find a good reference.

---

> ### Author Response · Authors · 2024-11-27
>
> # LPGW VS PGW (We refer Appendix M.2 in the updated pdf for details discussion)
> LPGW utilize the partial matching property of PGW while significantly improving computational efficiency. The larger the dataset, the more pronounced the computational advantage becomes. For instance, in the MNIST classification task with a training dataset of 4000 samples, LPGW trains the model in 5 minutes, whereas PGW takes approximately 15 days.

---

### Meta-Review · Area_Chair_rMnP · 2024-12-17

**Metareview:**

The paper proposes the Linear Partial Gromov-Wasserstein embedding, which significantly improves the computational efficiency of Partial Gromov-Wasserstein distance while preserving its partial matching properties. Reviewers initially raised concerns regarding the novelty of the work, questioning whether the combination of linearization and partial GW techniques constituted a significant advancement. Additionally, the reviewer noted the reliance on strong assumptions, such as the existence of Monge maps, and the coarse nature of the approximations. However, the authors clarified that their primary contribution lies in the embedding formulation, distinct from PGW distance, and provided empirical results to validate its robustness and practical utility, even in scenarios where Monge mappings do not exist.

**Additional Comments On Reviewer Discussion:**

During the rebuttal period, the authors addressed the reviewers’ concerns thoroughly, particularly by clarifying the theoretical role of approximations, improving explanations of Monge mapping assumptions, and highlighting LPGW’s computational advantages. These clarifications and the strong experimental results showing the method’s efficiency and robustness solidified confidence in the paper’s contributions, leading to the final recommendation for acceptance.

---

### Decision · Program_Chairs · 2025-01-22

Accept (Poster)